# Parallel Simulation for Log-concave Sampling and Score-based Diffusion Models

**Huanjian Zhou** [1 2]   **Masashi Sugiyama** [2 1]

## Abstract

Sampling from high-dimensional probability distributions is fundamental in machine learning and statistics. As datasets grow larger, computational efficiency becomes increasingly important, particularly in reducing *adaptive complexity*, namely the number of sequential rounds required for sampling algorithms. While recent works have introduced several parallelizable techniques, they often exhibit suboptimal convergence rates and remain significantly weaker than the latest lower bounds for log-concave sampling. To address this, we propose a novel parallel sampling method that improves adaptive complexity dependence on dimension $d$ reducing it from $\widetilde{\mathcal{O}}(\log^2 d)$ to $\widetilde{\mathcal{O}}(\log d)$. Our approach builds on parallel simulation techniques from scientific computing.

## 1. Introduction

We study the problem of sampling from a probability distribution with density $\pi(\boldsymbol{x}) \propto \exp(-f(\boldsymbol{x}))$ where $f : \mathbb{R}^d \to \mathbb{R}$ is a smooth potential. We consider two types of setting. **Problem (a):** the distribution is known only up to a normalizing constant (Chewi, 2023), and this kind of problem is fundamental in many fields such as Bayesian inference, randomized algorithms, and machine learning (Robert et al., 1999; Marin et al., 2007; Nakajima et al., 2019). **Problem (b):** known as the score-based generative models (SGMs) (Song & Ermon, 2019), we are given an approximation of $\nabla \log \pi_t$, where $\pi_t$ is the density of a specific process at time $t$. The law of this process converges to $\pi$ over time. SGMs are state-of-the-art in applications like image generation (Ho et al., 2022a; Dhariwal & Nichol, 2021), audio and video generation (Ho et al., 2022b; Yang et al., 2023a), and inverse problems (Song et al., 2022).

[1]The University of Tokyo, Tokyo, Japan [2]Center for Advanced Intelligence Project, RIKEN, Tokyo, Japan. Correspondence to: Huanjian Zhou <zhou@ms.k.u-tokyo.ac.jp>.

*Proceedings of the 42nd International Conference on Machine Learning*, Vancouver, Canada. PMLR 267, 2025. Copyright 2025 by the author(s).

For Problem (a), specifically log-concave sampling, starting from the seminal papers of Dalalyan & Tsybakov (2012), Dalalyan (2017), and Durmus & Moulines (2017), there has been a flurry of recent works on proving non-asymptotic guarantees based on simulating a process which converges to $\pi$ over time (Wibisono, 2018; Vempala & Wibisono, 2019; Mou et al., 2021; Altschuler & Talwar, 2023). Moreover, these processes, such as Langevin dynamics, converge exponentially quickly to $\pi$ under mild conditions (Dalalyan, 2017; Mou et al., 2021; Bernard et al., 2022). Such dynamics-based algorithms for Problem (a) share a common feature with the inference process of SGMs that they are actually a numerical simulation of an initial-value problem of differential equations (Hodgkinson et al., 2021). Thanks to the exponentially fast convergence of the process, significant efforts have been conducted on discretizing these processes using numerical methods such as the forward Euler, backward Euler (proximal method), exponential integrator, mid-point, and high-order Runge-Kutta methods (Vempala & Wibisono, 2019; Wibisono, 2019; Shen & Lee, 2019; Li et al., 2019; Oliva & Akyildiz, 2024).

Furthermore, in recent years, there have been increasing interest and significant advances in understanding the convergence of inherently dynamics-based SGMs (Bortoli, 2022; Chen et al., 2023c; Lee et al., 2023; Pedrotti et al., 2024; Chen et al., 2023b; Tang & Zhao, 2024; Li & Yan, 2024). Notably, polynomial-time convergence guarantees have been established (Chen et al., 2023c;b; Benton et al., 2024; Liang et al., 2025), and various discretization schemes for SGMs have been analyzed (Lu et al., 2022a;b; Huang et al., 2025).

The algorithms underlying the above results are highly sequential. However, with the increasing size of data sets for sampling, we need to develop a theory for algorithms with limited iterations. For example, the widely-used denoising diffusion probabilistic models (Ho et al., 2020) may take 1000 denoising steps to generate one sample, while the evaluations of a neural network-based score function can be computationally expensive (Song et al., 2021).

As a comparison, recently, the (naturally parallelizable) Picard methods for diffusion models reduced the number of steps to around 50 (Shih et al., 2024). Furthermore, in terms

of the dependency on the dimension $d$ and accuracy $\varepsilon$, Picard methods for both Problems (a) and (b) were proven to be able to return an $\varepsilon$-accurate solution within $\mathcal{O}(\log^2(d/\varepsilon^2))$ iterations, improved from previous $\mathcal{O}(d^a/\varepsilon^b)$ with some $a, b > 0$. However, the $\mathcal{O}(\log^2(d/\varepsilon^2))$ adaptive complexity[1] may not be yet optimal for both Problem (a) and (b). This motivates our investigation into the question:

*Can we achieve logarithmic adaptive complexity for both log-concave sampling and sampling for SGMs?*

**Our Contributions**

In this work, we propose a novel sampling method that employs a highly parallel discretization approach for continuous processes, with applications to the overdamped Langevin diffusion (Chewi, 2023) and the stochastic differential equation (SDE) implementation of processes in SGMs (Chen et al., 2024) for Problems (a) and (b), respectively.

**Faster parallel log-concave sampling.** We first present an improved result for parallel sampling from a strongly log-concave and log-smooth distribution. Specifically, we improve the upper bound from $\widetilde{\mathcal{O}}\left(\log^2\left(\frac{d}{\varepsilon^2}\right)\right)$ (Anari et al., 2024) to $\widetilde{\mathcal{O}}\left(\log\left(\frac{d}{\varepsilon^2}\right)\right)$, with slightly scaling the number of processors and gradient evaluations from $\mathcal{O}\left(\frac{d}{\varepsilon^2}\right)$ to $\mathcal{O}\left(\frac{d}{\varepsilon^2}\log\left(\frac{d}{\varepsilon^2}\right)\right)$.

Compared with methods based on underdamped Langevin diffusion (Shen & Lee, 2019; Yu & Dalalyan, 2024; Anari et al., 2024), our method exhibits higher space complexity[2]. This is primarily because underdamped Langevin diffusion typically follows a smoother trajectory than overdamped Langevin diffusion, allowing for larger grid spacing and consequently, a reduced number of grids. We summarize the comparison in Table 1. In this paper, we will focus on the adaptive complexity and discretization schemes for overdamped Langevin diffusion.

**Faster parallel sampling for diffusion models.** We then present an improved result for diffusion models. Specifically, we propose an efficient algorithm with $\widetilde{\mathcal{O}}\left(\log\left(\frac{d}{\varepsilon^2}\right)\right)$ adaptive complexity for SDE implementations of diffusion models (Song & Ermon, 2019). Our method surpasses all the existing parallel methods for diffusion models having $\widetilde{\mathcal{O}}\left(\log^2\left(\frac{d}{\varepsilon^2}\right)\right)$ adaptive complexity (Chen et al., 2024;

---

[1]Adaptive complexity refers to the minimal number of sequential rounds required for an algorithm to achieve a desired accuracy, assuming polynomially many queries can be executed in parallel at each round (Balkanski & Singer, 2018).

[2]We note, in this paper, that the space complexity refers to the number of words (Cohen-Addad et al., 2023; Chen et al., 2024) instead of the number of bits (Goldreich, 2008) to denote the approximate required storage.

*Table 1.* Comparison with existing parallel methods for strongly log-concave sampling. The symbol ♠ represents that the results hold under a weaker condition, the log-Sobolev inequality.

| Work dynamics | Measure | Adaptive Complexity | Space Complexity |
|---|---|---|---|
| (Shen & Lee, 2019, Theorem 4) underdamped Langevin diffusion | $W_2$ | $\widetilde{\mathcal{O}}\left(\log^2\left(\frac{\sqrt{d}}{\varepsilon}\right)\right)$ | $\widetilde{\mathcal{O}}\left(\frac{d^{3/2}}{\varepsilon}\right)$ |
| (Yu & Dalalyan, 2024, Corollary 2) underdamped Langevin diffusion | $W_2$ | $\widetilde{\mathcal{O}}\left(\log^2\left(\frac{d}{\varepsilon^2}\right)\right)$ | $\widetilde{\mathcal{O}}\left(\frac{d^{3/2}}{\varepsilon}\right)$ |
| (Anari et al., 2024, Theorem 15) underdamped Langevin diffusion | TV | $\widetilde{\mathcal{O}}\left(\log^2\left(\frac{d}{\varepsilon^2}\right)\right)$♠ | $\widetilde{\mathcal{O}}\left(\frac{d^{3/2}}{\varepsilon}\right)$ |
| (Anari et al., 2024, Theorem 13) overdamped Langevin diffusion | KL | $\widetilde{\mathcal{O}}\left(\log^2\left(\frac{d}{\varepsilon^2}\right)\right)$♠ | $\widetilde{\mathcal{O}}\left(\frac{d^2}{\varepsilon^2}\right)$ |
| Theorem 4.2 overdamped Langevin diffusion | KL | $\widetilde{\mathcal{O}}\left(\log\left(\frac{d}{\varepsilon^2}\right)\right)$ | $\widetilde{\mathcal{O}}\left(\frac{d^2}{\varepsilon^2}\right)$ |

*Table 2.* Comparison with existing parallel methods for sampling for diffusion models.

| Works Implementation | Measure | Adaptive Complexity | Space Complexity |
|---|---|---|---|
| (Chen et al., 2024, Theorem 3.5) ODE / Picard method | TV | $\widetilde{\mathcal{O}}\left(\log^2\left(\frac{d}{\varepsilon^2}\right)\right)$ | $\widetilde{\mathcal{O}}\left(\frac{d^{3/2}}{\varepsilon^2}\right)$ |
| (Gupta et al., 2025, Theorem B.13) ODE / Parallel midpoint method | TV | $\widetilde{\mathcal{O}}\left(\log^2\left(\frac{d}{\varepsilon^2}\right)\right)$ | $\widetilde{\mathcal{O}}\left(\frac{d^{3/2}}{\varepsilon^2}\right)$ |
| (Chen et al., 2024, Theorem 3.3) SDE / Picard method | KL | $\widetilde{\mathcal{O}}\left(\log^2\left(\frac{d}{\varepsilon^2}\right)\right)$ | $\widetilde{\mathcal{O}}\left(\frac{d^2}{\varepsilon^2}\right)$ |
| Theorem 5.4 SDE / Parallel Picard method | KL | $\widetilde{\mathcal{O}}\left(\log\left(\frac{d}{\varepsilon^2}\right)\right)$ | $\widetilde{\mathcal{O}}\left(\frac{d^2}{\varepsilon^2}\right)$ |

Gupta et al., 2025), with slightly increasing the number of the processors and gradient evaluations and the space complexity for SDEs. We summarize the comparison in Table 2. Similarly, the better space complexity of the ordinary differential equation (ODE) implementations is attributed to the smoother trajectories of ODEs, which are more readily discretized.

## 2. Problem Set-up

In this section, we introduce some preliminaries and key ingredients of log-concave sampling and diffusion models in Sections 2.1 and 2.2, respectively. Following this, Section 2.3 provides an introduction to the fundamentals of Picard iterations.

### 2.1. Log-concave Sampling

*Problem (a)* (**Sampling task**). Given the potential function $f : \mathcal{D} \to \mathbb{R}$, the goal of the sampling task is to draw a sample from the density $\pi_f = Z_f^{-1}\exp(-f)$, where $Z_f := \int_{\mathcal{D}}\exp(-f(\boldsymbol{x}))\mathrm{d}\boldsymbol{x}$ is the normalizing constant.

**Distribution and function class.** If $f$ is (strongly) convex, the density $\pi_f$ is said to be (strongly) *log-concave*. If $f$ is twice-differentiable and $\nabla^2 f \preceq \beta\boldsymbol{I}$ (where $\preceq$ denotes the Loewner order and $\boldsymbol{I}$ is the identity matrix), we say the potential $f$ is $\beta$-*smooth* and the density $\pi_f$ is $\beta$-*log-smooth*.

We define *relative Fisher information* of probability density $\rho$ w.r.t. $\pi$ as $\mathsf{FI}(\rho\|\pi) = \mathbb{E}_\rho[\|\nabla\log(\rho/\pi)\|^2]$ and the *Kullback–Leibler (KL) divergence* of $\rho$ from $\pi$ as $\mathsf{KL}(\rho\|\pi) = \mathbb{E}_\rho\log(\rho/\pi)$. If $\pi$ is $\alpha$-strongly log-concave, then the following relation between KL divergence and Fisher informa-

tion holds:

$$\mathsf{KL}(\rho\|\pi) \leq \frac{1}{2\alpha}\mathsf{FI}(\rho\|\pi) \text{ for all probability measures } \rho.$$

**Langevin Dynamics.** One of the most commonly-used dynamics for sampling is Langevin dynamics (Chewi, 2023), which is the solution to the following SDE,

$$\mathrm{d}\boldsymbol{x} = -\nabla f(\boldsymbol{x})\mathrm{d}t + \sqrt{2}\mathrm{d}\boldsymbol{B}_t,$$

where $(\boldsymbol{B}_t)_{t\in[0,T]}$ is a standard Brownian motion in $\mathbb{R}^d$. If $\pi \propto \exp(-f)$ satisfies strongly log-concavity, then the law of the Langevin diffusion converges exponentially fast to $\pi$ (Bakry et al., 2014).

**Score function for sampling task.** We assume the score function $\boldsymbol{s} : \mathbb{R}^d \to \mathbb{R}$ is a pointwise accurate estimate of $\nabla f$, i.e., $\|\boldsymbol{s}(\boldsymbol{x}) - \nabla f(\boldsymbol{x})\| \leq \delta$ for all $\boldsymbol{x} \in \mathbb{R}^d$ and some sufficiently small constant $\delta \in \mathbb{R}_+$.

**Measures of the output.** For two densities $\rho$ and $\pi$, we define the *total variation* (TV) as

$$\mathsf{TV}(\rho, \pi) = \sup\{\rho(E) - \pi(E) \mid E \text{ is an event}\}.$$

We have the following relation between the KL divergence and TV distance, known as the *Pinsker inequality*,

$$\mathsf{TV}(\rho, \pi) \leq \sqrt{\frac{1}{2}\mathsf{KL}(\rho\|\pi)}.$$

We denote by $\mathsf{W}_2$ the *Wasserstein distance* between $\rho$ and $\pi$, which is defined as

$$\mathsf{W}_2^2(\rho, \pi) = \inf_{\Pi}\mathbb{E}_{(X,Y)\sim\Pi}\left[\|X - Y\|^2\right],$$

where the infimum is over coupling distributions $\prod$ of $(X, Y)$ such that $X \sim \rho$, $Y \sim \pi$. If $\pi$ is $\alpha$-strongly log-concave, the following transport-entropy inequality, known as Talagrand's $\mathsf{T}_2$ inequality, holds (Otto & Villani, 2000) for all $\rho \in \mathcal{P}_2(\mathbb{R}^d)$, i.e., with finite second moment,

$$\frac{\alpha}{2}\mathsf{W}_2^2(\rho, \pi) \leq \mathsf{KL}(\rho\|\pi).$$

**Complexity.** For any sampling algorithm, we consider the *adaptive complexity* defined as unparallelizable evaluations of the score function (Chen et al., 2024), and use the notion of the *space complexity* to denote the approximate required storage during the inference. We note, in this paper, that the space complexity refers to the number of words (Cohen-Addad et al., 2023; Chen et al., 2024) instead of the number of bits (Goldreich, 2008) to denote the approximate required storage.

## 2.2. Score-based Diffusion Models

**Sampling for diffusion models.** In score-based diffusion models, one considers forward process $(\boldsymbol{x}_t)_{t\in[0,T]}$ in $\mathbb{R}^d$ governed by the canonical Ornstein-Uhlenbeck (OU) process (Ledoux, 2000):

$$\mathrm{d}\boldsymbol{x}_t = -\frac{1}{2}\boldsymbol{x}_t\mathrm{d}t + \mathrm{d}\boldsymbol{B}_t, \qquad \boldsymbol{x}_0 \sim \boldsymbol{q}_0, \qquad t \in [0, T], \quad (1)$$

where $\boldsymbol{q}_0$ is the initial distribution over $\mathbb{R}^d$. The corresponding backward process $(\bar{\boldsymbol{x}}_t)_{t\in[0,T]}$ in $\mathbb{R}^d$ follows an SDE defined as

$$\begin{cases} \mathrm{d}\bar{\boldsymbol{x}}_t = \left[\frac{1}{2}\bar{\boldsymbol{x}}_t + \nabla\log\bar{p}_t(\bar{\boldsymbol{x}}_t)\right]\mathrm{d}t + \mathrm{d}\boldsymbol{B}_t & t \in [0, T], \\ \bar{\boldsymbol{x}}_0 \sim p_0 \approx \mathcal{N}(\boldsymbol{0}_d, \boldsymbol{I}_d) \end{cases}$$

$$(2)$$

where $\mathcal{N}(\cdot, \cdot)$ represents the normal distribution over $\mathbb{R}^d$. In practice, the score function $\nabla\log\bar{p}_t(\bar{\boldsymbol{x}}_t)$ is estimated by neural network (NN) $\boldsymbol{s}_t^\theta : \mathbb{R}^d \mapsto \mathbb{R}^d$, where $\theta$ is the parameters of NN. The backward process is approximated by

$$\begin{cases} \mathrm{d}\boldsymbol{y}_t = \left[\frac{1}{2}\boldsymbol{y}_t + \boldsymbol{s}_t^\theta(\boldsymbol{y}_t)\right]\mathrm{d}t + \mathrm{d}\boldsymbol{B}_t & t \in [0, T], \\ \boldsymbol{y}_0 \sim \mathcal{N}(\boldsymbol{0}_d, \boldsymbol{I}_d). \end{cases} \quad (3)$$

*Problem (b)* (**Sampling task for SGMs**). Given the learned NN-based score function $\boldsymbol{s}_t^\theta$, the goal is to simulate the approximated backward process such that the law of the output is close to $\boldsymbol{q}_0$.

**Distribution class.** For SGMs, we assume the data density $p_0$ has finite second moments and is normalized such that $\mathrm{cov}_{p_0}(\boldsymbol{x}_0) = \mathbb{E}_{p_0}\left[(\boldsymbol{x}_0 - \mathbb{E}_{p_0}[\boldsymbol{x}_0])(\boldsymbol{x}_0 - \mathbb{E}_{p_0}[\boldsymbol{x}_0])^\top\right] = \boldsymbol{I}_d$. Such a finite moment assumption is standard across previous theoretical works on SGMs (Chen et al., 2023a;b;c) and we adopt the normalization to simplify true score function-related computations as Benton et al. (2024) and Chen et al. (2024) did.

**OU process and inverse process** The OU process and its inverse process also converge to the target distribution exponentially fast in various divergences and metrics such as the 2-Wasserstein metric $\mathsf{W}_2$; see Ledoux (2000). Furthermore, the discrepancy between the terminal distributions of the backward process (2) and its approximation version (3) scales polynomially with respect to the length of the time horizon and the score matching error. (Huang et al. (2025, Theorem 3.5) or setting the step size $h \to 0$ for the results in Chen et al. (2023a;b;c)).

**Score function for SGMs.** For the NN-based score, we assume the score function is $L^2$-accurate, bounded and Lipschitz; we defer the details in Section 5.2.

## 2.3. Picard Iterations

Consider the integral form of the initial value problem, $\boldsymbol{x}_t = \boldsymbol{x}_0 + \int_0^t f_s(\boldsymbol{x}_s)\mathrm{d}s + \sqrt{2}\boldsymbol{B}_t$. The main idea of Picard iterations (Clenshaw, 1957) is to approximate the difference over time slice $[t_n, t_{n+1}]$ as

$$
\begin{aligned}
&\boldsymbol{x}_{t_{n+1}} - \boldsymbol{x}_{t_n} \\
&= \int_{t_n}^{t_{n+1}} f_s(\boldsymbol{x}_s)\mathrm{d}s + \sqrt{2}(\boldsymbol{B}_{t_{n+1}} - \boldsymbol{B}_{t_n}) \\
&\approx \sum_{i=1}^{M} \boldsymbol{w}_i f_{t_n+\tau_{n,i}}(\boldsymbol{x}_{t_n+\tau_{n,i}})\mathrm{d}s + \sqrt{2}(\boldsymbol{B}_{t_{n+1}} - \boldsymbol{B}_{t_n}),
\end{aligned}
$$

with a discrete grids of $M$ collocation points as $\boldsymbol{x}_{t_n} = \boldsymbol{x}_{t_n+\tau_{n,0}} \leq \boldsymbol{x}_{t_n+\tau_{n,1}} \leq \boldsymbol{x}_{t_n+\tau_{n,2}} \leq \cdots \leq \boldsymbol{x}_{t_n+\tau_{n,M}} = \boldsymbol{x}_{t_{n+1}}$. We update the points in a wave-like fashion, which inherently allows for parallelization: for $m' = 1, \ldots, M$,

$$
\begin{aligned}
\boldsymbol{x}_{t_n+\tau_{n,m}}^{p+1} &= \boldsymbol{x}_0 + \sum_{m'=1}^{m-1} \boldsymbol{w}_{m'} f_{t_n+\tau_{n,m'}}(\boldsymbol{x}_{t_n+\tau_{n,m'}}^p) \\
&\quad + \sqrt{2}(\boldsymbol{B}_{t_n+\tau_{n,m}} - \boldsymbol{B}_{t_n}).
\end{aligned}
$$

Various collocation points have been proposed, including uniform points and Chebyshev points (Bai & Junkins, 2011). In this paper, however, we focus exclusively on the simplest case of uniform points, and extension to other cases is future work. Picard iterations are known to converge exponentially fast and, under certain conditions, even factorially fast for ODEs and backward SDEs (Hutzenthaler et al., 2021).

## 3. Technical Overview

We adopt the time splitting for the time horizon used in the existing parallel methods (Shen & Lee, 2019; Gupta et al., 2025; Chen et al., 2024; Anari et al., 2024; Yu & Dalalyan, 2024). With same time grids, our algorithm, however, depart crucially from prior work in the design of parallelism across the time slices, and the modification for controlling the score estimation error. Below we summarize these algorithmic contributions and technical novelties.

**Recap of existing parallel sampling methods.** Existing works for parallel sampling apply the following generic discretization schemes (Shen & Lee, 2019; Gupta et al., 2025; Chen et al., 2024; Anari et al., 2024; Yu & Dalalyan, 2024). At a high level, these methods divide the time horizon into many large time slices and each slice is further subdivided into grids with a small enough step size. Instead of sequentially updating the grid points, they update all grids at the same time slice simultaneously using exponentially fast converging Picard iterations (Alexander, 1990; Chen et al., 2024; Anari et al., 2024), or randomized midpoint methods (Shen & Lee, 2019; Yu & Dalalyan, 2024; Gupta

et al., 2025). With $\widetilde{\mathcal{O}}(\log d)$ Picard iterations for $\widetilde{\mathcal{O}}(\log d)$ time slices, the total adaptive complexity of their algorithms is $\widetilde{\mathcal{O}}(\log^2 d)$. However, while sequential updating of each time slice is not necessary for simulating the process, it remains unclear how to parallelize across time slices for sampling to obtain $\mathcal{O}(\log d)$ time complexity.

**Algorithmic novelty: parallel methods across time slices.** Naïvely, if we directly update all the grids simultaneously, the Picard iterations will not converge when the total length is $T = \widetilde{\mathcal{O}}(\log d)$. Instead of updating all time slices together or updating the time slice sequentially, we update the time slices in a *diagonal* style as illustrated in Figure 1. For any $j$-th update at the $n$-th time slice (corresponding the rectangle in the $n$-th column from the left and the $j$-th row from the top in Figure 1), there will be two inputs: (a) the right boundary point of the previous time slice, which has been updated $j$ times, and (b) the points on the girds of the same time slice that have been updated $j - 1$ times. Then we perform $P$ times Picard iterations with these inputs, where the hyperparameter $P$ depends on the smoothness of the score function. This repetition is simple but essential for preventing the accumulation of score-matching errors, which could otherwise grow exponentially w.r.t. the length of the time horizon. The main difference compared to the existing Picard methods is that for a fixed time slice, the starting points in our method are updated gradually, whereas in existing methods, the starting points remain fixed once processed.

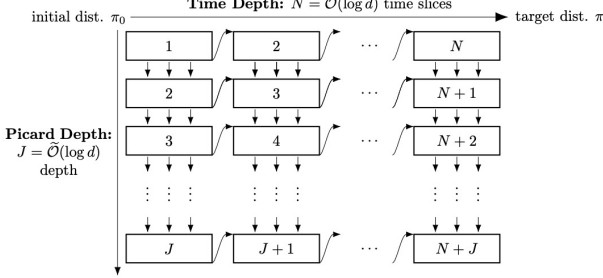

*Figure 1.* Illustration of the parallel Picard method: each rectangle represents an update, and the number within each rectangle indicates the index of the Picard iteration. The approximate time complexity is $N + J = \widetilde{\mathcal{O}}(\log d)$.

**Challenges for convergence.** Similar to the arguments for sequential methods or parallel methods with sequentially updating the time slices, we use the standard techniques such as the interpolation method or Girsanov's theorem (Vempala & Wibisono, 2019; Oksendal, 2013; Chewi, 2023) to decompose the total error w.r.t. KL into four components: (i) convergence error of the continuous process, (ii) discretization error, (iii) parallelization error, and (iv) score estimation error (See Eq. (4), Lemma B.1, and Lemma C.4). For (i) the convergence error of the continuous processes,

their exponential convergence rates allow this error to be effectively controlled by setting the total time length to $\mathcal{O}(\log \frac{d}{\varepsilon^2})$, regardless of the specific discretization scheme. For (ii), the discretization error scales approximately as $d \cdot \frac{h}{M}$, where $h$ denotes the time slice length, $M$ the number of discretization points per slice, and $\frac{h}{M}$ the grid resolution (See Eq. (4), and Lemma C.5). Setting $\frac{h}{M} \approx \mathcal{O}(\varepsilon^2/d)$ ensures the discretization error remains within $\mathcal{O}(\varepsilon^2)$. The technical challenges rise from controlling the remaining two errors, which we summarize below.

*(iii) Parallelization error:* the parallelization error primarily arises from updating with $s(x^{j-1})$ instead of $s(x^j)$ in the Picard iteration, in contrast to the sequential method, where $j$ indexes the steps along the Picard direction. In existing parallel methods, the sequential update across time slices benefits the convergence of truncation errors, $\mathbb{E}\left[\left\|x^j - x^{j-1}\right\|^2\right]$. Assuming the truncation errors in the previous time slice have converged, its right boundary serves as the starting point for all grids in the current $O(1)$-length time slice which results in an initial bias of $\mathcal{O}(d)$. Subsequently, by performing $\mathcal{O}(\log d)$ exponentially fast Picard iterations, the truncation error will converge. However, in our diagonal-style updating scheme across time, the truncation error interacts with inputs from both the previous time slice and prior updates in the same time slice. Consequently, the bias-convergence loop that holds in sequential updating no longer holds.

*(iv) Score estimation error:* If the score function itself is Lipschitz continuous (Assumption 5.3 for Problem (b)), no additional score matching error will arise during the Picard iterations. This allows the total score estimation error to remain bounded under mild conditions (Assumption 5.1). However, for Problem (a), since it is the velocity field $\nabla f$ instead of the score function $s$ that is Lipschitz, additional score estimation errors will occur during each update. For the sequential algorithm, these additional score estimation errors are contained within the bias-convergence loop, ensuring the total score estimation error remains to be bounded. Conversely, for our diagonal-style updating algorithm, the absence of convergence along the time direction causes these additional score estimation errors to accumulate exponentially over the time direction.

**Technical novelty.** Our technical contributions address these challenges by the appropriate selection of the number of Picard iterations within each update $P$ and the depth of the Picard iterations $J$. We outline the details of the choices below.

In the following, we assume that the truncation error at the $n$-th time slice and the $j$-th iteration scales with $L_n^j$, and that the additional score estimation error for each update scales with $\delta^2$.

To address the initial challenge related to the truncation error, we choose the Picard depth as $J = \mathcal{O}(N + \log d)$. We first bound the error of the output for each update with respect to its inputs as $L_n^j \leq \mathsf{a} L_{n-1}^j + \mathsf{b} L_n^{j-1}$, where $\mathsf{a}$ and $\mathsf{b}$ are constants. By carefully choosing the length of the time slices, we can ensure that $\mathsf{b} < 1$ along the Picard iteration direction. Consequently, the truncation error will converge if the iteration depth $J$ is sufficiently large, such that $\mathsf{a}^N \mathsf{b}^J$ is sufficiently small. This requirement implies that $J = \mathcal{O}(N + \log d)$.

To mitigate the additional score estimation error for Problem (a), we perform $P$ Picard iterations within each update. The interaction between the truncation error and additional score estimation error can be expressed as $L_n^j \leq \mathsf{a} L_{n-1}^j + \mathsf{b} L_n^{j-1} + \mathsf{c} \delta^2$, where $\mathsf{a}, \mathsf{b}, \mathsf{c}$ are constants. To ensure the total score estimation error remains bounded, it is necessary to have $\mathsf{a}, \mathsf{b} < 1$, which guarantees convergence along both the time and Picard directions. By the convergence of the Picard iteration, we can achieve $\mathsf{b} < 1$. For $\mathsf{a}$, the right boundary point of the previous time slice, and prior updates within the same time slice introduce discrepancies in the truncation error. For the impact from the previous time slice, we make use of the contraction of gradient decent to ensure convergence. However, since the grid gap scale as $1/d$, the contraction factor is close to $1$. Consequently, we have to minimize the impact from prior updates within the same time slice, which scales as $\mathcal{O}(1)$ by repeating $P = \log \mathcal{O}(1)$ Picard iterations for each update.

**Balance between time and Picard directions.** We note that the Picard method, despite being the simplest approach for time parallelism, has achieved the state of art performance in certain specific settings. On the one hand, the continuous processes need to run for at least $\mathcal{O}(\log d)$ time. To ensure convergence within every time slice, the time slice length have to be set as $\mathcal{O}(1)$, resulting in a necessity for at least $\mathcal{O}(\log d)$ iterations. On the other hand, with a proper initialization $\mathcal{O}(d)$, Picard iterations converge within $\mathcal{O}(\log d)$ iterations. Our parallelization balances the convergence of the continuous diffusion and the Picard iterations to achieve the improved results.

**Related works in scientific computation.** Similar parallelism across time slices has also been proposed in scientific computation (Gear, 1991; Gander, 2015; Ong & Schroder, 2020), especially for parallel Picard iterations (Wang, 2023). Compared with prior work in scientific computation, our approach exhibits several significant differences. Firstly, our primary objective differs from that in simulation. In sampling, we aim to ensure that the output distribution closely approximates the target distribution, whereas simulation seeks to make some points on the discrete grid closely match the true dynamics. Second, our algorithm differs sig-

nificantly from that of Wang (2023). In our algorithm, each update takes the inputs without the corrector operation. Furthermore, we perform $P$ Picard iterations in each update to prevent error accumulation over time $T = \widetilde{\mathcal{O}}(\log d)$. However, these two fields are connected through the sampling strategies that ensure each discrete point closely approximates the true process at every sampling step.

# 4. Parallel Picard Method for Strongly Log-concave Sampling

In this section, we present parallel Picard methods for strongly log-concave sampling (Algorithm 1) and show it holds improved convergence rate w.r.t. the KL divergence and total variance (Theorem 4.2 and Corollary 4.3). We illustrate the algorithm in Section 4.1, and give a proof sketch in Section 4.3. All the missing proofs can be found in Appendix B.

## 4.1. Algorithm

Our parallel Picard method for strongly log-concave sampling is summarized in Algorithm 1. In Lines 2–7, we generate the noises and initialize the value at the grid via Langevin Monte Carlo (Chewi, 2023) with a stepsize $h = \mathcal{O}(1)$. In Lines 8–26, the time slices are updated in a diagonal manner within the outer loop, as illustrated in Figure 1. In Lines 12–14 and Lines 21–23, we repeat $P$ Picard iterations for each update to ensure convergence.

*Remark* 4.1. Parallelization should be understood as evaluating the score function concurrently, with each time slice potentially being computed in an asynchronous parallel manner, resulting in the overall $P(N + J) + N$ adaptive complexity.

## 4.2. Theoretical Guarantees

The following theorem summarizes our theoretical analysis for Algorithm 1.

**Theorem 4.2.** *Suppose $\pi$ is $\alpha$-strongly log-concave and $\beta$-log-smooth, and the score function $\boldsymbol{s}$ is $\delta$-accurate. Let $\kappa = \beta/\alpha$. Suppose*

$$\beta h = 0.1, \qquad M \geq \frac{\kappa d}{\varepsilon^2}, \qquad N \geq 10\kappa \log\left(\frac{\mathsf{KL}(\mu_0\|\pi)}{\varepsilon^2}\right),$$

$$\delta \leq 0.2\sqrt{\alpha}\varepsilon, \qquad P \geq \frac{2\log\kappa}{3} + 4,$$

*and* $\quad J - N \geq \log\left(N^3\left(\frac{\kappa\delta^2 h + \kappa\mathsf{KL}(\mu_0\|\pi) + \kappa^2 d}{\varepsilon^2}\right)\right).$

*then Algorithm 1 runs within $N + (N + J)P$ iterations with at most $MN$ queries per iteration and outputs a sample with marginal distribution $\rho$ such that*

$$\max\left\{\frac{\sqrt{\alpha}}{2}\mathsf{W}_2(\rho, \pi), \mathsf{TV}(\rho, \pi)\right\} \leq \sqrt{\frac{\mathsf{KL}(\rho, \pi)}{2}} \leq 2\varepsilon.$$

---

**Algorithm 1** Parallel Picard Method for sampling

1: **Input:** $\boldsymbol{x}_0 \sim \mu_0$, approximate score function $\boldsymbol{s} \approx \nabla f$, the number of the iterations in outer loop $J$, the number of the iteration in inner loop $P$, the number of time slices $N$, the length of time slices $h$, the number of points on each time slices $M$.
2: **for** $n = 0, \ldots, N - 1$ **do**
3:     **for** $m = 0, \ldots, M$ (in parallel) **do**
4:       $B_{nh+m/Mh} = B_{nh} + \mathcal{N}(0, (mh/M)\boldsymbol{I}_d)$,
5:       $\boldsymbol{x}_{-1,M}^j = \boldsymbol{x}_0$, for $j = 0, \ldots, J$,
6:       $\boldsymbol{x}_{n,m}^0 = \boldsymbol{x}_{n-1,M}^0 - \frac{hm}{M}\boldsymbol{s}(\boldsymbol{x}_{n-1,M}^0) + \sqrt{2}(B_{nh+mh/M} - B_{nh})$,
7:     **end for**
8: **end for**
9: **for** $k = 1, \ldots, N$ **do**
10:     **for** $j = 1, \ldots, \min\{k - 1, J\}$ and $m = 1, \ldots, M$ (in parallel) **do**
11:       let $n = k - j$, $\boldsymbol{x}_{n,0}^j = \boldsymbol{x}_{n-1,M}^j$, and $\boldsymbol{x}_{n,m}^{j,0} = \boldsymbol{x}_{n,m}^{j-1}$,
12:       **for** $p = 1, \ldots, P$ **do**
13:         $\boldsymbol{x}_{n,m}^{j,p} = \boldsymbol{x}_{n,0}^j - \frac{h}{M}\sum_{m'=0}^{m-1}\boldsymbol{s}(\boldsymbol{x}_{n,m'}^{j,p-1}) + \sqrt{2}(B_{nh+mh/M} - B_{nh})$,
14:       **end for**
15:       $\boldsymbol{x}_{n,m}^j = \boldsymbol{x}_{n,m}^{j,P}$,
16:     **end for**
17: **end for**
18: **for** $k = N + 1, \ldots, N + J - 1$ **do**
19:     **for** $n = \max\{0, k - J\}, \ldots, N - 1$ and $m = 1, \ldots, M$ (in parallel) **do**
20:       let $j = k - n$, $\boldsymbol{x}_{n,0}^j = \boldsymbol{x}_{n-1,M}^j$, and $\boldsymbol{x}_{n,m}^{j,0} = \boldsymbol{x}_{n,m}^{j-1}$,
21:       **for** $p = 1, \ldots, P$ **do**
22:         $\boldsymbol{x}_{n,m}^{j,p} = \boldsymbol{x}_{n,0}^j - \frac{h}{M}\sum_{m'=0}^{m-1}\boldsymbol{s}(\boldsymbol{x}_{n,m'}^{j,p-1}) + \sqrt{2}(B_{nh+mh/M} - B_{nh})$,
23:       **end for**
24:       $\boldsymbol{x}_{n,m}^j = \boldsymbol{x}_{n,m}^{j,P}$,
25:     **end for**
26: **end for**
27: **Return:** $\boldsymbol{x}_{N-1,M}^J$.

---

To make the guarantee more explicit, we can combine it with the following well-known initialization bound, see, e.g., Dwivedi et al. (2019, Section 3.2).

**Corollary 4.3.** *Suppose that $\pi = \exp(-f)$ is $\alpha$-strongly log-concave and $\beta$-log-smooth, and let $\kappa = \beta/\alpha$. Let $\boldsymbol{x}^\star$ be the minimizer of $f$. Then, for $\mu_0 = \mathcal{N}(\boldsymbol{x}^\star, \beta^{-1})$, it holds that $\mathsf{KL}(\mu_0\|\pi) \leq \frac{d}{2}\log\kappa$. Consequently, setting*

$$h = \frac{1}{10\beta}, \quad M = \frac{\kappa d}{\varepsilon^2}, \quad N = 10\kappa\log\left(\frac{d\log\kappa}{\varepsilon^2}\right),$$

$$\delta \leq 0.2\sqrt{\alpha}\varepsilon, \quad P \geq \frac{2\log\kappa}{3} + 4,$$

$$and \quad J - N = \mathcal{O}\Big( \log \frac{\kappa^2 d \log \kappa}{\varepsilon^2} \Big),$$

*then Algorithm 1 runs within $N + (N + J)P = \widetilde{\mathcal{O}}(\kappa \log \frac{d}{\varepsilon^2})$ iterations with at most $MN = \widetilde{\mathcal{O}}(\frac{\kappa^2 d}{\varepsilon^2} \log \frac{d}{\varepsilon^2})$ queries per iteration and outputs a sample with marginal distribution $\rho$ such that*

$$\max \left\{ \frac{\sqrt{\alpha}}{2} \mathsf{W}_2(\rho, \pi), \mathsf{TV}(\rho, \pi) \right\} \leq \sqrt{\frac{\mathsf{KL}(\rho, \pi)}{2}} \leq 2\varepsilon.$$

*Remark* 4.4. The main drawback of our method is the sub-optimal space complexity due to its application to over-damped Langevin diffusion which has a less smooth trajectory compared to underdamped Langevin diffusion. However, we anticipate that our method could achieve comparable space complexity when adapted to underdamped Langevin diffusion.

*Remark* 4.5. Regarding the condition number $\kappa$, our method achieves the same adaptive complexity of $\mathcal{O}(\kappa)$ as both the state-of-the-art sequential method and existing parallel approaches (Anari et al., 2024; Yu & Dalalyan, 2024; Altschuler & Chewi, 2024). Whether parallelization can improve the dependence on the condition number remains an open question, which we leave for future work.

*Remark* 4.6. When the number of computation cores, denoted by $\ell$, is limited, the adaptive complexity of our algorithm is $\widetilde{\mathcal{O}}\big(\frac{\kappa^2 d}{\varepsilon^2 \ell} \log^2 \frac{d}{\varepsilon^2}\big)$. This matches the state-of-the-art adaptive complexity of Anari et al. (2024) and Yu & Dalalyan (2024) when $\ell \leq \frac{\kappa d}{\varepsilon^2}$. However, our algorithm achieves improved adaptive complexity when $\ell \geq \frac{\kappa d}{\varepsilon^2}$, with potential applications as demonstrated in Nishihara et al. (2014), De Souza et al. (2022), Hafych et al. (2022) and Glatt-Holtz et al. (2024).

### 4.3. Proof Sketch of Theorem 4.2: Performance Analysis of Algorithm 1

The detailed proof of Theorem 4.2 is deferred to Appendix B. As discussed in Section 3, by interpolation methods (Anari et al., 2024), we decompose the error w.r.t. the KL divergence into four error components (corollary B.4):

$$\mathsf{KL}(\rho\|\pi)$$
$$\lesssim \underbrace{e^{-\Theta(N)}\mathsf{KL}(\mu_0\|\pi)}_{Convergence\ of\ Langevin\ dynamics} + \underbrace{\frac{dh}{M}}_{discretization\ error}$$
$$+ \underbrace{\sum_{n=1}^{N-1} e^{-\Theta(n)}\mathcal{E}_{N-n}^J +}_{parallization\ error} \underbrace{\delta^2}_{score\ estimation\ error}, \quad (4)$$

where $\mathcal{E}_n^j$ represents the truncation error of the grids at $n$-th time slice after $j$ update. For the right terms, with the choice of $N = \mathcal{O}(\log \frac{d}{\varepsilon^2})$, $M = \mathcal{O}(dh/\varepsilon^2)$ and $\delta \leq \varepsilon$, which

ensures a sufficiently long time horizon the sufficiently long time horizon $T = Nh = \mathcal{O}(\log \frac{d}{\varepsilon^2})$, densely spaced grids with a gap $h/M = \mathcal{O}(\varepsilon^2/d)$ and a small score matching error, respectively, we can conclude that

$$e^{-\Theta(N)}\mathsf{KL}(\mu_0\|\pi) + \frac{dh}{M} + \delta^2 \lesssim \varepsilon^2$$

Thus, we will focus on proving the convergence of the truncation error $\mathcal{E}_n^j$ in the Picard iterations, and avoiding the additional accumulation of the score estimation error during Picard iterations as discussed before.

Considering that the truncation error expands at most exponentially along the time direction, but diminishes exponentially with an increased depth of the Picard iterations, convergence can be achieved by ensuring that the depth of the Picard iterations surpasses the number of time slices as $J \geq N + \mathcal{O}(\log \frac{d}{\varepsilon^2})$ with initialization error bounded by $\mathcal{O}(d)$ (the second part of Corollary B.7 and second part of Corollary B.9).

Due to the non-Lipschitzness of the score function, we can only bound $\mathcal{E}_n^j$ by quantity $\mathsf{a}\Delta_{n-1}^j + \mathsf{b}\mathcal{E}_n^{j-1} + \mathsf{c}\delta^2 h^2$ (Lemma B.5 and Lemma B.8), where $\Delta_{n-1}^j$ represents the truncation error at the right boundary of the previous time slice. Here, the coefficients are given by: $\mathsf{a} = 1 - 0.1\frac{\beta h}{\kappa} + \mathcal{O}(\kappa)(3\beta^2 h^2)^P$, $\mathsf{b} = \mathcal{O}((\beta^2 h^2)^P)$ and $\mathsf{c} = \mathcal{O}(\kappa\delta^2 h^2)$. Intuitively, $\mathsf{a}$ comes from the contraction of the gradient mapping with an additional term from the Picard direction, $\mathsf{b}$ reflects convergence along the Picard direction, and $\mathsf{c}$ accounts for the accumulation of score estimation error $\delta$ over time length $h$, with an additional scaling by $\kappa$ due to Young's inequality. To control the growth of the score error, it is essential that the coefficients $\mathsf{a}$ and $\mathsf{b}$ remain strictly less than one. Setting $P = \Theta(\log \kappa)$ is sufficient to ensure this condition.

## 5. Parallel Picard Method for Sampling of Diffusion Models

In this section, we present parallel Picard methods for diffusion models in Section 5.1 and assumptions in Section 5.2. Then we show it holds improved convergence rate w.r.t. the KL divergence (Theorem 5.4). All the missing details can be found in Appendix C.

### 5.1. Algorithm

Due to space limitations, the detailed methodology for the parallelization of Picard methods for diffusion models is provided in Appendix C.1 and Algorithm 2. It keeps same parallel structure as that illustrated in Figure 1. Notably, it exhibits the following distinctions in comparison to the parallel Picard methods for strongly log-concave sampling presented in Algorithm 1:

- Since the score function itself is Lipschitz, there will not be additional score matching error during Picard iterations. As a result, we perform single Picard iteration in one update, i.e., $P = 1$;

- Instead of uniform discrete grids, we employ a shrinking step size discretization scheme towards the data end, and the early stopping technique which is unvoidable to show the convergence for diffusion models (Chen et al., 2024). We show the details in Appendix C.1;

- We use an exponential integrator instead of the Euler-Maruyama Integrator in Picard iterations, where an additional high-order discretization error term would emerge (Chen et al., 2023a), which we believe would not affect the overall $\mathcal{O}(\log d)$ adaptive complexity with parallel sampling.

## 5.2. Assumptions

Our theoretical analysis of the algorithm assumes mild conditions regarding the data distribution's regularity and the approximation properties of NNs as discussed in Chen et al. (2024). These assumptions align with those established in previous theoretical works, such as those described by Chen et al. (2023c;a;b; 2024).

**Assumption 5.1** (($L^2([0, t_N])$ $\delta_2$-**accurate learned score**). The learned NN-based score $s_t^\theta$ is $\delta_2$-accurate in the sense of

$$\mathbb{E}_{\breve{p}}\left[\sum_{n=0}^{N-1}\sum_{m=0}^{M_n-1}\epsilon_{n,m}\big\|s_{t_n+\tau_{n,m}}^\theta(\breve{\boldsymbol{x}}_{t_n+\tau_{n,m}})\right.$$
$$\left.-\nabla\log\breve{p}_{t_n+\tau_{n,m}}(\breve{\boldsymbol{x}}_{t_n+\tau_{n,m}})\big\|^2\right]\leq\delta_2^2.$$

**Assumption 5.2** (**Regular and normalized data distribution**). The data density $p_0$ has finite second moments and is normalized such that $\mathrm{cov}_{p_0}(\boldsymbol{x}_0) = \boldsymbol{I}_d$.

**Assumption 5.3** (**Bounded and Lipschitz learned NN-based score**). The learned NN-based score function $s_t^\theta$ has a bounded $\mathcal{C}^1$ norm, i.e. , $\big\|\|s_t^\theta(\cdot)\|\big\|_{L^\infty([0,T])} \leq M_{\boldsymbol{s}}$ with Lipschitz constant $L_{\boldsymbol{s}}$.

## 5.3. Theoretical Guarantees

**Theorem 5.4.** *Under Assumptions 5.1, 5.2, and 5.3, given the following choices of the order of the parameters*

$$h = \Theta(1), \quad N = \mathcal{O}\Big(\log\frac{d}{\varepsilon^2}\Big), \quad M = \mathcal{O}\Big(\frac{d}{\varepsilon^2}\log\frac{d}{\varepsilon^2}\Big),$$

$$T = \mathcal{O}\Big(\log\frac{d}{\varepsilon^2}\Big), \quad \delta \leq \varepsilon, \quad \text{and} \quad J = \mathcal{O}\Big(N+\log\frac{Nd}{\varepsilon^2}\Big)$$

*the parallel Picard algorithm for diffusion models (Algorithm 2) generates samples from satisfies the following error bound,*

$$\mathsf{KL}(p_\eta\|\widetilde{q}_{t_N}) \lesssim de^{-T} + \frac{dT}{M} + \varepsilon^2 + \delta_2^2 \lesssim \varepsilon^2, \quad (5)$$

with total $2N + J = \widetilde{\mathcal{O}}\big(\log\frac{d}{\varepsilon^2}\big)$ *adaptive complexity and* $dM = \widetilde{\mathcal{O}}\big(\frac{d^2}{\varepsilon^2}\big)$ *space complexity for parallelizable $\delta_2$-accurate score function computations.*

*Remark* 5.5. Compared to existing parallel methods, our method improves the adaptive complexity from $\mathcal{O}(\log^2\frac{d}{\varepsilon^2})$ to $\mathcal{O}(\log\frac{d}{\varepsilon^2})$. Its main drawback is suboptimal space complexity due to the less smooth trajectory in SDE implementations, but we believe it can achieve comparable space complexity when adapted to ODE implementations.

*Remark* 5.6. When the number of computation cores $\ell$ is limited, the adaptive complexity is $\widetilde{\mathcal{O}}\big(\frac{\kappa^2d}{\varepsilon^2\ell}\log^2\frac{d}{\varepsilon^2}\big)$, matching the state-of-the-art result of Chen et al. (2024) for $\ell \leq \frac{\kappa d}{\varepsilon^2}$. However, employing a large batch size $\ell \geq \frac{\kappa d}{\varepsilon^2}$ in diffusion models may not always yield substantial benefits (Shih et al., 2024; Li et al., 2024b;a).

*Remark* 5.7. We note that the uniformly Lipschitz assumption (Assumption 5.3) may be too strong. In particular, the required Lipschitz constant can become quite large even becoming unbounded near the zero point (Salmona et al., 2022; Yang et al., 2023b). In this case, to ensure convergence of the Picard iterations under these conditions, the quantity $L_{\boldsymbol{s}}^2 e^{h_n} h_n$ must be sufficiently small. This requirement implies that the length of each time slice, $h_n$, should scale as $\mathcal{O}(1/L_{\boldsymbol{s}}^2)$. Consequently, the number of time slices becomes $N = \mathcal{O}(L_{\boldsymbol{s}}^2\log d)$, leading to an overall iteration complexity of $N = \mathcal{O}(L_{\boldsymbol{s}}^2\log d)$. We also believe our algorithm, which is based on a diagonal-style update, is robust to this assumption by adaptively adjusting the length of the time slices.

## 6. Discussion and Conclusion

In this work, we proposed novel parallel Picard methods for various sampling tasks. Notably, we obtain $\varepsilon^2$-accurate sample w.r.t. the KL divergence within $\widetilde{\mathcal{O}}\big(\log\frac{d}{\varepsilon^2}\big)$, which represents a significant improvement from $\widetilde{\mathcal{O}}\big(\log^2\frac{d}{\varepsilon^2}\big)$ for diffusion models. Furthermore compared with the existing methods applied to the overdamped Langevin dynamics or the SDE implementations for diffusion models, our space complexity only scales by a logarithmic factor.

Our study opens several promising theoretical directions. First, as an analogue to simulation methods in scientific computing, it highlights the potential of leveraging alternative discretization techniques for faster and more efficient sampling. Another direction is exploring smoother dynamics to reduce space complexity in these methods.

Lastly, although our highly parallel methods may introduce engineering challenges, such as the memory bandwidth, we believe our theoretical works will motivates the empirical development of parallel algorithms for both sampling and diffusion models.

## Acknowledgment

The authors thank Sinho Chewi for very helpful conversations. HZ was supported by International Graduate Program of Innovation for Intelligent World and Next Generation Artificial Intelligence Research Center. MS was supported by JST ASPIRE Grant Number JPMJAP2405.

## Impact Statement

This work focuses on the theory of accelerating the sampling via parallelism. As far as we can see, there is no foreseeable negative impact on the society.

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

# A. Useful tools

## A.1. Girsanov's Theorem

Following the notation introduced in Chen et al. (2024, Appendix A.2), we consider a probability space $(\Omega, \mathcal{F}, p)$ on which $(\boldsymbol{w}_t(\omega))_{t \geq 0}$ is a Wiener process in $\mathbb{R}^d$, with the filtration $\{\mathcal{F}_t\}_{t \geq 0}$. For an Itô process $\boldsymbol{z}_t(\omega)$ satisfies $\mathrm{d}\boldsymbol{z}_t(\omega) = \boldsymbol{\alpha}(t, \omega)\mathrm{d}t + \boldsymbol{\Sigma}(t, \omega)\mathrm{d}\boldsymbol{w}_t(\omega)$, we denote the marginal distribution of $\boldsymbol{z}_t$ by $p_t$, and the path measure of the process $z_t$ by $p_{t_1:t_2}$.

**Definition A.1.** Assume $\mathcal{B}$ is the Borel $\sigma$-algebra on $\mathbb{R}^d$. For any $0 \leq t_1 < t_2$, we define $\mathcal{V}$ as the class of functions $f(t, \omega) : [0, +\infty) \times \Omega \to \mathbb{R}$ which is $\mathcal{B} \times \mathcal{F}_t$ measurable, $\mathcal{F}_t$-adapted for any $t \geq 0$ and satisfies

$$\mathbb{E}\left[\exp\left(\int_0^t f^2(t, \omega)dt\right)\right] < +\infty, \quad \forall t > 0.$$

For vectors and matrices, we say it belongs to $\mathcal{V}^n$ or $\mathcal{V}^{m \times n}$ if each component of the vector or each entry of the matrix belongs to $\mathcal{V}$.

**Theorem A.2** ((Chen et al., 2024, Corollary A.4)). *Let $\boldsymbol{\alpha}(t, \omega) \in \mathcal{V}^m$, $\boldsymbol{\Sigma}(t, \omega) \in \mathcal{V}^{m \times n}$, and $(\boldsymbol{w}_t(\omega))_{t \geq 0}$ be a Wiener process on the probability space $(\Omega, \mathcal{F}, q)$. For $t \in [0, T]$, suppose $\boldsymbol{z}_t(\omega)$ is an Itô process with the following SDE:*

$$\mathrm{d}\boldsymbol{z}_t(\omega) = \boldsymbol{\alpha}(t, \omega)\mathrm{d}t + \boldsymbol{\Sigma}(t, \omega)\mathrm{d}\boldsymbol{w}_t(\omega), \tag{6}$$

*and there exist processes $\boldsymbol{\delta}(t, \omega) \in \mathcal{V}^n$ and $\boldsymbol{\beta}(t, \omega) \in \mathcal{V}^m$ such that:*

1. *$\boldsymbol{\Sigma}(t, \omega)\boldsymbol{\delta}(t, \omega) = \boldsymbol{\alpha}(t, \omega) - \boldsymbol{\beta}(t, \omega)$;*

2. *The process $M_t(\omega)$ as defined below is a martingale with respect to the filtration $\{\mathcal{F}_t\}_{t \geq 0}$ and probability measure $q$:*

$$M_t(\omega) = \exp\left(-\int_0^t \boldsymbol{\delta}(s, \omega)^\top \mathrm{d}\boldsymbol{w}_s(\omega) - \frac{1}{2}\int_0^t \|\boldsymbol{\delta}(s, \omega)\|^2 \mathrm{d}s\right),$$

*then there exists another probability measure $p$ on $(\Omega, \mathcal{F})$ such that:*

1. *$p \ll q$ with the Radon-Nikodym derivative $\frac{\mathrm{d}p}{\mathrm{d}q}(\omega) = M_T(\omega)$,*

2. *The process $\widetilde{\boldsymbol{w}}_t(\omega)$ as defined below is a Wiener process on $(\Omega, \mathcal{F}, p)$:*

$$\widetilde{\boldsymbol{w}}_t(\omega) = \boldsymbol{w}_t(\omega) + \int_0^t \boldsymbol{\delta}(s, \omega)\mathrm{d}s,$$

3. *Any continuous path in $\mathcal{C}([t_1, t_2], \mathbb{R}^m)$ generated by the process $\boldsymbol{z}_t$ satisfies the following SDE under the probability measure $p$:*

$$\mathrm{d}\widetilde{\boldsymbol{z}}_t(\omega) = \boldsymbol{\beta}(t, \omega)\mathrm{d}t + \boldsymbol{\Sigma}(t, \omega)\mathrm{d}\widetilde{\boldsymbol{w}}_t(\omega). \tag{7}$$

**Corollary A.3** ((Chen et al., 2024, Corollary A.5)). *Suppose the conditions in Theorem A.2 hold, then for any $t_1, t_2 \in [0, T]$ with $t_1 < t_2$, the path measure of the SDE (7) under the probability measure $p$ in the sense of $p_{t_1:t_2} = p(\boldsymbol{z}_{t_1:t_2}^{-1}(\cdot))$ is absolutely continuous with respect to the path measure of the SDE (6) in the sense of $q_{t_1:t_2} = q(\boldsymbol{z}_{t_1:t_2}^{-1}(\cdot))$. Moreover, the KL divergence between the two path measures is given by*

$$\mathsf{KL}(p_{t_1:t_2}\|q_{t_1:t_2}) = \mathsf{KL}(p_{t_1}\|q_{t_1}) + \mathbb{E}_{\omega \sim p|_{\mathcal{F}_{t_1}}}\left[\frac{1}{2}\int_{t_1}^{t_2} \|\boldsymbol{\delta}(t, \omega)\|^2 \mathrm{d}t\right].$$

## A.2. Comparison Inequalities

**Theorem A.4** (Gronwall inequality (Dragomir, 2003, Theorem 1)). *Let $x$, $\Psi$ and $\chi$ be real continuous functions defined in $[a, b]$, $\chi(t) \geq 0$ for $t \in [a, b]$. We suppose that on $[a, b]$ we have the inequality*

$$x(t) \leq \Psi(t) + \int_a^t \chi(s)x(s)\mathrm{d}s.$$

*Then*

$$x(t) \leq \Psi(t) + \int_a^t \chi(s)\Psi(s)\exp\left[\int_s^t \chi(u)\mathrm{d}u\right]\mathrm{d}s.$$

### A.3. Help Lemmas for diffusion models

**Lemma A.5 ((Chen et al., 2023a, Lemma 9)).** *For $\widehat{q}_0 \sim \mathcal{N}(0, I_d)$ and $\breve{p} = p_T$ is the distribution of the solution to the forward process ((2)), we have*

$$\mathsf{KL}(\breve{p}_0\|\widehat{q}_0) \lesssim de^{-T}.$$

## B. Missing Proof for Log-concave Sampling

We denotes $\mathsf{KL}_n^j = \mathsf{KL}(\mu_{n,M}^j\|\pi)$ where $\mu_{n,M}^j$ represents the law of $\boldsymbol{x}_{n,M}^j$. We define the truncation error from prior update as, $\mathcal{E}_n^j := \max\limits_{m=1,\ldots,M} \mathbb{E}\left[\left\|\boldsymbol{x}_{n,m}^{j,P} - \boldsymbol{x}_{n,m}^{j,P-1}\right\|^2\right]$, and truncation error at right boundary point as, $\Delta_n^j := \mathbb{E}\left[\left\|\boldsymbol{x}_{n,M}^j - \boldsymbol{x}_{n,M}^{j-1}\right\|^2\right]$.

### B.1. One Step Analysis of $\mathsf{KL}_n^j$: From $\mathsf{KL}$'s Convergence to Picard Convergence

In this section, we use the interpolation method to analyse the change of $\mathsf{KL}_n^j$ along time direction, which will be bounded by discretization error and score error.

**Lemma B.1.** *Assume $\beta h \leq 0.1$. For any $j = 1, \ldots, J$, $n = 1, \ldots, N - 1$, we have*

$$\mathsf{KL}_n^j \leq \exp(-1.2\alpha h)\mathsf{KL}_{n-1}^j + \frac{0.5\beta dh}{M} + 4.4\beta^2 h\mathcal{E}_n^j + 2.1\delta^2 h.$$

*Furthermore, for initialization part, i.e., $j = 0$, $n = 0, \ldots, N - 1$, we have*

$$\mathsf{KL}_n^0 \leq \exp\left(-\alpha(n+1)h\right)\mathsf{KL}(\mu_0\|\pi) + \frac{8\beta^2 dh}{\alpha},$$

*Remark* B.2. In the first equation, the term $\exp(-1.2\alpha h)\mathsf{KL}_{n-1}^j$ characterizes the convergence of the continuous diffusion. Additionally, the second and third terms quantify the discretization error. Adopting $P = 0$ and $M = 1$ reverts to the classical scenario, where the discretization error approximates $\mathcal{O}(hd)$, as discussed in Section 4.1 of Chewi (2023). Moreover, the second term is influenced by the density of the grids, while the third term is dependent on the convergence of the Picard iterations. The fourth term accounts for the score error.

*Proof.* We will use the interpolation method and follow the proof of Theorem 13 in Anari et al. (2024). For $j \in [J]$, $n = 0, \ldots, N - 1$ and $m = 0, \ldots, M - 1$, it is easy to see that

$$\boldsymbol{x}_{n,m+1}^j = \boldsymbol{x}_{n,m}^j - \frac{h}{M}\boldsymbol{s}(\boldsymbol{x}_{n,m}^{j,P-1}) + \sqrt{2}(B_{nh+(m+1)/h} - B_{nh+mh/M}).$$

Let $\boldsymbol{x}_t$ denote the linear interpolation between $\boldsymbol{x}_{n,m+1}^j$ and $\boldsymbol{x}_{n,m}^j$, i.e., for $t \in \left[nh + \frac{mh}{M}, nh + \frac{(m+1)hh}{M}\right]$, let

$$\boldsymbol{x}_t = \boldsymbol{x}_{n,m}^j - \left(t - nh - \frac{mh}{M}\right)\boldsymbol{s}(\boldsymbol{x}_{n,m}^{j,P-1}) + \sqrt{2}(B_t - B_{nh+mh/M}).$$

Note that $\boldsymbol{s}(\boldsymbol{x}_{n,m}^{j,P})$ is a constant vector field. Let $\mu_t$ be the law of $\boldsymbol{x}_t$. The same argument as in Vempala & Wibisono (2019, Lemma 3/Equation 32) yields the differential inequality

$$\partial_t\mathsf{KL}(\mu_t\|\pi) = -\mathsf{FI}(\mu_t\|\pi) + \mathbb{E}\left\langle \nabla f(\boldsymbol{x}_t) - \boldsymbol{s}(\boldsymbol{x}_{n,m}^{j,P-1}), \nabla \log \frac{\mu_t(\boldsymbol{x}_t)}{\pi(\boldsymbol{x}_t)}\right\rangle$$

$$\leq -\frac{3}{4}\mathsf{FI}(\mu_t\|\pi) + \mathbb{E}\left[\left\|\nabla f(\boldsymbol{x}_t) - \boldsymbol{s}(\boldsymbol{x}_{n,m}^{j,P-1})\right\|^2\right], \tag{8}$$

where we used $(a, b) \leq \frac{1}{4}\|a\|^2 + \|b\|^2$ and $\mathbb{E}\left[\left\|\nabla \log \frac{\mu_t(\boldsymbol{x}_t)}{\pi(\boldsymbol{x}_t)}\right\|^2\right] = \mathsf{FI}(\mu_t\|\pi)$. For the first term, by $\alpha$ strongly-log-concavity of $\pi$, we have $\mathsf{KL}(\mu_t\|\pi) \leq \frac{1}{2\alpha}\mathsf{FI}(\mu_t\|\pi)$. For the second term, we have

$$\mathbb{E}\left[\left\|\nabla f(\boldsymbol{x}_t) - \boldsymbol{s}(\boldsymbol{x}_{n,m}^{j,P-1})\right\|^2\right]$$

$$\leq 2\mathbb{E}\left[\left\|\nabla f(\boldsymbol{x}_t) - \nabla f(\boldsymbol{x}_{n,m}^{j,P-1})\right\|^2\right] + 2\mathbb{E}\left[\left\|\nabla f(\boldsymbol{x}_{n,m}^{j,P-1}) - \boldsymbol{s}(\boldsymbol{x}_{n,m}^{j,P-1})\right\|^2\right]$$

$$\leq 2\beta^2\mathbb{E}\left[\left\|\boldsymbol{x}_t - \boldsymbol{x}_{n,m}^{j,P-1}\right\|^2\right] + 2\delta^2. \tag{9}$$

Moreover,

$$\mathbb{E}\left[\left\|\boldsymbol{x}_t - \boldsymbol{x}_{n,m}^{j,P-1}\right\|^2\right] \leq 2\mathbb{E}\left[\left\|\boldsymbol{x}_t - \boldsymbol{x}_{n,m}^{j}\right\|^2\right] + 2\mathbb{E}\left[\left\|\boldsymbol{x}_{n,m}^{j,P} - \boldsymbol{x}_{n,m}^{j,P-1}\right\|^2\right] \tag{10}$$

For the first term, which will be influenced by density of grids, we have

$$\begin{aligned}
&\mathbb{E}\left[\left\|\boldsymbol{x}_t - \boldsymbol{x}_{n,m}^{j}\right\|^2\right] \\
&\leq \left(t - nh - \frac{mh}{M}\right)^2 \mathbb{E}\left[\left\|\boldsymbol{s}(\boldsymbol{x}_{n,m}^{j,P-1})\right\|^2\right] + d\left(t - nh - \frac{mh}{M}\right) \\
&\leq \frac{h^2}{M^2}\mathbb{E}\left[\left\|\boldsymbol{s}(\boldsymbol{x}_{n,m}^{j,P-1})\right\|^2\right] + d\left(t - nh - \frac{mh}{M}\right) \\
&\leq \frac{2h^2}{M^2}\mathbb{E}\left[\left\|\nabla f(\boldsymbol{x}_{n,m}^{j,P-1})\right\|^2\right] + \frac{2\delta^2 h^2}{M^2} + \frac{dh}{M} \\
&\leq \frac{4\beta^2 h^2}{M^2}\mathbb{E}\left[\left\|\boldsymbol{x}_t - \boldsymbol{x}_{n,m}^{j,P-1}\right\|^2\right] + \frac{4h^2}{M^2}\mathbb{E}\left[\left\|\nabla f(\boldsymbol{x}_t)\right\|^2\right] + \frac{2\delta^2 h^2}{M^2} + \frac{dh}{M}.
\end{aligned} \tag{11}$$

Taking $\beta h \leq \frac{1}{10}$, and combining (10) and (11), we have

$$\mathbb{E}\left[\left\|\boldsymbol{x}_t - \boldsymbol{x}_{n,m}^{j,P-1}\right\|^2\right] \leq \frac{4.4h^2}{M^2}\mathbb{E}\left[\left\|\nabla f(\boldsymbol{x}_t)\right\|^2\right] + \frac{2.2\delta^2 h^2}{M^2} + \frac{1.1dh}{M} + 2.2\mathbb{E}\left[\left\|\boldsymbol{x}_{n,m}^{j} - \boldsymbol{x}_{n,m}^{j,P-1}\right\|^2\right]. \tag{12}$$

For the first term, we recall the following lemma.

**Lemma B.3** ((Chewi et al., 2024, Lemma 16)).

$$\mathbb{E}\left[\left\|\nabla f(\boldsymbol{x}_t)\right\|^2\right] \leq \mathsf{FI}(\mu_t\|\pi) + 2\beta d.$$

Combining (8), (9), (12) and $\beta h \leq \frac{1}{10}$, we have for $j \in [J]$, $n = 0, \ldots, n-1$, $m = 0, \ldots, M-1$, and $t \in \left[nh + \frac{mh}{M}, nh + \frac{(m+1)hh}{M}\right]$,

$$\begin{aligned}
&\partial_t \mathsf{KL}(\mu_t\|\pi) \\
&\leq -\frac{3}{4}\mathsf{FI}(\mu_t\|\pi) + \mathbb{E}\left[\left\|\nabla f(\boldsymbol{x}_t) - \boldsymbol{s}(\boldsymbol{x}_{n,m}^{j,P-1})\right\|^2\right] \\
&\leq -\frac{3}{4}\mathsf{FI}(\mu_t\|\pi) + 2\beta^2\mathbb{E}\left[\left\|\boldsymbol{x}_t - \boldsymbol{x}_{n,m}^{j,P-1}\right\|^2\right] + 2\delta^2 \\
&\leq -\frac{3}{4}\mathsf{FI}(\mu_t\|\pi) + \frac{8.8\beta^2 h^2}{M^2}\mathbb{E}\left[\left\|\nabla f(\boldsymbol{x}_t)\right\|^2\right] + \frac{4.4\beta^2\delta^2 h^2}{M^2} + \frac{2.2\beta^2 dh}{M} + 4.4\beta^2\mathbb{E}\left[\left\|\boldsymbol{x}_{n,m}^{j,P} - \boldsymbol{x}_{n,m}^{j,P-1}\right\|^2\right] + 2\delta^2 \\
&\leq -\frac{3}{4}\mathsf{FI}(\mu_t\|\pi) + \frac{0.1}{M^2}\mathbb{E}\left[\left\|\nabla V(X_t)\right\|^2\right] + \frac{0.1\delta^2}{M^2} + \frac{2.2\beta^2 dh}{M} + 4.4\beta^2\mathcal{E}_n^j + 2\delta^2 \\
&\leq -\frac{3}{4}\mathsf{FI}(\mu_t\|\pi) + \frac{0.1}{M^2}\left(\mathsf{FI}(\mu_t\|\pi) + 2\beta d\right) + \frac{0.1\delta^2}{M^2} + \frac{2.2\beta^2 dh}{M} + 4.4\beta^2\mathcal{E}_n^j + 2\delta^2 \\
&\leq -1.2\alpha\mathsf{KL}(\mu_t\|\pi) + \frac{0.5\beta d}{M} + 4.4\beta^2\mathcal{E}_n^j + 2.1\delta^2
\end{aligned}$$

Since this inequality holds independently of $m$, we integral from $t = nh$ to $t = (n+1)h$,

$$\mathsf{KL}_n^j \leq \exp(-1.2\alpha h)\mathsf{KL}_{n-1}^j + \frac{0.5\beta dh}{M} + 4.4\beta^2 h\mathcal{E}_n^j + 2.1\delta^2 h.$$

As for $j = 0$, actually, Line 4-7 performs a Langevin Monte Carlo with step size $h$, by Theorem 4.2.6 in Chewi (2023), we have

$$\mathsf{KL}_n^0 \leq \exp\left(-\alpha nh\right)\mathsf{KL}_0^0 + \frac{8dh\beta^2}{\alpha},$$

with $0 < h \leq \frac{1}{4L}$.

$\square$

**Corollary B.4.** *Assume $\beta h \leq 0.1$. We have*

$$\mathsf{KL}_{N-1}^J \leq e^{-1.2\alpha(N-1)h}\left(\mathsf{KL}(\mu_0\|\pi) + 4.4\beta^2h\Delta_0^J\right) + \sum_{n=1}^{N-1}e^{-1.2\alpha(n-1)h}4.4\beta^2h\mathcal{E}_{N-n}^J + \frac{0.5\beta d}{\alpha M} + \frac{2.1\delta^2}{\alpha}.$$

*Furthermore, if $\mathcal{E}_{N-n}^J$ has a uniform bound as $\mathcal{E}_{N-n}^J \leq \mathcal{E} + 500\delta^2h^2$, we have*

$$\mathsf{KL}_{N-1}^J \leq e^{-1.2\alpha(N-1)h}\left(\mathsf{KL}(\mu_0\|\pi) + 4.4\beta^2h\Delta_0^J\right) + 5\beta\kappa\mathcal{E} + \frac{0.5\beta d}{\alpha M} + \frac{2.5\delta^2}{\alpha}.$$

*Proof.* By Lemma B.1, we decompose $\mathsf{KL}_{N-1}^J$ as

$$\begin{aligned}
\mathsf{KL}_{N-1}^J &\leq e^{-1.2\alpha(N-1)h}\mathsf{KL}_0^J + \sum_{n=1}^{N-1}e^{-1.2\alpha(n-1)h}\left(\frac{0.5\beta dh}{M} + 4.4\beta^2h\mathcal{E}_{N-n}^J + 2.1\delta^2h\right) \\
&\leq e^{-1.2\alpha(N-1)h}\left(\mathsf{KL}(\mu_0\|\pi) + 4.4\beta^2h\Delta_0^J\right) \\
&\quad + \frac{4.4\beta^2h(\mathcal{E} + 500\delta^2h^2) + \frac{0.5\beta dh}{M} + 2.1\delta^2h}{1 - \exp(-1.2\alpha h)} \\
&\leq e^{-1.2\alpha(N-1)h}\left(\mathsf{KL}(\mu_0\|\pi) + 4.4\beta^2h\Delta_0^J\right) + \frac{1.1}{\alpha h}4.4\beta^2h\mathcal{E} + \frac{1.1}{\alpha h}\frac{0.5\beta dh}{M} \\
&\quad + \frac{1.1}{\alpha h}25\delta^2h \\
&= e^{-1.2\alpha(N-1)h}\left(\mathsf{KL}(\mu_0\|\pi) + 4.4\beta^2h\Delta_0^J\right) + 5\kappa\beta\mathcal{E} + \frac{0.6\beta d}{\alpha M} + \frac{28\delta^2}{\alpha},
\end{aligned}$$

where the third inequality holds since $0 < x < 0.4$, we have $1.1 - 1.1\exp(-1.2x) - x > 0$. It is clear that $\alpha h < \beta h < 0.1$. $\qquad\square$

## B.2. One Step Analysis of $\Delta_n^j$

In this section, we analyze the one step change of $\Delta_n^j$ first.

**Lemma B.5.** *Assume $\beta h = \frac{1}{10}$ and $P \geq \frac{2\log\kappa}{3} + 4$. For any $j = 2, \ldots, J$, $n = 1, \ldots, N-1$, we have*

$$\Delta_n^j \leq \left(1 - \frac{0.005}{\kappa}\right)\Delta_{n-1}^j + 4.4\left(\frac{1}{M} + 10\kappa\right)h^2\delta^2 + 4.4\left(\frac{1}{M} + 10\kappa\right)\beta^2h^2\mathcal{E}_n^{j-1}.$$

*Furthermore, for $j = 1$, $n = 1, \ldots, N-1$, we have*

$$\Delta_n^1 \leq \Delta_{n-1}^1 + \left(\frac{1}{M} + 10\kappa\right)\left(5\delta^2h^2 + 6\beta^2dh^3 + 0.4\beta^2h^2\frac{\mathsf{KL}_{n-1}^0}{\alpha}\right).$$

*Proof.* **Decomposition when $j \geq 2$.** In fact, for $j \in [J]$, $n = 0, \ldots, N-1$, $m = 0, \ldots, M-1$, and $p = 1, \ldots, P$, it is easy to see that

$$\boldsymbol{x}_{n,m+1}^{j,p} = \boldsymbol{x}_{n,m}^{j,p} - \frac{h}{M}\boldsymbol{s}(\boldsymbol{x}_{n,m}^{j,p-1}) + \sqrt{2}(B_{nh+(m+1)/h} - B_{nh+mh/M}).$$

For any $j = 2, \ldots, J$, $n = 1, \ldots, N-1$, by the contraction of $\phi(\boldsymbol{x}) = \boldsymbol{x} - \frac{h}{M}\nabla f(\boldsymbol{x})$ (Lemma 2.2 in Altschuler & Talwar (2023)), for any $m = 1, \ldots, M$, we have,

$$\begin{aligned}
&\mathbb{E}\left[\left\|\boldsymbol{x}_{n,m}^{j,P} - \boldsymbol{x}_{n,m}^{j-1,P}\right\|^2\right] \\
&= \mathbb{E}\left[\left\|\boldsymbol{x}_{n,m-1}^{j,P} - \frac{h}{M}\boldsymbol{s}(\boldsymbol{x}_{n,m-1}^{j,P-1}) - \left(\boldsymbol{x}_{n,m-1}^{j-1,P} - \frac{h}{M}\boldsymbol{s}(\boldsymbol{x}_{n,m-1}^{j-1,P-1})\right)\right\|^2\right]
\end{aligned}$$

$$\leq (1+\eta)\mathbb{E}\left[\left\|\boldsymbol{x}_{n,m-1}^{j,P} - \frac{h}{M}\nabla f(\boldsymbol{x}_{n,m-1}^{j,P}) - \left(\boldsymbol{x}_{n,m-1}^{j-1,P} - \frac{h}{M}\nabla f(\boldsymbol{x}_{n,m-1}^{j-1,P})\right)\right\|^2\right]$$

$$+ \left(2+\frac{2}{\eta}\right)\mathbb{E}\left[\left\|\frac{h}{M}\nabla f(\boldsymbol{x}_{n,m-1}^{j,P}) - \frac{h}{M}\nabla f(\boldsymbol{x}_{n,m-1}^{j,P-1}) + \frac{h}{M}\nabla f(\boldsymbol{x}_{n,m-1}^{j-1,P}) - \frac{h}{M}\nabla f(\boldsymbol{x}_{n,m-1}^{j-1,P-1})\right\|^2\right]$$

$$+ \left(2+\frac{2}{\eta}\right)\mathbb{E}\left[\left\|\frac{h}{M}\nabla f(\boldsymbol{x}_{n,m-1}^{j,P-1}) - \frac{h}{M}\boldsymbol{s}(\boldsymbol{x}_{n,m-1}^{j,P-1}) + \frac{h}{M}\nabla f(\boldsymbol{x}_{n,m-1}^{j-1,P-1}) - \frac{h}{M}\boldsymbol{s}(\boldsymbol{x}_{n,m-1}^{j-1,P-1})\right\|^2\right]$$

$$\leq (1+\eta)\left(1-\frac{\alpha h}{M}\right)^2\mathbb{E}\left[\left\|\boldsymbol{x}_{n,m-1}^{j,P} - \boldsymbol{x}_{n,m-1}^{j-1,P}\right\|^2\right] + \left(4+\frac{4}{\eta}\right)\frac{h^2}{M^2}\delta^2$$

$$+ \left(4+\frac{4}{\eta}\right)\frac{\beta^2 h^2}{M^2}\mathbb{E}\left[\left\|\boldsymbol{x}_{n,m-1}^{j,P} - \boldsymbol{x}_{n,m-1}^{j,P-1}\right\|^2\right] + \left(4+\frac{4}{\eta}\right)\frac{\beta^2 h^2}{M^2}\mathbb{E}\left[\left\|\boldsymbol{x}_{n,m-1}^{j-1,P} - \boldsymbol{x}_{n,m-1}^{j-1,P-1}\right\|^2\right].$$

By setting $\eta = \frac{\alpha h}{M} = \frac{1}{10\kappa M}$, we have

$$\mathbb{E}\left[\left\|\boldsymbol{x}_{n,M}^{j,P} - \boldsymbol{x}_{n,M}^{j-1,P}\right\|^2\right]$$

$$\leq \left(1-\frac{\alpha h}{M}\right)^M\mathbb{E}\left[\left\|\boldsymbol{x}_{n,0}^{j,P} - \boldsymbol{x}_{n,0}^{j-1,P}\right\|^2\right] + \left(4+\frac{4}{\eta}\right)\frac{h^2}{M}\delta^2$$

$$+ \sum_{m=1}^{M}\left(4+\frac{4}{\eta}\right)\frac{\beta^2 h^2}{M^2}\mathbb{E}\left[\left\|\boldsymbol{x}_{n,m-1}^{j,P} - \boldsymbol{x}_{n,m-1}^{j,P-1}\right\|^2\right]$$

$$+ \sum_{m=1}^{M}\left(4+\frac{4}{\eta}\right)\frac{\beta^2 h^2}{M^2}\mathbb{E}\left[\left\|\boldsymbol{x}_{n,m-1}^{j-1,P} - \boldsymbol{x}_{n,m-1}^{j-1,P-1}\right\|^2\right]$$

$$\leq \exp(-\alpha h)\Delta_{n-1}^j + \left(4+\frac{4}{\eta}\right)\frac{h^2}{M}\delta^2 + \left(4+\frac{4}{\eta}\right)\frac{\beta^2 h^2}{M}\mathcal{E}_n^j + \left(4+\frac{4}{\eta}\right)\frac{\beta^2 h^2}{M}\mathcal{E}_n^{j-1}$$

$$\leq (1-0.1\alpha h)\Delta_{n-1}^j + \left(4+\frac{4}{\eta}\right)\frac{h^2}{M}\delta^2 + \left(4+\frac{4}{\eta}\right)\frac{\beta^2 h^2}{M}\mathcal{E}_n^j + \left(4+\frac{4}{\eta}\right)\frac{\beta^2 h^2}{M}\mathcal{E}_n^{j-1}$$

$$= \left(1-\frac{0.01}{\kappa}\right)\Delta_{n-1}^j + 4\left(\frac{1}{M} + 10\kappa\right)h^2\delta^2 + 4\left(\frac{1}{M} + 10\kappa\right)\beta^2 h^2\mathcal{E}_n^j$$

$$+ 4\left(\frac{1}{M} + 10\kappa\right)\beta^2 h^2\mathcal{E}_n^{j-1}. \tag{13}$$

In the following, we further decompose $\mathcal{E}_n^j$. For any $n = 0, \ldots, N-1$, $j \in [J]$, $p = 2, \ldots, P$, and $m = 1, \ldots, M$, we can decompose $\mathbb{E}\left[\left\|\boldsymbol{x}_{n,m}^{j,p} - \boldsymbol{x}_{n,m}^{j,p-1}\right\|^2\right]$ as follows. By definition (Line 12 or 18 in Algorithm 1), we have

$$\mathbb{E}\left[\left\|\boldsymbol{x}_{n,m}^{j,p} - \boldsymbol{x}_{n,m}^{j,p-1}\right\|^2\right]$$

$$= \frac{h^2}{M^2}\mathbb{E}\left[\left\|\sum_{m'=0}^{m-1}\boldsymbol{s}(\boldsymbol{x}_{n,m'}^{j,p-1}) - \sum_{m'=0}^{m-1}\boldsymbol{s}(\boldsymbol{x}_{n,m'}^{j,p-2})\right\|^2\right]$$

$$\leq \frac{h^2 m}{M^2}\sum_{m'=0}^{m-1}\mathbb{E}\left[\left\|\boldsymbol{s}(\boldsymbol{x}_{n,m'}^{j,p-1}) - \boldsymbol{s}(\boldsymbol{x}_{n,m'}^{j,p-2})\right\|^2\right]$$

$$\leq \frac{h^2 m}{M^2}\sum_{m'=0}^{m-1}3\left[\mathbb{E}\left[\left\|\nabla f(\boldsymbol{x}_{n,m'}^{j,p-1}) - \nabla f(\boldsymbol{x}_{n,m'}^{j,p-2})\right\|^2\right] + \mathbb{E}\left[\left\|\nabla f(\boldsymbol{x}_{n,m'}^{j,p-1}) - \boldsymbol{s}(\boldsymbol{x}_{n,m'}^{j,p-1})\right\|^2\right]\right.$$

$$\left. + \mathbb{E}\left[\left\|\nabla f(\boldsymbol{x}_{n,m'}^{j,p-2}) - \boldsymbol{s}(\boldsymbol{x}_{n,m'}^{j,p-2})\right\|^2\right]\right]$$

$$\leq 3\beta^2 h^2\max_{m'=1,\ldots,M}\mathbb{E}\left[\left\|\boldsymbol{x}_{n,m'}^{j,p-1} - \boldsymbol{x}_{n,m'}^{j,p-2}\right\|^2\right] + 6\delta^2 h^2. \tag{14}$$

Furthermore,

$$
\mathbb{E}\left[\left\|\boldsymbol{x}_{n,m-1}^{j,1} - \boldsymbol{x}_{n,m-1}^{j,0}\right\|^2\right]
$$

$$
= \mathbb{E}\left[\left\|\boldsymbol{x}_{n-1,M}^{j} - \frac{h}{M}\sum_{m'=0}^{m-1}\boldsymbol{s}(\boldsymbol{x}_{n,m'}^{j,0}) - \left(\boldsymbol{x}_{n-1,M}^{j-1} - \frac{h}{M}\sum_{m'=0}^{m-1}\boldsymbol{s}(\boldsymbol{x}_{n,m'}^{j-1,P-1})\right)\right\|^2\right]
$$

$$
\leq 2\mathbb{E}\left[\left\|\boldsymbol{x}_{n-1,M}^{j} - \boldsymbol{x}_{n-1,M}^{j-1}\right\|^2\right] + 2\frac{h^2 m}{M^2}\sum_{m'=0}^{m-1}\mathbb{E}\left[\left\|\boldsymbol{s}(\boldsymbol{x}_{n,m'}^{j-1,P}) - \boldsymbol{s}(\boldsymbol{x}_{n,m'}^{j-1,P-1})\right\|^2\right]
$$

$$
\leq 2\Delta_{n-1}^{j} + 6\beta^2 h^2 \mathcal{E}_n^{j-1} + 12\delta^2 h^2. \tag{15}
$$

Combining (14) and (15), we have

$$
\mathcal{E}_n^{j} = \mathbb{E}\left[\left\|\boldsymbol{x}_{n,m-1}^{j,P} - \boldsymbol{x}_{n,m-1}^{j,P-1}\right\|^2\right] \leq 2\cdot 0.03^{P-1}\Delta_{n-1}^{j} + 6\cdot 0.03^{P}\mathcal{E}_n^{j-1} + 6.6\delta^2 h^2. \tag{16}
$$

Substitute it into (13), we have for any $j = 2, \ldots, J$, $n = 1, \ldots, N-1$,

$$
\Delta_n^{j} \leq \left(1 - \frac{0.01}{\kappa} + 8\left(\frac{1}{M} + 10\kappa\right)0.03^{P}\right)\Delta_{n-1}^{j} + 4.4\left(\frac{1}{M} + 10\kappa\right)h^2\delta^2
$$

$$
+ 4.4\left(\frac{1}{M} + 10\kappa\right)\beta^2 h^2 \mathcal{E}_n^{j-1} \tag{17}
$$

$$
\leq \left(1 - \frac{0.005}{\kappa}\right)\Delta_{n-1}^{j} + 4.4\left(\frac{1}{M} + 10\kappa\right)h^2\delta^2 + 4.4\left(\frac{1}{M} + 10\kappa\right)\beta^2 h^2 \mathcal{E}_n^{j-1}, \tag{18}
$$

where the second inequality holds since $P \geq \frac{2\log\kappa}{3} + 4$ implies $8\left(\frac{1}{M} + 10\kappa\right)0.03^{P} \leq \frac{0.005}{\kappa}$.

**Decomposition when $j = 1$.** When $j = 1$, similarly, we have for $p = 1, \ldots, P$,

$$
\boldsymbol{x}_{n,m+1}^{1,p} = \boldsymbol{x}_{n,m}^{1,p} - \frac{h}{M}\boldsymbol{s}(\boldsymbol{x}_{n,m}^{1,p-1}) + \sqrt{2}(B_{nh+(m+1)/h} - B_{nh+mh/M}),
$$

and

$$
\boldsymbol{x}_{n,m+1}^{0} = \boldsymbol{x}_{n,m}^{0} - \frac{h}{M}\boldsymbol{s}(\boldsymbol{x}_{n-1,M}^{0}) + \sqrt{2}(B_{nh+(m+1)/h} - B_{nh+mh/M}).
$$

Thus by the contraction of $\phi(\boldsymbol{x}) = \boldsymbol{x} - \frac{h}{M}\nabla f(\boldsymbol{x})$ (Lemma 2.2 in Altschuler & Talwar (2023)), we have

$$
\mathbb{E}\left[\left\|\boldsymbol{x}_{n,m+1}^{1,P} - \boldsymbol{x}_{n,m+1}^{0}\right\|^2\right]
$$

$$
= \mathbb{E}\left[\left\|\boldsymbol{x}_{n,m}^{1,P} - \frac{h}{M}\boldsymbol{s}(\boldsymbol{x}_{n,m'}^{1,P-1}) - \left(\boldsymbol{x}_{n,m}^{0} - \frac{h}{M}\boldsymbol{s}(\boldsymbol{x}_{n-1,M}^{0})\right)\right\|^2\right]
$$

$$
\leq (1+\eta)\mathbb{E}\left[\left\|\boldsymbol{x}_{n,m}^{1,P} - \frac{h}{M}\nabla f(\boldsymbol{x}_{n,m}^{1,P}) - \left(\boldsymbol{x}_{n,m}^{0} - \frac{h}{M}\nabla f(\boldsymbol{x}_{n,m}^{0})\right)\right\|^2\right]
$$

$$
+ \left(2 + \frac{2}{\eta}\right)\mathbb{E}\left[\left\|\frac{h}{M}\nabla f(\boldsymbol{x}_{n,m}^{1,P}) - \frac{h}{M}\nabla f(\boldsymbol{x}_{n,m}^{1,P-1}) + \frac{h}{M}\nabla f(\boldsymbol{x}_{n,m}^{0}) - \frac{h}{M}\nabla f(\boldsymbol{x}_{n-1,M}^{0})\right\|^2\right]
$$

$$
+ \left(2 + \frac{2}{\eta}\right)\mathbb{E}\left[\left\|\frac{h}{M}\nabla f(\boldsymbol{x}_{n,m}^{1,P-1}) - \frac{h}{M}\boldsymbol{s}(\boldsymbol{x}_{n,m}^{1,P-1}) + \frac{h}{M}\nabla f(\boldsymbol{x}_{n-1,M}^{0}) - \frac{h}{M}\boldsymbol{s}(\boldsymbol{x}_{n-1,M}^{0})\right\|^2\right]
$$

$$
\leq (1+\eta)\left(1 - \frac{\alpha h}{M}\right)^2 \mathbb{E}\left[\left\|\boldsymbol{x}_{n,m}^{1,P} - \boldsymbol{x}_{n,m}^{0}\right\|^2\right] + \left(4 + \frac{4}{\eta}\right)\frac{\delta^2 h^2}{M^2}
$$

$$
+ \left(4 + \frac{4}{\eta}\right)\frac{\beta^2 h^2}{M^2}\mathbb{E}\left[\left\|\boldsymbol{x}_{n,m}^{1,P} - \boldsymbol{x}_{n,m}^{1,P-1}\right\|^2\right] + \left(4 + \frac{4}{\eta}\right)\frac{\beta^2 h^2}{M^2}\mathbb{E}\left[\left\|\boldsymbol{x}_{n,m}^{0} - \boldsymbol{x}_{n-1,M}^{0}\right\|^2\right].
$$

For third term $\mathbb{E}\left[\left\|\boldsymbol{x}_{n,m}^{1,P} - \boldsymbol{x}_{n,m}^{1,P-1}\right\|^2\right]$, we have

$$\mathbb{E}\left[\left\|\boldsymbol{x}_{n,m}^{1,P} - \boldsymbol{x}_{n,m}^{1,P-1}\right\|^2\right]$$

$$= \mathbb{E}\left[\left\|\frac{h}{M}\sum_{m'=0}^{m} \boldsymbol{s}(\boldsymbol{x}_{n,m'}^{1,P-1}) - \boldsymbol{s}(\boldsymbol{x}_{n,m'}^{1,P-2})\right\|^2\right]$$

$$\leq \frac{mh^2}{M^2}\sum_{m'=0}^{m}\mathbb{E}\left[\left\|\boldsymbol{s}(\boldsymbol{x}_{n,m'}^{1,P-1}) - \boldsymbol{s}(\boldsymbol{x}_{n,m'}^{1,P-2})\right\|^2\right]$$

$$\leq 3\beta^2 h^2 \max_{m'=0,\ldots,M}\mathbb{E}\left[\left\|\boldsymbol{x}_{n,m'}^{1,P-1} - \boldsymbol{x}_{n,m'}^{1,P-2}\right\|^2\right] + 6\delta^2 h^2.$$

Thus

$$\mathbb{E}\left[\left\|\boldsymbol{x}_{n,m}^{1,P} - \boldsymbol{x}_{n,m}^{1,P-1}\right\|^2\right] \leq 0.03^{P-1} \max_{m'=0,\ldots,M}\mathbb{E}\left[\left\|\boldsymbol{x}_{n,m'}^{1,1} - \boldsymbol{x}_{n,m'}^{1,0}\right\|^2\right] + 6.2\delta^2 h^2. \tag{19}$$

For $\mathbb{E}\left[\left\|\boldsymbol{x}_{n,m}^{1,1} - \boldsymbol{x}_{n,m}^{1,0}\right\|^2\right]$, by definition, we have

$$\mathbb{E}\left[\left\|\boldsymbol{x}_{n,m}^{1,1} - \boldsymbol{x}_{n,m}^{1,0}\right\|^2\right]$$

$$= \mathbb{E}\left[\left\|\boldsymbol{x}_{n-1,M}^{1} - \frac{h}{M}\sum_{m'=0}^{m-1}\boldsymbol{s}(\boldsymbol{x}_{n,m'}^{0}) - \left(\boldsymbol{x}_{n-1,M}^{0} - \frac{h}{M}\sum_{m'=0}^{m-1}\boldsymbol{s}(\boldsymbol{x}_{n-1,M}^{0})\right)\right\|^2\right]$$

$$\leq 2\mathbb{E}\left[\left\|\boldsymbol{x}_{n-1,M}^{1} - \boldsymbol{x}_{n-1,M}^{0}\right\|^2\right] + 2\mathbb{E}\left[\left\|\frac{h}{M}\sum_{m'=0}^{m-1}\boldsymbol{s}(\boldsymbol{x}_{n,m'}^{0}) - \frac{h}{M}\sum_{m'=0}^{m-1}\boldsymbol{s}(\boldsymbol{x}_{n-1,M}^{0})\right\|^2\right]$$

$$\leq 2\mathbb{E}\left[\left\|\boldsymbol{x}_{n-1,M}^{1} - \boldsymbol{x}_{n-1,M}^{0}\right\|^2\right] + 2\frac{h^2 m}{M^2}\sum_{m'=0}^{m-1}\mathbb{E}\left[\left\|\boldsymbol{s}(\boldsymbol{x}_{n,m'}^{0}) - \boldsymbol{s}(\boldsymbol{x}_{n-1,M}^{0})\right\|^2\right]$$

$$\leq 2\mathbb{E}\left[\left\|\boldsymbol{x}_{n-1,M}^{1} - \boldsymbol{x}_{n-1,M}^{0}\right\|^2\right] + 6\beta^2 h^2 \max_{m'\in[M]}\mathbb{E}\left[\left\|\boldsymbol{x}_{n,m'}^{0} - \boldsymbol{x}_{n-1,M}^{0}\right\|^2\right] + 12\delta^2 h^2. \tag{20}$$

For $\mathbb{E}\left[\left\|\boldsymbol{x}_{n,m}^{0} - \boldsymbol{x}_{n-1,M}^{0}\right\|^2\right]$, by definition of $\boldsymbol{x}_{n,m}^{0}$ (Line 7 in Algorithm 1), we have

$$\mathbb{E}\left[\left\|\boldsymbol{x}_{n,m}^{0} - \boldsymbol{x}_{n-1,M}^{0}\right\|^2\right]$$

$$= \frac{h^2 m^2}{M^2}\mathbb{E}\left[\left\|\boldsymbol{s}(\boldsymbol{x}_{n-1,M}^{0})\right\|^2\right] + \frac{dhm}{M}$$

$$\leq 2\delta^2 h^2 + 2h^2\mathbb{E}\left[\left\|\nabla f(\boldsymbol{x}_{n-1,M}^{0})\right\|^2\right] + dh$$

$$\leq 2\delta^2 h^2 + 2h^2\left(2\beta d + \frac{4\beta^2}{\alpha}\mathsf{KL}(\mu_{n-1,M}^{0}\|\pi)\right) + dh$$

$$= 4h^2\beta d + 2h^2\delta^2 + \frac{8\beta^2 h^2}{\alpha}\mathsf{KL}_{n-1}^{0} + dh, \tag{21}$$

where the last inequality is implied from the following lemma, (Vempala & Wibisono, 2019, Lemma 10)

$$\mathbb{E}\left[\left\|\nabla f(\boldsymbol{x}_{n-1,M}^{0})\right\|^2\right] \leq 2\beta d + \frac{4\beta^2}{\alpha}\mathsf{KL}(\mu_{n-1,M}^{0}\|\pi).$$

Combining (19), (20), and (21), and $P \geq 4$, we have

$$\mathbb{E}\left[\left\|\boldsymbol{x}_{n,m}^{1,P} - \boldsymbol{x}_{n,m}^{1,P-1}\right\|^2\right]$$

$$\leq 0.03^{P-1} \max_{m'=0,\ldots,M} \mathbb{E}\left[\left\|\boldsymbol{x}_{n,m'}^{1,1} - \boldsymbol{x}_{n,m'}^{1,0}\right\|^2\right] + 6.2h^2\delta^2$$

$$\leq 0.03^{P-1}\left[2\Delta_{n-1}^1 + 6\beta^2h^2\left(4h^2\beta d + 2h^2\delta^2 + \frac{8\beta^2h^2}{\alpha}\mathsf{KL}_{n-1}^0 + dh\right) + 12\delta^2h^2\right] + 6.2h^2\delta^2$$

$$\leq 2\cdot 0.03^{P-1}\Delta_{n-1}^1 + 6.3h^2\delta^2 + 0.01dh + 0.01\frac{\beta^2h^2}{\alpha}\mathsf{KL}_{n-1}^0. \tag{22}$$

By setting $\eta = \frac{\alpha h}{M} = \frac{1}{10\kappa M}$, we have

$$\mathbb{E}\left[\left\|\boldsymbol{x}_{n,M}^{1,P} - \boldsymbol{x}_{n,M}^0\right\|^2\right]$$

$$\leq \left(1 - \frac{\alpha h}{M}\right)^M \mathbb{E}\left[\left\|\boldsymbol{x}_{n,0}^{1,P} - \boldsymbol{x}_{n,0}^0\right\|^2\right] + \left(4 + \frac{4}{\eta}\right)\frac{\delta^2h^2}{M}$$

$$+ \left(4 + \frac{4}{\eta}\right)\frac{\beta^2h^2}{M}\left(2\cdot 0.03^{P-1}\Delta_{n-1}^1 + 6.3h^2\delta^2 + 0.01dh + 0.01\frac{\beta^2h^2}{\alpha}\mathsf{KL}_{n-1}^0\right)$$

$$+ \left(4 + \frac{4}{\eta}\right)\frac{\beta^2h^2}{M}\left(4h^2\beta d + 2h^2\delta^2 + \frac{8\beta^2h^2}{\alpha}\mathsf{KL}_{n-1}^0 + dh\right)$$

$$\leq \left(1 - \frac{0.01}{\kappa} + 4\left(\frac{1}{M} + 10\kappa\right)0.03^P\right)\Delta_{n-1}^1 + \left(\frac{1}{M} + 10\kappa\right)\left(5\delta^2h^2 + 6\beta^2dh^3 + 0.4\beta^2h^2\frac{\mathsf{KL}_{n-1}^0}{\alpha}\right)$$

$$\leq \Delta_{n-1}^1 + \left(\frac{1}{M} + 10\kappa\right)\left(5\delta^2h^2 + 6\beta^2dh^3 + 0.4\beta^2h^2\frac{\mathsf{KL}_{n-1}^0}{\alpha}\right)$$

where the last inequality holds since $P \geq \frac{2\log\kappa}{3} + 4$ implies $8\left(\frac{1}{M} + 10\kappa\right)0.03^P \leq \frac{0.005}{\kappa}$. $\qquad\square$

When $n = 0$, the update is identical to the Picard iteration shown in Anari et al. (2024), thus we have the following lemma.

**Lemma B.6** (Lemma 18 in Anari et al. (2024)). *For $j = 1, \ldots, J$, we have*

$$\Delta_0^j \leq 0.03^P\Delta_0^{j-1} + 6.2\delta^2h^2,$$

*with $\Delta_0^0 := \max_{m=0,\ldots,M}\mathbb{E}\left[\left\|\boldsymbol{x}_{0,m}^0 - \boldsymbol{x}_0\right\|^2\right] \leq \frac{4\beta^2h^2}{\alpha}\mathsf{KL}(\mu_0\|\pi) + 1.4dh + 2\delta^2h^2$.*

**Corollary B.7.** *For $n = 1, \ldots, N-1$, we have*

$$\Delta_n^1 \leq n\left(\frac{1}{M} + 10\kappa\right)\left(5.1\delta^2h^2 + 0.5\frac{\beta^2h^2}{\alpha}\mathsf{KL}(\mu_0\|\pi) + 10\kappa^2\beta^2dh^3\right).$$

*Furthermore, for $j = 1, \ldots, J$ and $n = 0$, we have*

$$\Delta_0^j \leq 0.03^{jP}\frac{4\beta^2h^2}{\alpha}\mathsf{KL}(\mu_0\|\pi) + 1.4\cdot 0.03^{jP}dh + 6.7\delta^2h^2.$$

*Proof.* By Lemma B.6, we have

$$\Delta_0^j \leq 0.03^P\Delta_0^{j-1} + 6.2\delta^2h^2$$

$$\leq 0.03^{jP}\Delta_0^0 + 6.6\delta^2h^2$$

$$\leq 0.03^{jP}\left(\frac{4\beta^2h^2}{\alpha}\mathsf{KL}(\mu_0\|\pi) + 1.4dh + 2\delta^2h^2\right) + 6.6\delta^2h^2$$

$$\leq 0.03^{jP}\frac{4\beta^2h^2}{\alpha}\mathsf{KL}(\mu_0\|\pi) + 1.4\cdot 0.03^{jP}dh + 6.7\delta^2h^2.$$

Combining Lemma B.1 and Lemma B.5, we have

$$
\begin{aligned}
\Delta_n^1 &\le \Delta_0^1 + \sum_{i=1}^n \left( \frac{1}{M} + 10\kappa \right) \left( 5\delta^2 h^2 + 6\beta^2 dh^3 + 0.4\beta^2 h^2 \frac{\mathsf{KL}_{i-1}^0}{\alpha} \right) \\
&\le \Delta_0^1 + n \left( \frac{1}{M} + 10\kappa \right) \left( 5\delta^2 h^2 + 6\beta^2 dh^3 \right) \\
&\quad + \sum_{i=1}^n \left( \frac{1}{M} + 10\kappa \right) 0.4 \frac{\beta^2 h^2}{\alpha} \left( \exp\left(-\alpha n h\right) \mathsf{KL}(\mu_0\|\pi) + \frac{8\beta^2 dh}{\alpha} \right) \\
&\le \Delta_0^1 + n \left( \frac{1}{M} + 10\kappa \right) \left( 5\delta^2 h^2 + 6\beta^2 dh^3 + 0.4 \frac{\beta^2 h^2}{\alpha} \mathsf{KL}(\mu_0\|\pi) + 3.2\kappa^2 \beta^2 dh^3 \right) \\
&\le n \left( \frac{1}{M} + 10\kappa \right) \left( 5.1\delta^2 h^2 + 0.5 \frac{\beta^2 h^2}{\alpha} \mathsf{KL}(\mu_0\|\pi) + 10\kappa^2 \beta^2 dh^3 \right).
\end{aligned}
$$

$\square$

## B.3. One Step Analysis of $\mathcal{E}_n^j$

In this section, we analyze the one step change of $\mathcal{E}_n^j$.

**Lemma B.8.** *For any $j = 2, \ldots, J$, $n = 1, \ldots, N-1$, we have*

$$
\mathcal{E}_n^j \le 2 \cdot 0.03^{P-1} \Delta_{n-1}^j + 2 \cdot 0.03^P \mathcal{E}_n^{j-1} + 7\delta^2 h^2.
$$

*Furthermore, for $n = 1, \ldots, N-1$, we have*

$$
\mathcal{E}_n^1 \le 2 \cdot 0.03^{P-1} \Delta_{n-1}^1 + 6.3 h^2 \delta^2 + 0.01 dh + 0.01 \frac{\beta^2 h^2}{\alpha} \mathsf{KL}_{n-1}^0.
$$

*Proof.* By (16), the first inequality holds. By (22), the second inequality holds. $\square$

**Corollary B.9.** *For $n = 1, \ldots, N-1$, we have*

$$
\mathcal{E}_n^1 \le n \left( 5.5\delta^2 h^2 + 0.1 \frac{\beta^2 h^2}{\alpha} \mathsf{KL}(\mu_0\|\pi) + 0.1\kappa^2 dh \right).
$$

*Proof.* Combining Lemma B.1, Lemma B.8 and Corollary B.7, we have

$$
\begin{aligned}
\mathcal{E}_n^1 &\le 2 \cdot 0.03^{P-1} \Delta_{n-1}^1 + 6.3 h^2 \delta^2 + 0.01 dh + 0.01 \frac{\beta^2 h^2}{\alpha} \mathsf{KL}_{n-1}^0 \\
&\le 2 \cdot 0.03^{P-1} \Delta_{n-1}^1 + 6.3 h^2 \delta^2 + 0.01 dh \\
&\quad + 0.01 \frac{\beta^2 h^2}{\alpha} \left( \exp\left(-\alpha(n+1)h\right) \mathsf{KL}(\mu_0\|\pi) + \frac{8\beta^2 dh}{\alpha} \right) \\
&\le 2 \cdot 0.03^{P-1} \Delta_{n-1}^1 + 6.3 h^2 \delta^2 + 0.02\kappa dh + 0.01 \frac{\beta^2 h^2}{\alpha} \mathsf{KL}(\mu_0\|\pi) \\
&\le 2 \cdot 0.03^{P-1} \left( n \left( \frac{1}{M} + 10\kappa \right) \left( 5.1\delta^2 h^2 + 0.5 \frac{\beta^2 h^2}{\alpha} \mathsf{KL}(\mu_0\|\pi) + 10\kappa^2 \beta^2 dh^3 \right) \right) \\
&\quad + 6.3 h^2 \delta^2 + 0.02\kappa dh + 0.01 \frac{\beta^2 h^2}{\alpha} \mathsf{KL}(\mu_0\|\pi) \\
&\le n \cdot 0.06 \left( 5.1\delta^2 h^2 + 0.5 \frac{\beta^2 h^2}{\alpha} \mathsf{KL}(\mu_0\|\pi) + 0.1\kappa^2 dh \right) \\
&\quad + 6.3 h^2 \delta^2 + 0.02\kappa dh + 0.01 \frac{\beta^2 h^2}{\alpha} \mathsf{KL}(\mu_0\|\pi) \\
&\le n \left( 5.5\delta^2 h^2 + 0.1 \frac{\beta^2 h^2}{\alpha} \mathsf{KL}(\mu_0\|\pi) + 0.1\kappa^2 dh \right).
\end{aligned}
$$

where the fifth inequality holds since $P \ge \frac{2\log\kappa}{3} + 4$ implies $\left( \frac{1}{M} + 10\kappa \right) 0.03^{P-1} \le 0.03$. $\square$

### B.4. Proof of Theorem 4.2

We define an energy function as

$$L_n^j = \Delta_{n-1}^j + \kappa \mathcal{E}_n^{j-1}.$$

We note that $2 \cdot 0.03^{P-1} L_n^j + 7\delta^2 h^2 \geq \mathcal{E}_n^j$. By Lemma B.5 and Lemma B.8, we can decompose $L_n^j$ as

$$
\begin{aligned}
L_n^j &= \Delta_{n-1}^j + \kappa \mathcal{E}_n^{j-1} \\
&\leq \left(1 - \frac{0.005}{\kappa}\right) \Delta_{n-2}^j + 4.4 \left(\frac{1}{M} + 10\kappa\right) h^2 \delta^2 + 4.4 \left(\frac{1}{M} + 10\kappa\right) \beta^2 h^2 \mathcal{E}_{n-1}^{j-1} \\
&\quad + \kappa(0.03^{P-1} \Delta_{n-1}^{j-1} + 2 \cdot 0.03^P \mathcal{E}_n^{j-2} + 7\delta^2 h^2) \\
&\leq \left(1 - \frac{0.005}{\kappa}\right) \Delta_{n-2}^j + \kappa \left(1 - \frac{0.005}{\kappa}\right) \mathcal{E}_{n-1}^{j-1} + \kappa \cdot 0.03^{P-1} \Delta_{n-1}^{j-1} + \kappa \cdot 0.03^{P-1} \cdot \kappa \mathcal{E}_n^{j-2} \\
&\quad + 56\kappa\delta^2 h^2 \\
&= \left(1 - \frac{0.005}{\kappa}\right) L_{n-1}^j + \left(\kappa \cdot 0.03^{P-1}\right) L_n^{j-1} + 56\kappa\delta^2 h^2. \tag{23}
\end{aligned}
$$

Combining $P \geq \frac{2\log\kappa}{3} + 4$ implies $\kappa \cdot 0.03^{P-1} \leq 0.04$, we recursively bound $L_n^j$ as

$$
\begin{aligned}
L_n^j &\leq \sum_{a=2}^n 0.04^{j-2} \binom{n-a+j-2}{j-2} L_a^2 + \sum_{b=2}^j \left(\kappa \cdot 0.03^{P-1}\right)^{j-b} \left(1 - \frac{0.005}{\kappa}\right)^{n-1} \binom{n-1+j-b}{j-b} L_1^b \\
&\quad + \sum_{a=2}^j \sum_{b=2}^n \left(1 - \frac{0.001}{\kappa}\right)^{n-b} 0.04^{j-a} 65\kappa\delta^2 h^2 \\
&\leq \sum_{a=2}^n 0.04^{j-2} \binom{n-a+j-2}{j-2} L_a^2 + \sum_{b=2}^j \left(\kappa \cdot 0.03^{P-1}\right)^{j-b} \left(1 - \frac{0.005}{\kappa}\right)^{n-1} \binom{n-1+j-b}{j-b} L_1^b \\
&\quad + 68000\kappa^2 \delta^2 h^2. \tag{24}
\end{aligned}
$$

For the first term $\sum_{a=2}^n 0.04^{j-2} \binom{n-a+j-2}{j-2} L_a^2$, we first bound $L_a^2$. To do so, we first bound $\Delta_n^2$ as follows. Combining Lemma B.5 and Corollary B.9, we have

$$
\begin{aligned}
\Delta_n^2 &\leq \left(1 - \frac{0.005}{\kappa}\right) \Delta_{n-1}^2 + 4.4 \left(\frac{1}{M} + 10\kappa\right) h^2 \delta^2 + 4.4 \left(\frac{1}{M} + 10\kappa\right) \beta^2 h^2 \mathcal{E}_n^1 \\
&\leq \Delta_{n-1}^2 + 48.4\kappa h^2 \delta^2 + 48.4\kappa\beta^2 h^2 \left(n \left(5.5\delta^2 h^2 + 0.1 \frac{\beta^2 h^2}{\alpha} \mathsf{KL}(\mu_0 \| \pi) + 0.1\kappa^2 dh\right)\right) \\
&\leq \Delta_{n-1}^2 + 48.4\kappa\beta^2 h^2 n \left(55.5\delta^2 h^2 + 0.1 \frac{\beta^2 h^2}{\alpha} \mathsf{KL}(\mu_0 \| \pi) + 0.1\kappa^2 dh\right) \\
&\leq \Delta_0^2 + 48.4\kappa\beta^2 h^2 n^2 \left(55.5\delta^2 h^2 + 0.1 \frac{\beta^2 h^2}{\alpha} \mathsf{KL}(\mu_0 \| \pi) + 0.1\kappa^2 dh\right) \\
&\leq 0.03^{2P} \frac{4\beta^2 h^2}{\alpha} \mathsf{KL}(\mu_0 \| \pi) + 1.4 \cdot 0.03^{2P} dh + 6.7\delta^2 h^2 \\
&\quad + 48.4\kappa\beta^2 h^2 n^2 \left(55.5\delta^2 h^2 + 0.1 \frac{\beta^2 h^2}{\alpha} \mathsf{KL}(\mu_0 \| \pi) + 0.1\kappa^2 dh\right) \\
&\leq 48.4\kappa\beta^2 h^2 n^2 \left(67.2\delta^2 h^2 + 0.2 \frac{\beta^2 h^2}{\alpha} \mathsf{KL}(\mu_0 \| \pi) + 0.2\kappa^2 dh\right).
\end{aligned}
$$

Thus

$$
\begin{aligned}
L_a^2 \; &= \Delta_{a-1}^2 + \kappa \mathcal{E}_a^1 \\
&\leq 0.49\kappa(a-1)^2 \left( 67.2\delta^2 h^2 + 0.2\frac{\beta^2 h^2}{\alpha} \mathsf{KL}(\mu_0\|\pi) + 0.2\kappa^2 dh \right) \\
&\quad + \kappa \left( a \left( 5.5\delta^2 h^2 + 0.1\frac{\beta^2 h^2}{\alpha} \mathsf{KL}(\mu_0\|\pi) + 0.1\kappa^2 dh \right) \right) \\
&\leq \kappa a^2 \left( 39\delta^2 h^2 + 0.2\frac{\beta^2 h^2}{\alpha} \mathsf{KL}(\mu_0\|\pi) + 0.2\kappa^2 dh \right).
\end{aligned}
$$

Thus by $\binom{m}{n} \leq \left(\frac{em}{n}\right)^n$ for $m \geq n > 0$, we have

$$
\begin{aligned}
&\sum_{a=2}^{n} 0.04^{j-2} \binom{n-a+j-2}{j-2} L_a^2 \\
&\leq \sum_{a=2}^{n} 0.04^{j-2} e^{j-2} \left( \frac{n-a+j-2}{j-2} \right)^{j-2} L_a^2 \\
&\leq \sum_{a=2}^{n} 0.04^{j-2} e^{2j-4} L_a^2 \\
&\leq \sum_{a=2}^{n} 0.3^{j-2} \kappa a^2 \left( 39\delta^2 h^2 + 0.2\frac{\beta^2 h^2}{\alpha} \mathsf{KL}(\mu_0\|\pi) + 0.2\kappa^2 dh \right) \\
&\leq 0.3^{j-2} \kappa n^3 \left( 39\delta^2 h^2 + 0.2\frac{\beta^2 h^2}{\alpha} \mathsf{KL}(\mu_0\|\pi) + 0.2\kappa^2 dh \right).
\end{aligned}
\tag{25}
$$

For the second term $\sum_{b=2}^{j} \left( \kappa \cdot 0.03^{P-1} \right)^{j-b} \left( 1 - \frac{0.005}{\kappa} \right)^{n-1} \binom{n-1+j-b}{j-b} L_1^b$, we first bound $L_1^b$. Firstly, for $\mathcal{E}_1^{b-1}$, combining Corollary B.7 and Corollary B.9, we have

$$
\begin{aligned}
\mathcal{E}_1^{b-1} \; &\leq 2 \cdot 0.03^{P-1} \Delta_0^{b-1} + 2 \cdot 0.03^P \mathcal{E}_1^{b-2} + 7\delta^2 h^2 \\
&\leq 2 \cdot 0.03^{P-1} \left( 0.03^{(b-1)P} \frac{4\beta^2 h^2}{\alpha} \mathsf{KL}(\mu_0\|\pi) + 1.4 \cdot 0.03^{(b-1)P} dh + 6.7\delta^2 h^2 \right) \\
&\quad + 2 \cdot 0.03^P \mathcal{E}_1^{b-2} + 7\delta^2 h^2 \\
&\leq 2 \cdot 0.03^P \mathcal{E}_1^{b-2} + 0.03^b \left( 0.01\frac{4\beta^2 h^2}{\alpha} \mathsf{KL}(\mu_0\|\pi) + 0.01 dh \right) + 7.1\delta^2 h^2 \\
&\leq (2 \cdot 0.03^P)^{b-2} \mathcal{E}_1^1 + \sum_{i=0}^{b-3} \left( 2 \cdot 0.03^P \right)^i \left( 0.03^{b-i} \left( 0.01\frac{4\beta^2 h^2}{\alpha} \mathsf{KL}(\mu_0\|\pi) + 0.01 dh \right) + 7.1\delta^2 h^2 \right) \\
&\leq (2 \cdot 0.03^P)^{b-2} \mathcal{E}_1^1 + \sum_{i=0}^{b-3} 0.01^i 0.03^i \left( 0.03^{b-i} \left( 0.01\frac{4\beta^2 h^2}{\alpha} \mathsf{KL}(\mu_0\|\pi) + 0.01 dh \right) + 7.1\delta^2 h^2 \right) \\
&\leq (2 \cdot 0.03^P)^{b-2} \left( 5.5\delta^2 h^2 + 0.1\frac{\beta^2 h^2}{\alpha} \mathsf{KL}(\mu_0\|\pi) + 0.1\kappa^2 dh \right) \\
&\quad + 0.03^b \left( 0.02\frac{4\beta^2 h^2}{\alpha} \mathsf{KL}(\mu_0\|\pi) + 0.02 dh \right) + 7.2\delta^2 h^2 \\
&\leq 0.03^b \left( 0.1\frac{\beta^2 h^2}{\alpha} \mathsf{KL}(\mu_0\|\pi) + 0.1 dh \right) + 7.3\delta^2 h^2.
\end{aligned}
$$

As for $\Delta_0^b$ we have

$$
\Delta_0^b \leq 0.03^{bP} \frac{4\beta^2 h^2}{\alpha} \mathsf{KL}(\mu_0\|\pi) + 1.4 \cdot 0.03^{bP} dh + 6.7\delta^2 h^2.
$$

Thus, we bound the first term as

$$
\begin{aligned}
L_1^b \;&= \Delta_0^b + \kappa \mathcal{E}_1^{b-1} \\
&\le 0.03^{bP} \frac{4\beta^2 h^2}{\alpha} \mathsf{KL}(\mu_0 \| \pi) + 1.4 \cdot 0.03^{bP} dh + 6.7 \delta^2 h^2 \\
&\quad + \kappa 0.03^b \left( 0.1 \frac{\beta^2 h^2}{\alpha} \mathsf{KL}(\mu_0 \| \pi) + 0.1 dh \right) + 7.3 \delta^2 h^2 \\
&\le \kappa 0.03^b \left( 0.2 \frac{\beta^2 h^2}{\alpha} \mathsf{KL}(\mu_0 \| \pi) + 0.2 dh \right) + 14 \delta^2 h^2.
\end{aligned}
$$

Thus by $\binom{m}{n} \le \left( \frac{em}{n} \right)^n$ for $m \ge n > 0$, and $\sum_{i=0}^{m} \binom{n+i}{n} x^i = \frac{1 - (m+1)\binom{m+n+1}{n} B_x(m+1, n+1)}{(1-x)^{n+1}} \le \frac{1}{(1-x)^{n+1}}$ we have

$$
\begin{aligned}
&\sum_{b=2}^{j} \left( \kappa \cdot 0.03^{P-1} \right)^{j-b} \left( 1 - \frac{0.005}{\kappa} \right)^{n-1} \binom{n-1+j-b}{j-b} L_1^b \\
&\le \sum_{b=2}^{j} 0.04^{j-b} \left( 1 - \frac{0.005}{\kappa} \right)^{n-1} \binom{n-1+j-b}{j-b} \left( \kappa 0.03^b \left( 0.2 \frac{\beta^2 h^2}{\alpha} \mathsf{KL}(\mu_0 \| \pi) + 0.2 dh \right) \right) \\
&\quad + \sum_{b=2}^{j} \left( \kappa \cdot 0.03^{P-1} \right)^{j-b} \left( 1 - \frac{0.005}{\kappa} \right)^{n-1} \binom{n-1+j-b}{j-b} 14 \delta^2 h^2 \\
&\le \sum_{b=2}^{j} 0.04^j \binom{n-1+j-b}{j-b} \kappa \left( 0.2 \frac{\beta^2 h^2}{\alpha} \mathsf{KL}(\mu_0 \| \pi) + 0.2 dh \right) \\
&\quad + \sum_{b=2}^{j} \left( \kappa \cdot 0.03^{P-1} \right)^{j-b} \left( 1 - \frac{0.005}{\kappa} \right)^{n-1} \binom{n-1+j-b}{j-b} 14 \delta^2 h^2 \\
&\le \sum_{i=0}^{j-2} 0.04^j e^i \left( 1 + \frac{n-1}{i} \right)^i \kappa \left( 0.2 \frac{\beta^2 h^2}{\alpha} \mathsf{KL}(\mu_0 \| \pi) + 0.2 dh \right) \\
&\quad + \sum_{i=0}^{j-2} \left( \kappa \cdot 0.03^{P-1} \right)^i \left( 1 - \frac{0.005}{\kappa} \right)^{n-1} \binom{n-1+i}{i} 14 \delta^2 h^2 \\
&\le 0.11^j e^{n-1} \kappa \left( 0.2 \frac{\beta^2 h^2}{\alpha} \mathsf{KL}(\mu_0 \| \pi) + 0.2 dh \right) \\
&\quad + \frac{1}{(1 - \kappa \cdot 0.03^{P-1})^n} \left( 1 - \frac{0.005}{\kappa} \right)^{n-1} (6.6 + 7.9\kappa) \delta^2 h^2 \\
&\le 0.11^j e^{n-1} \kappa \left( 0.2 \frac{\beta^2 h^2}{\alpha} \mathsf{KL}(\mu_0 \| \pi) + 0.2 dh \right) + \frac{1}{(1 - \kappa \cdot 0.03^{P-1})}(6.6 + 7.9\kappa) \delta^2 h^2 \\
&\le 0.11^j e^{n-1} \left( 2.2 \kappa \left( \frac{4\beta^2 h^2}{\alpha} \mathsf{KL}(\mu_0 \| \pi) + 1.6 dh + 2 \delta^2 h^2 \right) \right) + 20 \kappa \delta^2 h^2,
\end{aligned}
$$

where the second-to-last inequality is implied by $8 \left( \frac{1}{M} + 10\kappa \right) 0.03^P \le \frac{0.005}{\kappa}$.

Combing (24) and (25), we bound $L_n^j$ as

$$
\begin{aligned}
L_n^j &\leq \sum_{a=2}^n 0.04^{j-2}\binom{n-a+j-2}{j-2}L_a^2 + \sum_{b=2}^j \left(\kappa \cdot 0.03^{P-1}\right)^{j-b}\left(1-\frac{0.005}{\kappa}\right)^{n-1}\binom{n-1+j-b}{j-b}L_1^b \\
&\quad + 68000\kappa^2\delta^2 h^2 \\
&\leq 0.3^{j-2}\kappa n^3\left(39\delta^2 h^2 + 0.2\frac{\beta^2 h^2}{\alpha}\mathsf{KL}(\mu_0\|\pi) + 0.2\kappa^2 dh\right) \\
&\quad + 0.11^j e^{n-1}\left(2.2\kappa\left(\frac{4\beta^2 h^2}{\alpha}\mathsf{KL}(\mu_0\|\pi) + 1.6dh + 2\delta^2 h^2\right)\right) + 20\kappa\delta^2 h^2 + 68000\kappa^2\delta^2 h^2 \\
&\leq 0.3^{j-2}e^{n-1}\kappa n^3\left(41\delta^2 h^2 + 1.8\kappa^2 dh + 0.5\kappa h\mathsf{KL}(\mu_0\|\pi)\right) + 68020\kappa^2\delta^2 h^2.
\end{aligned}
$$

Since $8\left(\frac{1}{M}+10\kappa\right)0.03^P \leq \frac{0.005}{\kappa}$ implies $\kappa^2 0.03^{P-1} \leq 0.003$, we have

$$
\begin{aligned}
\mathcal{E}_n^j &\\
&\leq 2\cdot 0.03^{P-1}L_n^j + 7\delta^2 h^2 \\
&\leq 2\cdot 0.03^{P-1}\left(0.3^{j-2}e^{n-1}\kappa n^3\left(41\delta^2 h^2 + 1.8\kappa^2 dh + 0.5\kappa h\mathsf{KL}(\mu_0\|\pi)\right) + 68020\kappa^2\delta^2 h^2\right) + 7\delta^2 h^2 \\
&\leq 0.3^{j-2}e^{n-1}n^3\left(\delta^2 h^2 + h\mathsf{KL}(\mu_0\|\pi) + \kappa dh\right) + 416\delta^2 h^2.
\end{aligned}
$$

Thus when $J - N \geq \log\left(N^3\left(\frac{\kappa\delta^2 h + \kappa\mathsf{KL}(\mu_0\|\pi) + \kappa^2 d}{\varepsilon^2}\right)\right)$, we have for any $n = 0, \ldots, N-1$

$$
\mathcal{E}_n^J \leq \frac{\varepsilon^2}{5\kappa\beta} + 416\delta^2 h^2.
$$

Recall

$$
\mathsf{KL}_{N-1}^J \leq e^{-1.2\alpha(N-1)h}\left(\mathsf{KL}(\mu_0\|\pi) + 4.4\beta^2 h\Delta_0^J\right) + 5\kappa\beta\mathcal{E} + \frac{0.6\beta d}{\alpha M} + \frac{28\delta^2}{\alpha},
$$

thus when $\delta^2 \leq \frac{\alpha\varepsilon^2}{29}$, $M \geq \frac{\kappa d}{\varepsilon^2}$, and $N \geq 10\kappa\log\frac{\mathsf{KL}(\mu_0\|\pi)}{\varepsilon^2}$, we have

$$
\begin{aligned}
\mathsf{KL}_{N-1}^J &\leq e^{-1.2\alpha(N-1)h}\left(\mathsf{KL}(\mu_0\|\pi) + 4.4\beta^2 h\Delta_0^J\right) + 5\kappa\beta\mathcal{E} + \frac{0.6\beta d}{\alpha M} + \frac{28\delta^2}{\alpha} \\
&\leq e^{-1.2\alpha(N-1)h}\left(\mathsf{KL}(\mu_0\|\pi) + 4.4\beta^2 h\left(0.03^{JP}\frac{4\beta^2 h^2}{\alpha}\mathsf{KL}(\mu_0\|\pi) + 1.4\cdot 0.03^{JP}dh + 6.7\delta^2 h^2\right)\right) \\
&\quad + 5\kappa\beta\mathcal{E} + \frac{0.6\beta d}{\alpha M} + \frac{28\delta^2}{\alpha} \\
&\leq e^{-1.2\alpha(N-1)h}\mathsf{KL}(\mu_0\|\pi) + \varepsilon^2 + 5\kappa\beta\mathcal{E} + \frac{0.6\beta d}{\alpha M} + \frac{29\delta^2}{\alpha} \\
&\leq 5\varepsilon^2.
\end{aligned}
$$

## C. Missing Details for Sampling for Diffusion Models

In this section, we begin by presenting the algorithm details in Appendix C.1. In Appendix C.2, following the approach of Chen et al. (2024), we apply Girsanov's Theorem and the interpolation method to decompose the KL divergence and bound the discretization error, accounting for the influence of the step size scheme and the estimation error of the score function. Finally, in Appendix C.3, we analyze the additional parallelization error and derive the overall error bound.

### C.1. Algorithm

In the parallel Picard method for diffusion model, we use the similar parallelization across time slices as illustrated in Figure 1. In Lines 2–6, we generate the noises and fix them. In Lines 7–10, we initialize the value at the grid via sequential method with a stepsize $h_n = \mathcal{O}(1)$. In Lines 12–21, we update the grids diagonally, using the exponential integrator in Lines 14 and 19 instead of the Euler-Maruyama scheme. The step size scheme also differs from that used for log-concave

sampling. Here, we follow the discretization scheme with early stopping and exponential decay described in Chen et al. (2024, Section 3.1.2).

---

**Algorithm 2** Parallel Picard Iteration Method for diffusion models

---

1: **Input:** $\widehat{\boldsymbol{y}}_0 \sim \widehat{\boldsymbol{q}}_0 = \mathcal{N}(0, I_d)$, the learned NN-based score function $\boldsymbol{s}_t^\theta(\cdot)$, the depth of Picard iterations $J$, the depth of inner Picard iteration $P$, and a discretization scheme $(T, (h_n)_{n=1}^N$ and $(\tau_{n,m})_{n\in[0:N-1], m\in[0:M]})$.

2: **for** $n = 0, \ldots, N-1$ **do**

3:    **for** $m = 0, \ldots, M$ (in parallel) **do**

4:       $\boldsymbol{\xi}_{n,m} \sim \mathcal{N}(0, I_d)$

5:    **end for**

6: **end for**

7: **for** $n = 0, \ldots, N-1$ **do**

8:    **for** $m = 0, \ldots, M_n$ (in parallel) **do**

9:       $\widehat{\boldsymbol{y}}_{-1,M}^j = \widehat{\boldsymbol{y}}_0$, for $j = 0, \ldots, J$,

$$
\begin{aligned}
\widehat{\boldsymbol{y}}_{n,\tau_{n,m}}^0 &= e^{\frac{\tau_{n,m}}{2}} \widehat{\boldsymbol{y}}_{n-1,\tau_{n,M}}^0 \\
&+ \sum_{m'=0}^{m-1} e^{\frac{\tau_{n,m}-\tau_{n,m'+1}}{2}} \left[ 2(e^{\epsilon_{n,m'}} - 1) \boldsymbol{s}_{t_n+\tau_{n,m'}}^\theta(\widehat{\boldsymbol{y}}_{n-1,\tau_{n,M}}^0) + \sqrt{e^{\epsilon_{n,m'}} - 1} \boldsymbol{\xi}_{m'} \right],
\end{aligned}
\tag{26}
$$

10:    **end for**

11: **end for**

12: **for** $k = 1, \ldots, N$ **do**

13:    **for** $j = 1, \ldots, \min\{k-1, J\}$ and $m = 0, \ldots, M_n$ (in parallel) **do**

14:       let $n = k - j$, and $\widehat{\boldsymbol{y}}_{n,0}^j = \widehat{\boldsymbol{y}}_{n-1,M_n}^j$,

$$
\begin{aligned}
\widehat{\boldsymbol{y}}_{n,\tau_{n,m}}^j &= e^{\frac{\tau_{n,m}}{2}} \widehat{\boldsymbol{y}}_{n,0}^j \\
&+ \sum_{m'=0}^{m-1} e^{\frac{\tau_{n,m}-\tau_{n,m'+1}}{2}} \left[ 2(e^{\epsilon_{n,m'}} - 1) \boldsymbol{s}_{t_n+\tau_{n,m'}}^\theta(\widehat{\boldsymbol{y}}_{n,\tau_{n,m'}}^{j-1}) + \sqrt{e^{\epsilon_{n,m'}} - 1} \boldsymbol{\xi}_{m'} \right],
\end{aligned}
\tag{27}
$$

15:    **end for**

16: **end for**

17: **for** $k = N+1, \ldots, N+J-1$ **do**

18:    **for** $n = \max\{0, k-J\}, \ldots, N-1$ and $m = 0, \ldots, M_n$ (in parallel) **do**

19:       let $j = k - n$, and $\widehat{\boldsymbol{y}}_{n,0}^j = \widehat{\boldsymbol{y}}_{n-1,M_n}^j$,

$$
\begin{aligned}
\widehat{\boldsymbol{y}}_{n,\tau_{n,m}}^j &= e^{\frac{\tau_{n,m}}{2}} \widehat{\boldsymbol{y}}_{n,0}^j \\
&+ \sum_{m'=0}^{m-1} e^{\frac{\tau_{n,m}-\tau_{n,m'+1}}{2}} \left[ 2(e^{\epsilon_{n,m'}} - 1) \boldsymbol{s}_{t_n+\tau_{n,m'}}^\theta(\widehat{\boldsymbol{y}}_{n,\tau_{n,m'}}^{j-1}) + \sqrt{e^{\epsilon_{n,m'}} - 1} \boldsymbol{\xi}_{m'} \right],
\end{aligned}
\tag{28}
$$

20:    **end for**

21: **end for**

22: **Return:** $\widehat{\boldsymbol{y}}_{N-1,M_{N-1}}^J$.

---

**Stepsize scheme.** We first present the stepsize schedule for diffusion models, which is the same as the discretization scheme in Chen et al. (2024). Specifically, we split the the time horizon $T$ into $N$ time slices with length $h_n \leq h = \frac{T}{N} = \Omega(1)$, and a large gap grid $(t_n)_{n=0}^N$ with $t_n = \sum_{i=1}^n h_i$. For any $n \in [0:N-1]$, we further split the $n$-th time slice into a grid $(\tau_{n,m})_{m=0}^{M_n}$ with $\tau_{n,0} = 0$ and $\tau_{n,M_n} = h_n$. We denote the step size of the $m$-th step in the $n$-th time slice as $\epsilon_{n,m} = \tau_{n,m+1} - \tau_{n,m}$, and let the total number of grids in the $n$-th time slice as $M_n$. The grids $(\tau_{n,m})_{m=0}^{M_n}$ is scheduled as follows,

1. for the first $N-1$ time slice, we use the uniform discretization: for $n = 0, \ldots, N-2$ and $m = 0, \ldots, M-1$.

$$
h_n = h, \quad \epsilon_{n,m} = \epsilon, \text{ and } M_n = M = \frac{h}{\epsilon},
$$

2. for the last time slice, we apply early stopping and exponential decay:

$$h_{N-1} = h - \delta, \quad \epsilon_{N-1,m} \le \epsilon \wedge \epsilon\left(h - \tau_{N-1,m+1}\right).$$

We also define the indexing functions as follows: for $\tau \in [t_n, t_{n+1}]$, we define $I_n(\tau) \in \mathbb{N}$ such that $\sum_{j=1}^{I_n(\tau)} \epsilon_{n,j} \le \tau < \sum_{j=1}^{I_n(\tau)+1} \epsilon_{n,j}$. We further define a piecewise function $g$ such that $g_n(\tau) = \sum_{j=1}^{I_n(\tau)} \epsilon_{n,j}$ and thus we have $I_n(\tau) = \lfloor \tau/\epsilon \rfloor$ and $g_n(\tau) = \lfloor \tau/\epsilon \rfloor \epsilon$.

**Exponential integrator for Picard iterations.** Compared with Line 12 and Line 18 in Algorithm 1, where we use a forward Euler-Maruyama scheme for Picard iterations, we use the following exponential integrator scheme (Zhang & Chen, 2023; Chen et al., 2024). Specifically, In $n$-th time slice $[t_n, t_n + \tau_{n,M_n}]$, for each grid $t_n + \tau_{n,m}$, we simulate the approximated backward process (3) with Picard iterations as

$$
\begin{aligned}
\widehat{\boldsymbol{y}}_{n,\tau_{n,m}}^{j+1} &= e^{\frac{\tau_{n,m}}{2}} \widehat{\boldsymbol{y}}_{n-1,\tau_{n,M}}^{j+1} \\
&+ \sum_{m'=0}^{m-1} e^{\frac{\tau_{n,m} - \tau_{n,m'+1}}{2}} \left[ 2(e^{\epsilon_{n,m'}} - 1) \boldsymbol{s}_{t_n+\tau_{n,m'}}^{\theta}(\widehat{\boldsymbol{y}}_{n-1,\tau_{n,M}}^{j}) + \sqrt{e^{\epsilon_{n,m'}} - 1} \boldsymbol{\xi}_{m'} \right].
\end{aligned}
$$

We note such update also inherently allows for parallelization for $m = 1, \dots, M_n$.

### C.2. Interpolation Processes and Decomposition of KL Divergence

Following the proof flow in Chen et al. (2024), we define the following processes:

1. the original backward process,

$$\mathrm{d}\breve{\boldsymbol{x}}_t = \left[ \frac{1}{2}\breve{\boldsymbol{x}}_t + \nabla \log \breve{p}_t(\breve{\boldsymbol{x}}_t)\mathrm{d}_t \right] + \mathrm{d}\boldsymbol{w}_t, \quad \text{with} \quad \breve{\boldsymbol{x}}_0 \sim p_T; \tag{29}$$

2. the approximated backward process,

$$\mathrm{d}\boldsymbol{y}_t = \left[ \frac{1}{2}\boldsymbol{y}_t + \boldsymbol{s}_t^{\theta}(\boldsymbol{y}_t) \right] \mathrm{d}t + \mathrm{d}\boldsymbol{w}_t, \quad \text{with} \quad \boldsymbol{y}_0 \sim \mathcal{N}(0, I_d);$$

3. the interpolation processes $(\widehat{\boldsymbol{y}}_{t_n,\tau}^{j})_{\tau \in [0,h]}$ over $\tau \in [0, h]$ conditioned on the filtration of the backward SDE (29) up to time $t$ $\mathcal{F}_t$, for any fixed $n = 0, \dots, N-1, j = 1, \dots, J$,

$$\mathrm{d}\widehat{\boldsymbol{y}}_{t_n,\tau}^{j}(\omega) = \left[ \frac{1}{2}\widehat{\boldsymbol{y}}_{t_n,\tau}^{j}(\omega) + \boldsymbol{s}_{t_n+g_n(\tau)}^{\theta}\left(\widehat{\boldsymbol{y}}_{t_n,g_n(\tau)}^{j-1}(\omega)\right) \right] \mathrm{d}\tau + \mathrm{d}\boldsymbol{w}_{t_n+\tau}(\omega); \tag{30}$$

with $\widehat{\boldsymbol{y}}_{t_n,0}^{j}(\omega) = \widehat{\boldsymbol{y}}_{t_{n-1},\tau_{n-1,M_{n-1}}}^{j}(\omega)$.

4. the initialization process,

$$\mathrm{d}\widehat{\boldsymbol{y}}_{t_n,\tau}^{0}(\omega) = \left[ \frac{1}{2}\widehat{\boldsymbol{y}}_{t_n,\tau}^{0}(\omega) + \boldsymbol{s}_{t_n+g_n(\tau)}^{\theta}\left(\widehat{\boldsymbol{y}}_{t_{n-1},\tau_{n-1,M}}^{0}(\omega)\right) \right] \mathrm{d}\tau + \mathrm{d}\boldsymbol{w}_{t_n+\tau}(\omega), \tag{31}$$

with $\widehat{\boldsymbol{y}}_{t_0,0}^{0} = \widehat{\boldsymbol{y}}_0$ and $\widehat{\boldsymbol{y}}_{t_n,0}^{0} = \widehat{\boldsymbol{y}}_{t_{n-1},\tau_{n-1,M}}$.

*Remark* C.1. The main difference compared to the auxiliary process defined in Chen et al. (2024) is the change of the start point across each update.

We can demonstrate that the interpolation processes remain well-defined after parallelization across time slices.

**Lemma C.2.** *The auxiliary process* $(\widehat{\boldsymbol{y}}_{t_n,\tau}^j(\omega))_{\tau\in[0,h_n]}$ *is* $\mathcal{F}_{t_n+\tau}$*-adapted for any* $j=1,\ldots,j$ *and* $n=0,\ldots,n-1$.

*Proof.* Since the initialization $\widehat{\boldsymbol{y}}_{t_n,\tau}^0(\omega)$ satisfies

$$\mathrm{d}\widehat{\boldsymbol{y}}_{t_n,\tau}^0(\omega) = \left[\frac{1}{2}\widehat{\boldsymbol{y}}_{t_n,\tau}^0(\omega) + \boldsymbol{s}_{t_n+g_n(\tau)}^\theta\left(\widehat{\boldsymbol{y}}_{t_{n-1},\tau_{n-1,M}}^0(\omega)\right)\right]\mathrm{d}\tau + \mathrm{d}\boldsymbol{w}_{t_n+\tau}(\omega),$$

we can claim $\widehat{\boldsymbol{y}}_{t_n,\tau}^0(\omega)$ is $\mathcal{F}_{t_n+\tau}$-adapted. Assume $\boldsymbol{y}_{t_n,\tau}$ is $\mathcal{F}_{t_n+\tau}$-adapted, by $g_n(\tau) \leq \tau$, the Itô integral $\int_0^\tau \boldsymbol{s}_{t_n+g_n(\tau')}^\theta(\boldsymbol{y}_{t_n,g_n(\tau')})\mathrm{d}\tau'$ is well-defined and $\mathcal{F}_{t_n+\tau}$-adapted. Therefore SDE

$$\mathrm{d}\boldsymbol{y}_{t_n,\tau}'(\omega) = \left[\frac{1}{2}\boldsymbol{y}_{t_n,\tau}'(\omega) + \boldsymbol{s}_{t_n+g_n(\tau)}^\theta\left(\boldsymbol{y}_{t_n,g_n(\tau)}(\omega)\right)\right]\mathrm{d}\tau + \mathrm{d}\boldsymbol{w}_{t_n+\tau}(\omega)$$

has a unique strong solution $(\boldsymbol{y}_{t_n,\tau}'(\omega))_{\tau\in[0,h_n]}$ that is also $\mathcal{F}_{t_n+\tau}$-adapted. The lemma is established through induction. $\square$

Finally, the following lemma shows the equivalence of our update rule and the auxiliary process, *i.e.*, the auxiliary process is an interpotation of the discrete points.

**Lemma C.3.** *For any* $n=0,\ldots,N-1$, *the update rule ((26)) in Algorithm 2 and the update rule ((27) or (28)) are equivalent to the exact solution of the auxiliary process* (31), *and* (30) *respectively, for any* $j=1,\ldots,J$, *and* $\tau\in[0,h_n]$.

*Proof.* Due to the similarity, we only prove the equivalence of the update rule ((26)). The dependency on $\omega$ will be omitted in the proof below.

For SDE (30), by multiplying $e^{-\frac{\tau}{2}}$ on both sides then integrating on both side from $0$ to $\tau$, we have

$$e^{-\frac{\tau}{2}}\widehat{\boldsymbol{y}}_{t_n,\tau}^j - \widehat{\boldsymbol{y}}_{t_n,0}^j = \sum_{m=0}^{M_n} 2\left(e^{-\frac{\tau\wedge\tau_{n,m}}{2}} - e^{-\frac{\tau\wedge\tau_{n,m+1}}{2}}\right)\boldsymbol{s}_{t_n+\tau_{n,m}}^\theta\left(\widehat{\boldsymbol{y}}_{t_n,\tau_{n,m}}^{j-1}\right) + \int_0^\tau e^{-\frac{\tau'}{2}}\mathrm{d}\boldsymbol{w}_{t_n+\tau'}.$$

Thus

$$\widehat{\boldsymbol{y}}_{t_n,\tau}^j = e^{\frac{\tau}{2}}\widehat{\boldsymbol{y}}_{t_n,0}^j + \sum_{m=0}^{M_n} 2\left(e^{-\frac{\tau\wedge\tau_{n,m}-\tau\wedge\tau_{n,m+1}}{2}} - 1\right)e^{\frac{0\vee(\tau-\tau_{n,m+1})}{2}}\boldsymbol{s}_{t_n+\tau_{n,m}}^\theta\left(\widehat{\boldsymbol{y}}_{t_n,\tau_{n,m}}^{j-1}\right)$$
$$+ \sum_{m=0}^{M_n}\int_{\tau\wedge\tau_{n,m}}^{\tau\wedge\tau_{n,m+1}} e^{\frac{\tau-\tau'}{2}}\mathrm{d}\boldsymbol{w}_{t_n+\tau'}.$$

By Itô isometry and let $\tau=\tau_{n,m}$ we get the desired result. $\square$

### C.2.1. DECOMPOSITION OF KL DIVERGENCE

Similar as the analysis in Section B.2 of Chen et al. (2024), we conclude the following lemma by Corollary A.3.

**Lemma C.4.** *Assume* $\boldsymbol{\delta}_{t_n}(\tau,\omega) = \boldsymbol{s}_{t_n+g_n(\tau)}^\theta(\widehat{\boldsymbol{y}}_{t_n,g_n(\tau)}^{J-1}(\omega)) - \nabla\log\bar{p}_{t_n+\tau}(\widehat{\boldsymbol{y}}_{t_n,\tau}^J(\omega))$. *Then we have the following one-step decomposition,*

$$\mathsf{KL}(\bar{p}_{t_{n+1}}\|\widehat{q}_{t_{n+1}}) \leq \mathsf{KL}(\bar{p}_{t_n}\|\widehat{q}_{t_n}) + \mathbb{E}_{\omega\sim q|_{\mathcal{F}_{t_n}}}\left[\frac{1}{2}\int_0^{h_n}\|\boldsymbol{\delta}_{t_n}(\tau,\omega)\|^2\,\mathrm{d}\tau\right].$$

Now, the problem remaining is reduced to bound the following discrepancy,

$$
\int_0^{h_n} \|\boldsymbol{\delta}_{t_n}(\tau,\omega)\|^2 \mathrm{d}\tau
$$

$$
= \int_0^{h_n} \left\| \boldsymbol{s}_{t_n+g_n(\tau)}^\theta(\widehat{\boldsymbol{y}}_{t_n,g_n(\tau)}^{J-1}(\omega)) - \nabla \log \breve{p}_{t_n+\tau}(\widehat{\boldsymbol{y}}_{t_n,\tau}^J(\omega)) \right\|^2 \mathrm{d}\tau
$$

$$
\leq 3 \left( \underbrace{\int_0^{h_n} \left\| \nabla \log \breve{p}_{t_n+g_n(\tau)}(\widehat{\boldsymbol{y}}_{t_n,g_n(\tau)}^J(\omega)) - \nabla \log \breve{p}_{t_n+\tau}(\widehat{\boldsymbol{y}}_{t_n,\tau}^J(\omega)) \right\|^2 \mathrm{d}\tau}_{:=A_{t_n}(\omega)} \right.
$$

$$
+ \underbrace{\int_0^{h_n} \left\| \boldsymbol{s}_{t_n+g_n(\tau)}^\theta(\widehat{\boldsymbol{y}}_{t_n,g_n(\tau)}^J(\omega)) - \nabla \log \breve{p}_{t_n+g_n(\tau)}(\widehat{\boldsymbol{y}}_{t_n,g_n(\tau)}^J(\omega)) \right\|^2 \mathrm{d}\tau}_{:=B_{t_n}(\omega)}
$$

$$
\left. + \underbrace{\int_0^{h_n} \left\| \boldsymbol{s}_{t_n+g_n(\tau)}^\theta(\widehat{\boldsymbol{y}}_{t_n,g_n(\tau)}^J(\omega)) - \boldsymbol{s}_{t_n+g_n(\tau)}^\theta(\widehat{\boldsymbol{y}}_{t_n,g_n(\tau)}^{J-1}(\omega)) \right\|^2 \mathrm{d}\tau}_{:=C_{t_n}(\omega)} \right), \tag{32}
$$

where $A_{t_n}(\omega)$ measures the discretization error, $B_{t_n}(\omega)$ measures the estimation error of score function, and $C_{t_n}(\omega)$ measures the error by Picard iteration.

### C.2.2. DISCRETIZATION ERROR AND ESTIMATION ERROR OF SCORE FUNCTION IN EVERY TIME SLICE

The following lemma from Benton et al. (2024); Chen et al. (2024) bounds the expectation of the discretization error $A_{t_n}$.

**Lemma C.5 (Discretization error (Benton et al., 2024, Section 3.1) or (Chen et al., 2024, Lemma B.7)).** *We have for* $n \in [0:N-2]$

$$
\mathbb{E}_{\omega \sim \breve{p}|_{\mathcal{F}_{t_n}}} \left[ A_{t_n}(\omega) \right] \lesssim \epsilon d h_n,
$$

*and*

$$
\mathbb{E}_{\omega \sim \breve{p}|_{\mathcal{F}_{t_n}}} \left[ A_{t_{N-1}}(\omega) \right] \lesssim \epsilon d \log \eta^{-1},
$$

*where $\eta$ is the parameter for early stopping.*

The following lemma from Chen et al. (2024) bounds the expectation of the estimation error of score function, $B_{t_n}$, and we restate the proof for the convenience.

**Lemma C.6 (Estimation error of score function (Chen et al., 2024, Section B.3)).** $\sum_{n=0}^{N-1} \mathbb{E}_{\omega \sim \breve{p}|_{\mathcal{F}_{t_n}}} [B_{t_n}] \leq \delta_2^2.$

*Proof.* By Assumption 5.1 and the the fact that the process $\widehat{\boldsymbol{y}}_{t_n,\tau}^J(\omega)$ follows the backward SDE with the true score function under the measure $\breve{p}$, we have

$$
\sum_{n=1}^{N-1} \mathbb{E}_{\omega \sim \breve{p}|_{\mathcal{F}_{t_n}}} \left[ B_{t_n}(\omega) \right]
$$

$$
\leq \mathbb{E}_{\omega \sim \breve{p}|_{\mathcal{F}_{t_n}}} \left[ \sum_{n=1}^{N-1} \int_0^{h_n} \left\| \boldsymbol{s}_{t_n+\tau}^\theta(\widehat{\boldsymbol{y}}_{t_n,\tau}^J(\omega)) - \nabla \log \breve{p}_{t_n+g_n(\tau)}(\widehat{\boldsymbol{y}}_{t_n,\tau}^J(\omega)) \right\|^2 \mathrm{d}\tau \right]
$$

$$
= \mathbb{E}_{\omega \sim \breve{p}|_{\mathcal{F}_{t_n}}} \left[ \sum_{n=1}^{N-1} \sum_{m=0}^{M_n} \epsilon_{n,m} \left\| \boldsymbol{s}_{t_n+\tau}^\theta(\widehat{\boldsymbol{y}}_{t_n,\tau}^J(\omega)) - \nabla \log \breve{p}_{t_n+g_n(\tau)}(\widehat{\boldsymbol{y}}_{t_n,\tau}^J(\omega)) \right\|^2 \mathrm{d}\tau \right]
$$

$$
= \mathbb{E}_{\omega \sim \breve{p}|_{\mathcal{F}_{t_n}}} \left[ \sum_{n=0}^{N-1} \sum_{m=0}^{M_n} \epsilon_{n,m} \left\| \boldsymbol{s}_{t_n+\tau}^\theta(\breve{\boldsymbol{x}}_{t_n+\tau}(\omega)) - \nabla \log \breve{p}_{t_n+g_n(\tau)}(\breve{\boldsymbol{x}}_{t_n+\tau}(\omega)) \right\|^2 \mathrm{d}\tau \right]
$$

$$
\leq \delta_2^2.
$$

☐

### C.2.3. ANALYSIS FOR INITIALIZATION

By setting the depth of iteration as $J = 1$ in Chen et al. (2024), our initialization parts (Lines 4-7 in Algorithm 2) and the initialization process ((31)) are identical to the Algorithm 1 and the the auxiliary process (Definition B.1) in Chen et al. (2024). We provide a brief overview of their analysis and reformulate it to align with our initialization. Let

$$A_{t_n}^0(\omega) := \int_0^{h_n} \left\| \nabla \log \breve{p}_{t_n+g_n(\tau)}(\widehat{\boldsymbol{y}}_{t_n,g_n(\tau)}^0(\omega)) - \nabla \log \breve{p}_{t_n+\tau}(\widehat{\boldsymbol{y}}_{t_n,\tau}^0(\omega)) \right\|^2 \mathrm{d}\tau$$

and

$$B_{t_n}^0(\omega) := \int_0^{h_n} \left\| \boldsymbol{s}_{t_n+g_n(\tau)}^\theta(\widehat{\boldsymbol{y}}_{t_n,g_n(\tau)}^0(\omega)) - \nabla \log \breve{p}_{t_n+g_n(\tau)}(\widehat{\boldsymbol{y}}_{t_n,g_n(\tau)}^0(\omega)) \right\|^2 \mathrm{d}\tau$$

**Lemma C.7 (Lemma B.5 or Lemma B.6 with $K = 1$ in Chen et al. (2024)).** *For any $n = 0, \ldots, N - 1$, suppose the initialization $\widehat{\boldsymbol{y}}_{t_n,0}^0$ follows the distribution of $\breve{x}_{t_n} \sim \breve{p}_{t_n}$, if $3e^{\frac{7}{2}h_n}h_n L_{\boldsymbol{s}} < 0.5$, then the following estimate*

$$\sup_{\tau \in [0,h_n]} \mathbb{E}_{\omega \sim \breve{p}|\mathcal{F}_{t_n}} \left[ \left\| \widehat{\boldsymbol{y}}_{t_n,\tau}^0(\omega) - \widehat{\boldsymbol{y}}_{t_n,0}^0(\omega) \right\|^2 \right] \le 2h_n e^{\frac{7}{2}h_n}(M_{\boldsymbol{s}} + 2d)$$

$$+ 6e^{\frac{7}{2}h_n} \mathbb{E}_{\omega \sim \breve{p}|\mathcal{F}_{t_n}} \left[ A_{t_n}^0(\omega) + B_{t_n}^0(\omega) \right].$$

Furthermore, the $A_{t_n}^0(\omega)$ and $B_{t_n}^0(\omega)$ can be bounded as

**Lemma C.8 ((Chen et al., 2024, Lemma B.7)).** *We have for $n \in [0 : N - 2]$*

$$\mathbb{E}_{\omega \sim \breve{p}|\mathcal{F}_{t_n}} \left[ A_{t_n}^0(\omega) \right] \lesssim \epsilon d h_n,$$

*and*

$$\mathbb{E}_{\omega \sim \breve{p}|\mathcal{F}_{t_n}} \left[ A_{t_{N-1}}^0(\omega) \right] \lesssim \epsilon d \log \eta^{-1},$$

*where $\eta$ is the parameter for early stopping.*

**Lemma C.9 ((Chen et al., 2024, Section B.3)).** $\sum_{n=1}^{N-1} \mathbb{E}_{\omega \sim \breve{p}|\mathcal{F}_{t_n}} \left[ B_{t_n}^0(\omega) \right] \le \delta_2^2.$

Thus we have the following uniform bound for our initialization.

**Corollary C.10.** *With the same assumption in Lemma C.7, we have*

$$\sup_{n=0,\ldots,N} \sup_{\tau \in [0,h_n]} \mathbb{E}_{\omega \sim \breve{p}|\mathcal{F}_{t_n}} \left[ \left\| \widehat{\boldsymbol{y}}_{t_n,\tau}^0(\omega) - \widehat{\boldsymbol{y}}_{t_n,0}^0(\omega) \right\|^2 \right] \lesssim d.$$

### C.3. Convergence of Picard iteration

Similarly, we define

$$\mathcal{E}_n^j = \sup_{\tau \in [0,h_n]} \mathbb{E}_{\omega \sim \breve{p}|\mathcal{F}_{t_n}} \left[ \|\widehat{\boldsymbol{y}}_{t_n,\tau}^j(\omega) - \widehat{\boldsymbol{y}}_{t_n,\tau}^{j-1}(\omega)\|^2 \right],$$

and

$$\Delta_n^j = \mathbb{E}_{\omega \sim \breve{p}|\mathcal{F}_{t_n}} \left[ \|\widehat{\boldsymbol{y}}_{t_n,\tau_{n,M}}^j(\omega) - \widehat{\boldsymbol{y}}_{t_n,\tau_{n,M}}^{j-1}(\omega)\|^2 \right].$$

Furthermore, we let $\mathcal{E}_I = \sup_{n=0,\ldots,N-1} \sup_{\tau \in [0,h_n]} \mathbb{E}_{\omega \sim \breve{p}|\mathcal{F}_{t_n}} \left[ \left\| \widehat{\boldsymbol{y}}_{n,\tau}^0 - \widehat{\boldsymbol{y}}_{n-1,\tau_{n,M}}^0 \right\|^2 \right]$. We note that by Corollary C.10, $\mathcal{E}_I \lesssim d$.

**Lemma C.11 (One-step decomposition of $\mathcal{E}_n^j$).** *Assume $L_{\boldsymbol{s}}^2 e^{2h_n} h_n \le 0.01$ and $e^{2h_n} \le 2$. For any $j = 2, \ldots, J$, $n = 0, \ldots, N - 1$, we have*

$$\mathcal{E}_n^j \le 2\Delta_{n-1}^j + 0.01\mathcal{E}_n^{j-1}.$$

*Furthermore, for $j = 1$, $n = 1, \ldots, N - 1$, we have*

$$\mathcal{E}_n^1 \le 2\Delta_n^1 + 0.01 \left( \sup_{\tau \in [0,h_n]} \mathbb{E}_{\omega \sim \breve{p}|\mathcal{F}_{t_n}} \left\| \widehat{\boldsymbol{y}}_{t_n,\tau}^0(\omega) - \widehat{\boldsymbol{y}}_{t_{n-1},\tau_{n-1,M}}^0(\omega) \right\|^2 \right).$$

*Proof.* For each $\omega \in \Omega$ conditioned on the filtration $\mathcal{F}_{t_n}$, consider the auxiliary process defined as in the previous section,

$$d\widehat{\boldsymbol{y}}_{t_n,\tau}^j(\omega) = \left[\frac{1}{2}\widehat{\boldsymbol{y}}_{t_n,\tau}^j(\omega) + \boldsymbol{s}_{t_n+g_n(\tau)}^\theta\left(\widehat{\boldsymbol{y}}_{t_n,g_n(\tau)}^{j-1}(\omega)\right)\right] d\tau + d\boldsymbol{w}_{t_n+\tau}(\omega),$$

and

$$d\widehat{\boldsymbol{y}}_{t_n,\tau}^{j-1}(\omega) = \left[\frac{1}{2}\widehat{\boldsymbol{y}}_{t_n,\tau}^{j-1}(\omega) + \boldsymbol{s}_{t_n+g_n(\tau)}^\theta\left(\widehat{\boldsymbol{y}}_{t_n,g_n(\tau)}^{j-2}(\omega)\right)\right] d\tau + d\boldsymbol{w}_{t_n+\tau}(\omega).$$

We have

$$d\left(\widehat{\boldsymbol{y}}_{t_n,\tau}^j(\omega) - \widehat{\boldsymbol{y}}_{t_n,\tau}^{j-1}(\omega)\right)$$
$$= \left[\frac{1}{2}\left(\widehat{\boldsymbol{y}}_{t_n,\tau}^j(\omega) - \widehat{\boldsymbol{y}}_{t_n,\tau}^{j-1}(\omega)\right) + \boldsymbol{s}_{t_n+g_n(\tau)}^\theta\left(\widehat{\boldsymbol{y}}_{t_n,g_n(\tau)}^{j-1}(\omega)\right) - \boldsymbol{s}_{t_n+g_n(\tau)}^\theta\left(\widehat{\boldsymbol{y}}_{t_n,g_n(\tau)}^{j-2}(\omega)\right)\right] d\tau.$$

Then we can calculate the derivative $\frac{d}{d\tau}\left\|\widehat{\boldsymbol{y}}_{t_n,\tau}^j(\omega) - \widehat{\boldsymbol{y}}_{t_n,\tau}^{j-1}(\omega)\right\|^2$ as

$$\frac{d}{d\tau}\left\|\widehat{\boldsymbol{y}}_{t_n,\tau}^j(\omega) - \widehat{\boldsymbol{y}}_{t_n,\tau}^{j-1}(\omega)\right\|^2$$
$$= 2\left(\widehat{\boldsymbol{y}}_{t_n,\tau}^j(\omega) - \widehat{\boldsymbol{y}}_{t_n,\tau}^{j-1}(\omega)\right)^\top \left[\frac{1}{2}\left(\widehat{\boldsymbol{y}}_{t_n,\tau}^j(\omega) - \widehat{\boldsymbol{y}}_{t_n,\tau}^{j-1}(\omega)\right) + \boldsymbol{s}_{t_n+g_n(\tau)}^\theta\left(\widehat{\boldsymbol{y}}_{t_n,g_n(\tau)}^{j-1}(\omega)\right) - \boldsymbol{s}_{t_n+g_n(\tau)}^\theta\left(\widehat{\boldsymbol{y}}_{t_n,g_n(\tau)}^{j-2}(\omega)\right)\right].$$

By integrating from $0$ to $\tau$, we have

$$\left\|\widehat{\boldsymbol{y}}_{t_n,\tau}^j(\omega) - \widehat{\boldsymbol{y}}_{t_n,\tau}^{j-1}(\omega)\right\|^2 - \left\|\widehat{\boldsymbol{y}}_{t_n,0}^j(\omega) - \widehat{\boldsymbol{y}}_{t_n,0}^{j-1}(\omega)\right\|^2$$
$$= \int_0^\tau \left\|\widehat{\boldsymbol{y}}_{t_n,\tau'}^j(\omega) - \widehat{\boldsymbol{y}}_{t_n,\tau'}^{j-1}(\omega)\right\|^2 d\tau'$$
$$\quad + \int_0^\tau 2\left(\widehat{\boldsymbol{y}}_{t_n,\tau}^j(\omega) - \widehat{\boldsymbol{y}}_{t_n,\tau'}^{j-1}(\omega)\right)^\top \left[\boldsymbol{s}_{t_n+g_n(\tau')}^\theta\left(\widehat{\boldsymbol{y}}_{t_n,g_n(\tau')}^{j-1}(\omega)\right) - \boldsymbol{s}_{t_n+g_n(\tau')}^\theta\left(\widehat{\boldsymbol{y}}_{t_n,g_n(\tau')}^{j-2}(\omega)\right)\right] d\tau'$$
$$\leq 2\int_0^\tau \left\|\widehat{\boldsymbol{y}}_{t_n,\tau'}^j(\omega) - \widehat{\boldsymbol{y}}_{t_n,\tau'}^{j-1}(\omega)\right\|^2 d\tau' + \int_0^\tau \left\|\boldsymbol{s}_{t_n+g_n(\tau')}^\theta\left(\widehat{\boldsymbol{y}}_{t_n,g_n(\tau')}^{j-1}(\omega)\right) - \boldsymbol{s}_{t_n+g_n(\tau')}^\theta\left(\widehat{\boldsymbol{y}}_{t_n,g_n(\tau')}^{j-2}(\omega)\right)\right\|^2 d\tau'$$
$$\leq 2\int_0^\tau \left\|\widehat{\boldsymbol{y}}_{t_n,\tau'}^j(\omega) - \widehat{\boldsymbol{y}}_{t_n,\tau'}^{j-1}(\omega)\right\|^2 d\tau' + L_{\boldsymbol{s}}^2 \int_0^\tau \left\|\widehat{\boldsymbol{y}}_{t_n,g_n(\tau')}^{j-1}(\omega) - \widehat{\boldsymbol{y}}_{t_n,g_n(\tau')}^{j-2}(\omega)\right\|^2 d\tau'.$$

By Theorem A.4, and $\widehat{\boldsymbol{y}}_{t_n,0}^{j,p}(\omega) = \widehat{\boldsymbol{y}}_{t_{n-1},\tau_{n-1,M}}^j(\omega)$, we have

$$\left\|\widehat{\boldsymbol{y}}_{t_n,\tau}^j(\omega) - \widehat{\boldsymbol{y}}_{t_n,\tau}^{j-1}(\omega)\right\|^2 \leq L_{\boldsymbol{s}}^2 e^{2\tau} \int_0^\tau \left\|\widehat{\boldsymbol{y}}_{t_n,g_n(\tau')}^{j-1}(\omega) - \widehat{\boldsymbol{y}}_{t_n,g_n(\tau')}^{j-2}(\omega)\right\|^2 d\tau' + e^{2\tau}\Delta_{n-1}^j.$$

By taking expectation, for all $\tau \in [0, h_n]$

$$\mathbb{E}_{\omega\sim\bar{p}|\mathcal{F}_{t_n}}\left\|\widehat{\boldsymbol{y}}_{t_n,\tau}^j(\omega) - \widehat{\boldsymbol{y}}_{t_n,\tau}^{j-1}(\omega)\right\|^2 - e^{2\tau}\Delta_{n-1}^j$$
$$\leq L_{\boldsymbol{s}}^2 e^{2\tau} \int_0^\tau \mathbb{E}_{\omega\sim\bar{p}|\mathcal{F}_{t_n}}\left\|\widehat{\boldsymbol{y}}_{t_n,g_n(\tau')}^{j-1}(\omega) - \widehat{\boldsymbol{y}}_{t_n,g_n(\tau')}^{j-2}(\omega)\right\|^2 d\tau'$$
$$\leq L_{\boldsymbol{s}}^2 e^{2\tau}\tau \sup_{\tau'\in[0,\tau]} \mathbb{E}_{\omega\sim\bar{p}|\mathcal{F}_{t_n}}\left\|\widehat{\boldsymbol{y}}_{t_n,\tau'}^{j-1}(\omega) - \widehat{\boldsymbol{y}}_{t_n,\tau'}^{j-2}(\omega)\right\|^2.$$

Thus

$$\sup_{\tau\in[0,h_n]} \mathbb{E}_{\omega\sim\bar{p}|\mathcal{F}_{t_n}}\left\|\widehat{\boldsymbol{y}}_{t_n,\tau}^{j-1}(\omega) - \widehat{\boldsymbol{y}}_{t_n,\tau}^{j-2}(\omega)\right\|^2$$
$$\leq e^{2h_n}\Delta_{n-1}^j + L_{\boldsymbol{s}}^2 e^{2h_n} h_n \mathcal{E}_n^{j-1}.$$

For $j = 1$, we consider the following two processes,

$$\mathrm{d}\widehat{\boldsymbol{y}}^1_{t_n,\tau}(\omega) = \left[\frac{1}{2}\widehat{\boldsymbol{y}}^1_{t_n,\tau}(\omega) + \boldsymbol{s}^\theta_{t_n+g_n(\tau)}\left(\widehat{\boldsymbol{y}}^0_{t_n,g_n(\tau)}(\omega)\right)\right]\mathrm{d}\tau + \mathrm{d}\boldsymbol{w}_{t_n+\tau}(\omega),$$

and

$$\mathrm{d}\widehat{\boldsymbol{y}}^0_{t_n,\tau}(\omega) = \left[\frac{1}{2}\widehat{\boldsymbol{y}}^0_{t_n,\tau}(\omega) + \boldsymbol{s}^\theta_{t_n+g_n(\tau)}\left(\widehat{\boldsymbol{y}}^0_{t_{n-1},\tau_{n-1,M}}(\omega)\right)\right]\mathrm{d}\tau + \mathrm{d}\boldsymbol{w}_{t_n+\tau}(\omega).$$

Similarly, we have

$$\sup_{\tau\in[0,h_n]} \mathbb{E}_{\omega\sim\bar{p}|\mathcal{F}_{t_n}} \left\|\widehat{\boldsymbol{y}}^1_{t_n,\tau}(\omega) - \widehat{\boldsymbol{y}}^0_{t_n,\tau}(\omega)\right\|^2$$

$$\le e^{2h_n}\Delta^1_n + L^2_{\boldsymbol{s}}e^{2h_n}h_n\left(\sup_{\tau\in[0,h_n]} \mathbb{E}_{\omega\sim\bar{p}|\mathcal{F}_{t_n}} \left\|\widehat{\boldsymbol{y}}^0_{t_n,\tau}(\omega) - \widehat{\boldsymbol{y}}^0_{t_{n-1},\tau_{n-1,M}}(\omega)\right\|^2\right).$$

$\square$

**Lemma C.12 (One-step decomposition of $\Delta^j_n$).** *Assume $L^2_{\boldsymbol{s}}e^{2h_n}h_n \le 0.01$ and and $e^{2h_n} \le 2$. For any $j = 2,\ldots,J$, $n = 1,\ldots,N-1$, we have*

$$\Delta^j_n \le 3\Delta^j_{n-1} + 0.4\mathcal{E}^{j-1}_n.$$

*Furthermore, for $j = 1$, $n = 1,\ldots,N-1$, we have*

$$\Delta^1_n \le 3\Delta^1_{n-1} + 0.4\sup_{\tau\in[0,h_n]} \mathbb{E}_{\omega\sim\bar{p}|\mathcal{F}_{t_n}}\left[\left\|\widehat{\boldsymbol{y}}^0_{n,\tau} - \widehat{\boldsymbol{y}}^0_{n-1,\tau_{n,M}}\right\|^2\right].$$

*For $n = 0$, we have $\Delta^j_0 \le 0.32\Delta^{j-1}_0$, and $\Delta^1_0 \le \sup_{\tau\in[0,h_0]} \mathbb{E}_{\omega\sim\bar{p}|\mathcal{F}_{t_0}}\left[\left\|\widehat{\boldsymbol{y}}^0_{t_0,\tau}(\omega) - \widehat{\boldsymbol{y}}^0_{t_0,0}(\omega)\right\|^2\right].$*

*Proof.* By definition of $\widehat{\boldsymbol{y}}^j_{t_n,\tau_{n,M}}(\omega)$ we have

$$\left\|e^{-\frac{h_n}{2}}\widehat{\boldsymbol{y}}^j_{t_n,\tau_{n,M}} - e^{-\frac{h_n}{2}}\widehat{\boldsymbol{y}}^{j-1}_{t_n,\tau_{n,M}}\right\|^2$$

$$= \left\|\widehat{\boldsymbol{y}}^j_{n,0} - \widehat{\boldsymbol{y}}^{j-1}_{n,0} + \sum_{m'=0}^{m-1} e^{\frac{-\tau_{n,m'+1}}{2}}2(e^{\epsilon_{n,m'}} - 1)\left[\boldsymbol{s}^\theta_{t_n+\tau_{n,m'}}(\widehat{\boldsymbol{y}}^{j-1}_{n,\tau_{n,m'}}) - \boldsymbol{s}^\theta_{t_n+\tau_{n,m'}}(\widehat{\boldsymbol{y}}^{j-2}_{n,\tau_{n,m'}})\right]\right\|^2$$

$$\le 2\left\|\widehat{\boldsymbol{y}}^j_{n,0} - \widehat{\boldsymbol{y}}^{j-1}_{n,0}\right\|^2 + 2\left\|\sum_{m'=0}^{m-1} e^{\frac{-\tau_{n,m'+1}}{2}}2(e^{\epsilon_{n,m'}} - 1)\left[\boldsymbol{s}^\theta_{t_n+\tau_{n,m'}}(\widehat{\boldsymbol{y}}^{j-1}_{n,\tau_{n,m'}}) - \boldsymbol{s}^\theta_{t_n+\tau_{n,m'}}(\widehat{\boldsymbol{y}}^{j-2}_{n,\tau_{n,m'}})\right]\right\|^2$$

$$\le 2\left\|\widehat{\boldsymbol{y}}^j_{n,0} - \widehat{\boldsymbol{y}}^{j-1}_{n,0}\right\|^2 + 32\epsilon^2_{n,m'}M\sum_{m'=0}^{M-1}\left\|\left[\boldsymbol{s}^\theta_{t_n+\tau_{n,m'}}(\widehat{\boldsymbol{y}}^{j-1}_{n,\tau_{n,m'}}) - \boldsymbol{s}^\theta_{t_n+\tau_{n,m'}}(\widehat{\boldsymbol{y}}^{j-2}_{n,\tau_{n,m'}})\right]\right\|^2$$

$$\le 2\left\|\widehat{\boldsymbol{y}}^j_{n,0} - \widehat{\boldsymbol{y}}^{j-1}_{n,0}\right\|^2 + 32h^2_n\sup_{\tau\in[0,h_n]}L^2_{\boldsymbol{s}}\left\|\widehat{\boldsymbol{y}}^{j-1}_{n,\tau} - \widehat{\boldsymbol{y}}^{j-2}_{n,\tau}\right\|^2,$$

where the second inequality is implied by that $e^x - 1 \le 2x$ when $x < 1$. By taking expectation, and the assumption that $L^2_{\boldsymbol{s}}e^{2h_n}h_n \le 0.1$ and $e^{2h_n} \le 2$, we have

$$e^{-\frac{h_n}{2}}\Delta^j_n = \mathbb{E}_{\omega\sim\bar{p}|\mathcal{F}_{t_n}}e^{-\frac{h_n}{2}}\left[\left\|\widehat{\boldsymbol{y}}^j_{t_n,\tau_{n,M}} - \widehat{\boldsymbol{y}}^{j-1}_{t_n,\tau_{n,M}}\right\|^2\right]$$

$$\le 2\mathbb{E}_{\omega\sim\bar{p}|\mathcal{F}_{t_n}}\left[\left\|\widehat{\boldsymbol{y}}^j_{n,0} - \widehat{\boldsymbol{y}}^{j-1}_{n,0}\right\|^2\right] + 32h^2_nL^2_{\boldsymbol{s}}\sup_{\tau\in[0,h_n]}\mathbb{E}_{\omega\sim\bar{p}|\mathcal{F}_{t_n}}\left[\left\|\widehat{\boldsymbol{y}}^{j-1}_{n,\tau} - \widehat{\boldsymbol{y}}^{j-2}_{n,\tau}\right\|^2\right]$$

$$\le 2\Delta^j_{n-1} + 0.32\mathcal{E}^{j-1}_n.$$

Thus

$$\Delta_n^j \leq 3\Delta_{n-1}^j + 0.4\mathcal{E}_n^{j-1}.$$

In the remaining part, we will bound $\Delta_n^1$. By definition, we have

$$\left\| e^{-\frac{h_n}{2}} \widehat{\boldsymbol{y}}_{t_n,\tau_{n,M}}^1(\omega) - e^{-\frac{h_n}{2}} \widehat{\boldsymbol{y}}_{t_n,\tau_{n,M}}^0(\omega) \right\|^2$$

$$= \left\| \widehat{\boldsymbol{y}}_{n,0}^1 - \widehat{\boldsymbol{y}}_{n-1,\tau_{n,M}}^0 + \sum_{m'=0}^{m-1} e^{\frac{-\tau_{n,m'+1}}{2}} 2(e^{\epsilon_{n,m'}} - 1) \left[ \boldsymbol{s}_{t_n+\tau_{n,m'}}^\theta (\widehat{\boldsymbol{y}}_{n-1,\tau_{n,m'}}^0) - \boldsymbol{s}_{t_n+\tau_{n,m'}}^\theta (\widehat{\boldsymbol{y}}_{n-1,\tau_{n,M}}^0) \right] \right\|^2$$

$$\leq 2 \left\| \widehat{\boldsymbol{y}}_{n,0}^1 - \widehat{\boldsymbol{y}}_{n-1,\tau_{n,M}}^0 \right\|^2 + 2 \left\| \sum_{m'=0}^{M-1} e^{\frac{-\tau_{n,m'+1}}{2}} 2(e^{\epsilon_{n,m'}} - 1) \left[ \boldsymbol{s}_{t_n+\tau_{n,m'}}^\theta (\widehat{\boldsymbol{y}}_{n-1,\tau_{n,m'}}^0) - \boldsymbol{s}_{t_n+\tau_{n,m'}}^\theta (\widehat{\boldsymbol{y}}_{n-1,\tau_{n,M}}^0) \right] \right\|^2$$

$$\leq 2 \left\| \widehat{\boldsymbol{y}}_{n,0}^1 - \widehat{\boldsymbol{y}}_{n-1,\tau_{n,M}}^0 \right\|^2 + 32 h_n^2 L_s^2 \sup_{\tau \in [0,h_n]} \left\| \widehat{\boldsymbol{y}}_{n-1,\tau}^0 - \widehat{\boldsymbol{y}}_{n-1,\tau_{n,M}}^0 \right\|^2,$$

where the second inequality is implied by that $e^x - 1 \leq 2x$ when $x < 1$. Thus with $L_s^2 e^{2h_n} h_n \leq 0.01$ and $e^{2h_n} \leq 2$, we have

$$e^{-\frac{h_n}{2}} \Delta_n^1 = \mathbb{E}_{\omega \sim \bar{p} | \mathcal{F}_{t_n}} e^{-\frac{h_n}{2}} \left[ \left\| \widehat{\boldsymbol{y}}_{t_n,\tau_{n,M}}^1 - \widehat{\boldsymbol{y}}_{t_n,\tau_{n,M}}^0 \right\|^2 \right]$$

$$\leq 2\mathbb{E}_{\omega \sim \bar{p} | \mathcal{F}_{t_n}} \left[ \left\| \widehat{\boldsymbol{y}}_{n,0}^1 - \widehat{\boldsymbol{y}}_{n-1,\tau_{n,M}}^0 \right\|^2 \right] + 32 h_n^2 L_s^2 \sup_{\tau \in [0,h_n]} \mathbb{E}_{\omega \sim \bar{p} | \mathcal{F}_{t_n}} \left[ \left\| \widehat{\boldsymbol{y}}_{n-1,\tau}^{1,P-1} - \widehat{\boldsymbol{y}}_{n-1,\tau_{n,M}}^0 \right\|^2 \right]$$

$$\leq 2\Delta_{n-1}^1 + 0.32 \sup_{\tau \in [0,h_n]} \mathbb{E}_{\omega \sim \bar{p} | \mathcal{F}_{t_n}} \left[ \left\| \widehat{\boldsymbol{y}}_{n,\tau}^0 - \widehat{\boldsymbol{y}}_{n-1,\tau_{n,M}}^0 \right\|^2 \right].$$

$\square$

Let $L_n^j = 2\Delta_{n-1}^j + 0.01\mathcal{E}_n^{j-1}$. We note that $L_n^j \geq \mathcal{E}_n^j$. Thus for $n \geq 1$ and $j \geq 2$,

$$L_n^j = 2\Delta_{n-1}^j + 0.01\mathcal{E}_n^{j-1}$$
$$\leq 2(80\Delta_{n-1}^j + 0.4\mathcal{E}_n^{j-1}) + 0.01L_n^j$$
$$\leq 160L_{n-1}^j + 0.01L_n^j. \tag{33}$$

We recursively bound $L_n^j$ as

$$L_n^j \leq \sum_{a=2}^n (0.01)^{j-2} 160^{n-a} \binom{n-a+j-2}{j-2} L_a^2 + \sum_{b=2}^j (0.01)^{j-b} 160^{n-1} \binom{n-1+j-b}{j-b} L_1^b.$$

**Bound for** $\sum_{a=2}^n (0.01)^{j-2} 160^{n-a} \binom{n-a+j-2}{j-2} L_a^2.$ Firstly, we bound $L_a^2$. To do so, by Lemma C.12, we bound $\Delta_n^1$ as

$$\Delta_n^1 \leq 3\Delta_{n-1}^1 + 4\mathcal{E}_I \leq 3^n \Delta_0^1 + \sum_{i=0}^{n-1} 4 \cdot 3^i \mathcal{E}_I \leq 4 \sum_{i=0}^n 3^i \mathcal{E}_I \leq 3^{n+2}\mathcal{E}_I.$$

and by Lemma C.11, bound $\mathcal{E}_n^1$ as

$$\mathcal{E}_n^1 \leq 2\Delta_n^1 + 0.1\mathcal{E}_I \leq 3^{n+3}\mathcal{E}_I.$$

Furthermore, by Lemma C.12, we bound $\Delta_n^2$ as

$$\Delta_n^2 \leq 3\Delta_{n-1}^2 + 0.4\mathcal{E}_n^1 \leq 3^n \Delta_0^2 + \sum_{i=0}^{n-1} 3^i \mathcal{E}_{n-i}^1 \leq 0.32 \cdot 3^n \mathcal{E}_I + 3^{n+3} n\mathcal{E}_I \leq 28 \cdot 3^n n\mathcal{E}_I.$$

Thus
$$L_a^2 = 2\Delta_{a-1}^2 + 0.01\mathcal{E}_a^1 \le 28 \cdot 3^a a \mathcal{E}_I.$$

Furthermore, by $\binom{m}{n} \le \left(\frac{em}{n}\right)^n$ for $m \ge n > 0$, we have

$$\sum_{a=2}^n (0.01)^{j-2} 160^{n-a} \binom{n-a+j-2}{j-2} L_a^2$$

$$\le (0.01)^{j-2} (28 \cdot 160^n n^2) e^{j-2} \left(\frac{n-a+j-2}{j-2}\right)^{j-2} \mathcal{E}_I$$

$$\le (e^2 \cdot 0.01)^{j-2} (28 \cdot 160^n n^2) \mathcal{E}_I.$$

**Bound for** $\sum_{b=2}^j (0.01)^{j-b} 160^{n-1} \binom{n-1+j-b}{j-b} L_1^b$**.** By Lemma C.11, we have

$$\mathcal{E}_1^j \le 0.01 \mathcal{E}_1^{j-1} + 2\Delta_0^j$$

$$\le (0.01)^j \mathcal{E}_I + \sum_{i=0}^{j-1} (0.01)^i 2\Delta_0^{j-i}.$$

Combining the fact that $\Delta_0^j \le 0.32^{j-1}\mathcal{E}_I$, we have

$$\mathcal{E}_1^j \le 7 \cdot j \cdot 0.32^j \mathcal{E}_I.$$

Thus

$$L_1^b = 2\Delta_0^j + 0.01\mathcal{E}_1^{b-1}$$

$$\le 2 \cdot 0.32^{b-1}\mathcal{E}_I + 0.01 \cdot 7 \cdot (b-1) \cdot 0.32^{b-1}\mathcal{E}_I$$

$$\le 7 \cdot b \cdot 0.32^{b-1}\mathcal{E}_I.$$

Furthermore, by $\sum_{i=0}^m \binom{n+i}{n} x^i = \frac{1-(m+1)\binom{m+n+1}{n} B_x(m+1,n+1)}{(1-x)^{n+1}} \le \frac{1}{(1-x)^{n+1}}$, we have

$$\sum_{b=2}^j (0.01)^{j-b} 160^{n-1} \binom{n-1+j-b}{n-1} L_1^b$$

$$\le \sum_{b=2}^j (0.01)^{j-b} 160^{n-1} \binom{n-1+j-b}{n-1} 7 \cdot b \cdot 0.32^{b-1}\mathcal{E}_I$$

$$\le 22 \cdot 0.87^j 440^{n-1} j \mathcal{E}_I.$$

Combining the above two results, we have
$$\mathcal{E}_n^J \le (e^2 \cdot 0.01)^{j-2} (28 \cdot 160^n n^2) \mathcal{E}_I + 22 \cdot 0.87^j 440^{n-1} j \mathcal{E}_I.$$

If $J - 45N \gtrsim \log \frac{N\mathcal{E}_I}{\varepsilon^2}$, for any $n = 0, \ldots, N$

$$\mathcal{E}_n^J \le \frac{\varepsilon^2}{N}. \tag{34}$$

### C.3.1. OVERALL ERROR BOUND

By the previous computation, we have

$$\mathsf{KL}(\bar{p}_{t_{n+1}} \| \widehat{q}_{t_{n+1}})$$

$$\le \mathsf{KL}(\bar{p}_{t_n} \| \widehat{q}_{t_n}) + \mathbb{E}_{\omega \sim q|_{\mathcal{F}_{t_n}}} \left[\frac{1}{2} \int_0^{h_n} \|\boldsymbol{\delta}_{t_n}(\tau, \omega)\|^2 \, d\tau\right]$$

$$\le \mathsf{KL}(\bar{p}_{t_n} \| \widehat{q}_{t_n}) + 3\mathbb{E}_{\omega \sim \bar{p}|_{\mathcal{F}_{t_n}}} \left[A_{t_n}(\omega) + B_{t_n}(\omega)\right] + 3L_s^2 h_n \mathcal{E}_n^J.$$

Combining Lemma A.5, Corollary C.10, and Eq. (34), we have

$$
\begin{aligned}
&\mathsf{KL}(\breve{p}_{t_{n+1}} \| \widehat{q}_{t_{n+1}}) \\
&\leq \mathsf{KL}(\breve{p}_0 \| \widehat{q}_0) + 3 \sum_{n=0}^{N-1} \left( \mathbb{E}_{\omega \sim \breve{p}|_{\mathcal{F}_{t_n}}} \left[ A_{t_n}(\omega) + B_{t_n}(\omega) \right] + L_{\boldsymbol{s}}^2 h_n \mathcal{E}_n^J \right) \\
&\lesssim d e^{-T} + \epsilon d (T + \log \eta^{-1}) + \delta_2^2 + \varepsilon^2, \\
&\lesssim \varepsilon^2,
\end{aligned}
$$

with parameters $J - 45N \geq \mathcal{O}(\log \frac{Nd}{\varepsilon^2})$, $h = \Theta(1)$, $N = \mathcal{O}(\log \frac{d}{\varepsilon^2})$, $T = \mathcal{O}(\log \frac{d}{\varepsilon^2})$ $\epsilon = \Theta(d^{-1}\varepsilon^2 \log^{-1} \frac{d}{\varepsilon^2})$, $M = \mathcal{O}(d\varepsilon^{-2} \log \frac{d}{\varepsilon^2})$, $\log \eta^{-1} \lesssim T$.

