# OpenReview forum: "Parallel Simulation for Log-concave Sampling and Score-based Diffusion Models"
_ICML.cc/2025/Conference — ICML 2025 spotlightposter_

### Official Review · Reviewer_aCjg · 2025-03-07

**Overall Recommendation:** 3

**Summary:**

The authors of this paper study two separate but related problems: sampling from an isoperimetric distribution and also sampling from a score-based diffusion model.  Under
some assumptions on the target distribution $\pi$---e.g. it satisfies a
log-Sobolev inequality (isoperimetric) and is $\beta$-log smooth---the
authors demonstrate an approach to generate samples arbitrarily close to being
sampled from the isoperimetric target or a score-based diffusion model that only takes
$O(\log(d/\epsilon^2))$ iterations, while the current state of the art takes
$O(\log^2(d/\epsilon^2))$ iterations. This result assumes that the number of
cores available to parallelize computation is $\Theta(d/\epsilon^2)$, but
they provide analogous results in the case they are bottlenecked on the
number of cores. Their results also typically require more space than some
alternatives, which they attribute to the more volatile paths of the
overdamped Langevin diffusion vis-a-vis the underdamped Langevin
diffusion. E.g., alternatives usually require
$\tilde{O}(d^{1.5}/\epsilon^2)$ space while this approach requires
$\tilde{O}(d^2/\epsilon^2)$.

The paper is purely theoretical in nature, and to prove their above result,
they build upon previous work that uses Picard iterations to subdivide the
sampling problem across time. However, rather treating each subdivided
segment as independent and parallelizing across the different stages, the
authors propose a method that iterates over segments based on both the previous time slice and previous stage of the current time slice to propagate their approximations.

## update after rebuttal
Based on the authors feedback and other reviews, I'm inclined to keep my score of a weak accept mostly out of my lack of confidence.

**Claims And Evidence:**

There are no empirical studies to back up their claim. That being said, the
authors provide a sketch of the proof in Section 3 and also a
detailed proof of their main claims in the appendix. E.g., Theorem 4.2, which establishes
the superior rate (in terms of iterations) for approximately sampling from an isoperimetric
distribution, is proved in Appendix B, while Theorem 5.4, which establishes a similar result but for approximately sampling from a score-based diffusion model, is tackled in Appendix C.

**Essential References Not Discussed:**

To the best of my knowledge, no.

**Experimental Designs Or Analyses:**

Again, there are no empirical results in this paper.

**Methods And Evaluation Criteria:**

There are no empirical analyses conducted in this paper.

**Other Comments Or Suggestions:**

L128: capitalized "The"

L144: Might have missed it, but what's V?

L183: Is the integral correct? Seems like the $f_t$ should be $f_s$?

L194: "girds"

**Other Strengths And Weaknesses:**

The paper's core contribution seems to be reordering the manner in which the
Picard iterations are taken to approximate sampling from the path of the
SDE. This appears to be an interesting idea, although the significance is a
bit tempered by the extra space complexity needed to execute this approach.

The paper is only theoretical in nature, so it is hard to state its
practical significance. It's case would be greatly bolstered if there were empirical
work demonstrating its efficacy. That being said, it appears to produce results on
the Pareto frontier of time and space complexity for sampling from an SDE.

The paper is fairly well written, although there are a few mistakes here and
there.

**Questions For Authors:**

[Q1] Are there any ways to empirically validate whether the approaches
outlined here hold? If so, could they be added to the paper and compared
against alternatives?

**Relation To Broader Scientific Literature:**

The Algorithm 1 presented in the paper could provide a way to speed up the
algorithms that sample from diffusion-based methods when the number of
compute resources (e.g. RAM and cores) is high. If demonstrated empirically,
this could have a substantial effect on the genAI industry.

**Theoretical Claims:**

No, I only skimmed the proofs in the appendix, these were not checked
line-by-line.

---

> ### Author Rebuttal · Authors · 2025-03-31
>
> We thank the reviewer for their detailed and constructive feedback. We are encouraged by your recognition of the core contribution—**the diagonal reordering of Picard iterations to achieve improved parallel iteration complexity, the soundness of the theoretical analysis,
> and the potential applicability of our approach in high-compute environments (e.g., with sufficient RAM and cores)**. We address the specific concerns and questions raised below:
>
> **"Are there any ways to empirically validate whether the approaches outlined here hold? If so, could they be added to the paper and compared against alternatives?"**
>
> We agree that empirical validation is important. While we are not aware of existing implementations of our diagonal update scheme, we believe it could be adapted to diffusion-based samplers by modifying their update scheduling. We will add a discussion on this potential direction in the revised version.
>
> **Regarding typos:**
> We will carefully proofread the manuscript and correct all grammatical and typographical errors to improve the clarity and overall presentation in the revised version.

---

### Official Review · Reviewer_KnEh · 2025-03-11

**Overall Recommendation:** 5

**Summary:**

The proposed paper introduces a novel parallel sampling technique that significantly enhances the time complexity of both sampling under isoperimetry and score-based diffusion model inference problems from $O(\log^2 d)$ to $O(\log d)$. The primary algorithmic innovation lies in the parallel sampling across time slices, rather than sequentially updating each time slice. Furthermore, the authors provide a corresponding lower bound, demonstrating the optimality of their proposed algorithm.

**Claims And Evidence:**

The assertions regarding the algorithmic and technical novelty, as well as the enhancement in time complexity, are substantiated, well-defined, and compelling.

**Essential References Not Discussed:**

The literature review is comprehensive.

**Experimental Designs Or Analyses:**

This is theoretical work and no experiment is involved.

**Methods And Evaluation Criteria:**

The proposed parallel sampling algorithm appears to be logically sound and presents a promising avenue for enhancing the time complexity, thereby making a significant contribution to the field.

**Other Comments Or Suggestions:**

It would be more beneficial to provide additional intuitive explanations and justifications for the validity of the claim $L_n^j \leq a L_{n-1}^j + b L_n^{j-1}$. It appears that only some intuition for Problem (a) is presented in the main content, which still lacks certain details (“make use of the contraction of gradient descent”). Consequently, readers may require a thorough exploration of the proofs to gain a comprehensive understanding of the algorithm. In my opinion, this formula is the essence of the algorithm and even the entire paper. It would be preferable to explicitly express $a$, $b$, and (in the presence of $\delta^2$) $c$ in terms of the length of the time slice, Lipschitz constant, and other relevant parameters. Furthermore, it would be beneficial to provide additional discussions on the selection of $J$, $N$, and $P$ (which is not adequately explained in the main content).

**Other Strengths And Weaknesses:**

The paper is well-structured, effectively summarizing the related works, proposing the novel algorithm, and presenting results in a mathematically rigorous manner. The authors have also included numerous explanatory remarks throughout the paper to facilitate comprehension.

**Questions For Authors:**

- Can the space complexity be improved from $O(d^2)$ to $O(d^{3/2})$?
- Is it possible to extend this parallel sampling technique to discrete diffusion models?

**Relation To Broader Scientific Literature:**

The work is well-positioned in the literature. The $O(\log d)$ time complexity is clearly an improvement from $O(\log^2 d)$ in [1] in the context of diffusion models and in [2] in the context of sampling.

[1] Chen, Haoxuan, et al. "Accelerating diffusion models with parallel sampling: Inference at sub-linear time complexity." Advances in Neural Information Processing Systems 37 (2024): 133661-133709.
[2] Anari, Nima, Sinho Chewi, and Thuy-Duong Vuong. "Fast parallel sampling under isoperimetry." The Thirty Seventh Annual Conference on Learning Theory. PMLR, 2024.

**Theoretical Claims:**

I have reviewed the appendices for the proofs, and they appear to be sound.

---

> ### Author Rebuttal · Authors · 2025-03-31
>
> Thank you for your detailed and positive feedback. We greatly appreciate your recognition of **our novel parallel sampling technique, the clarity of the paper’s structure, the soundness of the theoretical analysis, and the potential of our approach to substantially improve time complexity and contribute meaningfully to the field**. Below, we address your specific suggestions and questions:
>
>
>
> **"...provide additional intuitive explanations and justifications for the validity of the claim  $L_n^j\leq aL_{n−1}^j+bL_{n}^{j−1}$...selection of $J$, $N$, and $P$..."**
>
> We appreciate your thoughtful observation that this recurrence forms the core of our analysis. Recall we denote the truncation error at the $n$-th time slice and the $j$-th iteration as $L_n^j$ in the squared sense.
>
> - **For Problem (b):** by Young’s inequality and the definition of the exponential scheme, we can bound the change of $L_n^j$ in one update as
> $$
> L_n^j = a L_{n-1}^j + b L_n^{j-1}
> $$
> with $a = O(e^{h/2})$ and $b = O(L^2 e^{h/2} h^2)$. Thus, by choosing the time step $h$ sufficiently small relative to the Lipschitz constant $L$, we ensure convergence along the Picard direction (indexed by $j$). Specifically, this choice guarantees that $b < 1$ and $a > 1$, with $a$ remaining bounded by a constant.
>
> - **For Problem (a):** since the score itself is not Lipschitz smooth, an additional score estimation error term $c \delta^2$ appears in the one-step update of $L_n^j$. To ensure the total score estimation error remains bounded, it is necessary to have $a, b < 1$. Similarly, by Young’s inequality and the definition of the Euler–Maruyama scheme, we can bound the change of $L_n^j$ as
> $$
> L_n^j = a L_{n-1}^j + b L_n^{j-1} + c \delta^2
> $$with$$
> a = 1 - 0.1 \frac{\beta h}{\kappa} + O(\kappa)(3 \beta^2 h^2)^P,$$$$
> b = O((\beta^2 h^2)^P), \quad
> c = O(\kappa \delta^2 h^2)
> $$Here, intuitively, $a$ comes from the contraction of the gradient mapping with an additional term from the Picard direction, $b$ reflects convergence along the Picard direction, and $c$ accounts for the accumulation of score estimation error $\delta$ over time length $h$, with an additional scaling by $\kappa$ due to Young’s inequality.
>
> - **On the choice of $J$, $N$, and $P$**:
> To ensure convergence along the Picard direction, we choose the step size as $h \approx 1/\beta$. The total time horizon is $\alpha \log(d/\varepsilon^2)$ for Problem (a), and $\log(d/\varepsilon^2)$ for Problem (b), leading to a total of $N = O(\log(d/\varepsilon^2))$ time slices in both cases.
> Since Picard iterations converge exponentially fast, it is sufficient for each time slice to undergo $O(\log(d/\varepsilon^2))$ updates to ensure the desired overall accuracy. This implies that the total number of diagonal iterations satisfies $J = N + O(\log(d/\varepsilon^2))$.
> For the parameter $P$, which controls the number of steps within each blockwise Picard update, we require $a < 1$ in the recurrence relation to ensure contraction in Problem (a). This imposes the condition $P = \Theta(\log \kappa).$
>
> **"Can the space complexity be improved from $O(d^2)$ to $O(d^{3/2})$?"**
>
> Yes—if only total variation (TV) convergence is required, applying our method to underdamped Langevin dynamics or ODE-based implementations of diffusion models is sufficient to achieve a space complexity of $O(d^{3/2})$. However, extending this to obtain similar guarantees under KL divergence remains an open question, which we leave as an important direction for future work.
>
>
> **"Is it possible to extend this parallel sampling technique to discrete diffusion models?"**
>
> This is a great question. At a high level, our method operates by regrouping discretized grids along the time horizon and updating them in a diagonal fashion. We believe this diagonal scheduling approach offers a general framework for enabling parallelism along the time direction. While adapting the method to discrete diffusion models may require specific modifications, we believe the core strategy remains applicable and represents a promising direction for future work.

---

> > ### Comment · Reviewer_KnEh · 2025-04-02
> >
> > I would like to express my gratitude to the authors for their response. Upon implementing the reviewers' suggestions and clarifying certain parts of the paper, which would enhance potential readers' comprehension of their algorithm and proof methodology, I believe this work has made a significant contribution to the field. Consequently, I have revised my evaluation from 4 (Accept) to 5 (Strong Accept).

---

> > > ### Author Response · Authors · 2025-04-03
> > >
> > > Thank you for your kind feedback and updated evaluation. We truly appreciate your support!

---

### Official Review · Reviewer_Q5PH · 2025-03-15

**Overall Recommendation:** 4

**Summary:**

The paper obtains novel rates for the parallel complexity of sampling, both in the gradient oracle (MCMC) setting and the score-based denoising setting. The rates are O(log d/eps), which is sharp (in epsilon). The proof is based on a refined Picard iteration scheme, with careful analysis in order to obtain the rate.

**Claims And Evidence:**

The authors claim a parallel complexity of O(log d/varepsilon) for MCMC under standard assumptions, and for score-based models (SDE variant) under Lipschitz score (which is one of the standard settings). The authors provide detailed proofs for their claims.

**Essential References Not Discussed:**

As far as I am aware, the most relevant references have been covered.

**Experimental Designs Or Analyses:**

The paper is not empirical in nature, and so does not contain experiments.

**Methods And Evaluation Criteria:**

This is not applicable to the paper.

**Other Comments Or Suggestions:**

Line 129 is unfinished.
Line 425: parallalizable -> parallelizable

**Other Strengths And Weaknesses:**

The primary weakness of this result is that the analysis feels somewhat like an incremental result, utilizing the same Picard scheme (although improved) compared to prior work.

Nonetheless, the improvement in logarithmic factor (which is highly important in the parallel setting) guarantees a tight rate, and the work will be useful for future follow-ups concerning the underdamped Langevin or ODE variant of the score-based generative model.

Thus, I think this work has merit and should be accepted as it stands. Nonetheless, if the authors are capable it would be helpful to supplement this work with any additional results.

**Questions For Authors:**

A more thorough discussion of condition number (kappa) dependence in both time and space would be appreciated, with reference to prior work.

How difficult is it to handle the underdamped Langevin dynamics? What are the barriers and why can the current analysis not immediately give results?

Can the Lipschitz score estimation assumption be removed, similar to the work from Benton et al.?

**Relation To Broader Scientific Literature:**

The primary references are discussed; this result is related to the parallel complexity of MCMC samplers studied in Amari et al. and subsequent works. Whereas those works obtained $\log^2(d/\varepsilon)$ rates, the present rate obtains the same with a single logarithm.

**Theoretical Claims:**

As discussed, the theoretical claims relate to the parallel complexity, which ultimately arises from a detailed analysis of a Picard iteration scheme. The results seem rigorous on a brief skim.

---

> ### Author Rebuttal · Authors · 2025-03-31
>
> Thank you for your thoughtful comments and positive evaluation of our work, particularly your recognition of **the sharp iteration complexity in terms of $\varepsilon$ and the rigor of our analysis**. We also appreciate your conclusion that the work merits acceptance. Below, we address your questions and comments in detail:
>
> **"A more thorough discussion of condition number (kappa) dependence in both time and space would be appreciated, with reference to prior work."**
>
> Thank you for pointing this out. We will revise the paper to include a more explicit discussion of how the condition number $\kappa$ affects both the step size and the overall iteration complexity. In particular:
>
> - **For iteration complexity:**
> Our method achieves an iteration complexity of $O(\kappa)$, which matches the state-of-the-art sequential query complexity established by [AC24]. In the parallel setting, we believe it is inherently difficult to break the $O(\kappa)$ barrier using the Picard method, as the length of each time slice must scale inversely with the smoothness in order to preserve non-expansiveness along the Picard direction.
>
> - **For space complexity:**
> Our method builds on the regrouping of scheduled grids along the time direction, and such schedual was proposed by [VW19]. Since we update all grids simultaneously in the worst case, the space complexity is given by $d N_{\text{seq}} = d \kappa^2$, where $N_{\text{seq}} = \kappa^2$ is the number of time steps required in the sequential setting.
>
> [AC24] Shifted Composition III: Local Error Framework for KL Divergence, Jason M. Altschuler, Sinho Chewi, 2024
>
> [VW19] Santosh Vempala and Andre Wibisono. Rapid convergence of the unadjusted Langevin algorithm: isoperimetry suffices, 2019
>
> **"How difficult is it to handle the underdamped Langevin dynamics? What are the barriers and why can the current analysis not immediately give results?""**
>
> Applying our method to underdamped Langevin dynamics, as in [ACV24], yields $\log(d/\varepsilon^2)$ iteration complexity and improved space complexity $d^{3/2}/\varepsilon^2$. However, due to the failure of the triangle inequality for KL divergence, our current analysis cannot establish optimal iteration complexity for KL convergence—our main goal—so we leave this as future work.
>
> [ACV24] Fast parallel sampling under isoperimetry. Nima Anari, Sinho Chewi, Thuy-Duong Vuong, 2024
>
> **"Can the Lipschitz score estimation assumption be removed, similar to the work from Benton et al.?"**
>
> We believe not, at least for Picard-type methods, as the Lipschitz condition is crucial to ensure convergence along the Picard direction.

---

> > ### Comment · Reviewer_Q5PH · 2025-04-02
> >
> > I thank the authors for their remarks. I would note that, if it is possible to obtain some iteration complexity in KL(discretized process | true process), this could still be of value as it would already give some improvements to the parallel complexity in total variation.
> >
> > Furthermore, the Lipschitz score estimation assumption is typically tackled by a varying step-size schedule. Perhaps a similar idea could be attempted here?

---

> > > ### Author Response · Authors · 2025-04-03
> > >
> > > - Thank you for the remark. We agree that establishing iteration complexity in terms of KL(discretized process | true process) would indeed be valuable, as it can yield improvements in the parallel complexity under total variation as well. We believe this is a promising direction for future work.
> > >
> > > - Good question! A varying step-size schedule is a standard approach for handling large Lipschitz constants, as discussed in [YFZS24]. In our setting, if the scheduled time points are regrouped so that each time slice has length bounded by $O(1/\text{smoothness})$, convergence along the Picard direction can be maintained. Under this adjustment, our diagonal-style update remains applicable and effective. More broadly, we believe our regroup-and-diagonal-update framework can be applied to any schedule satisfying this time-slice condition.
> > >
> > > [YFZS24] Lipschitz Singularities in Diffusion Models, Zhantao Yang et al., 2024.

---

### Official Review · Reviewer_qiak · 2025-03-21

**Overall Recommendation:** 4

**Summary:**

Parallel sampling methods propose to speed up sampling by more efficiently simulating diffusions. Prior work splits up the simulation interval into $\log(d)$ chunks and performs $\log(d)$  iterations on each chunk sequentially. This yields an overall complexity of $\log(d)^2$. This paper proposes to remove the need for the $\log(d)$ sequential steps by communicating between the chunks diagonally as illustrated in their Figure 1. Two close settings of simulating a diffusion with an approximate score are analyzed in the paper: the overdamped langevin diffusion for sampling from un-normalized densities and the reverse diffusion of an OU process. In both cases, the authors show that the $\log(d)$ sequential steps are unnecessary if the chunks communicate diagonally.

## update after rebuttal

The authors have fixed their misstated theorem and have clarified their contribution. I think their diagonal update scheme is interesting and I agree that it does introduce some differences with prior analyses. I raise my score as a consequence.

**Claims And Evidence:**

The paper claims are the following:
1. Theorem 4.2: We can sample from smooth distributions verifying the log-sobolev inequality in $\log(d/\epsilon^2)$ steps by simulating the overdamped langevin diffusion with the diagonal communication scheme in Figure 1. Evidence: The authors provide proofs for this claim by closely extending the work of Anari et al 2024.
2. Theorem 5.4: We can simulate the reverse diffusion with a learnt, uniformly Lipschitz score, in $\log(d/\epsilon^2)$ steps. Evidence: The authors closely build on the proofs of Chen et al 2024 to show that their diagonal communication scheme obviates the need for $\log(d)$ sequential steps.

**Essential References Not Discussed:**

.

**Experimental Designs Or Analyses:**

.

**Methods And Evaluation Criteria:**

.

**Other Comments Or Suggestions:**

.

**Other Strengths And Weaknesses:**

The paper is very well written. The statements of problems a and b could be made more precise by explicitly stating the simulated diffusion.
The only minor weakness I see is the (understandable) closeness to prior work and the lack of discussion on the assumptions:
- The $L_\infty$ score approximation for problem a is in my view an unnecessary notational burden and does not capture any realistic setting. Approximate scores are either approximate in weaker metrics or are more common in the problem b setting.
- The uniform Lipschitz assumption in problem b is very strong. The Lipschitz constant of the score can (or, in cases with well separated mode, always) depends on the dimension and it would be beneficial to discuss it as it affects the stepsize $h$ if my understanding is correct.

**Questions For Authors:**

1. Could the authors clarify if their theorem 4.2 is misstated? Is it missing the strong-log-concavity assumption?
2. Could the authors expand on the possible dimension dependence on the uniform Lipschitz assumption and its effects on their bound?

**Relation To Broader Scientific Literature:**

The related work is discussed well. The work consists of very close modifications of prior analyses to incorporate the diagonal communication scheme.

**Theoretical Claims:**

Issues with Theorem 4.2: It appears that theorem 4.2 is misstated. A crucial ingredient seems to be the gradient mapping contraction which is used in lemma B.5. This only holds for strongly log-concave distributions. If my understanding is correct, Theorem 4.2 does not apply to the broader class of smooth LSI distributions. If my understanding is wrong, I kindly ask the authors to clarify.

---

> ### Author Rebuttal · Authors · 2025-03-31
>
> Thank you for your insightful questions and appreciation of our work, particularly recognizing the clarity of the writing and **the effectiveness of the diagonal communication scheme for parallel sampling**. Below, we address your questions and comments:
>
> **"...gradient mapping contraction...missing the strong-log-concavity assumption?"**
>
> You're absolutely right—the contraction property used in Lemma B.5 indeed requires the strong log-concavity of the target distribution. We will revise Theorem 4.2 to explicitly state this assumption and clarify its role in ensuring convergence under the diagonal update scheme. Additionally, we will update all related claims and discussions throughout the paper to consistently reflect this assumption.
>
> **"...by closely extending the work of Anari et al 2024... build on the proofs of Chen et al 2024... (understandable) closeness to prior work..."**
>
> While we adopt several standard tools in parts of our analysis—such as interpolation arguments (e.g., VW19 or Section 4.2 of Che23), and Girsanov-based comparisons in Appendix C.2 for the diffusion model—we emphasize that our work addresses analytical challenges not present in prior literature.
>
> In particular, our **diagonal update scheme** introduces **changing start points**,  which presents analytical challenges not encountered in prior work. As discussed in Section 3, this required new analysis to control error propagation along the Picard iteration and time direction. As Reviewer KnEh pointed, the recurrence
> $$
> L_n^j = a L_{n-1}^j + b L_n^{j-1} + c \delta^2
> $$
> captures the core of our theoretical analysis. We appreciate this insight and will provide additional intuition for this recurrence, highlighting its importance and distinguishing our analytic contributions from previous work.
>
> [VW19] Rapid convergence of the unadjusted langevin algorithm: Isoperimetry suffices.
>
> [Che23] Sinho Chewi. Log-concave sampling.
>
> [CRYR24] Accelerating Diffusion Models with Parallel Sampling: Inference at Sub-Linear Time Complexity
>
> **"The L∞ score approximation... unnecessary notational burden... does not capture any realistic setting..."**
>
> We would like to clarify that the $L^\infty$ approximation follows the same formulation as in [ACV24]. As discussed in their paper, this form of approximation is natural in the context of discrete sampling, where approximate scores can be efficiently computed in $O(1)$ iterations using access to a weighted counting oracle or Laplace transform. Furthermore, in spin glass models, the score function can be approximated in the $L^\infty$ norm using approximate message passing as shown in [AM22].
>
> Moreover, since the final accumulated error in our method depends on the score approximation error at each queried point with varying weights, a natural refinement is to define a weighted version of Assumption 5.1. Also, another alternative relaxation is to adopt a bias-variance decomposition, as in Assumption 4 of [BCESZ22], though it is still formulated in an $L^\infty$-style metric.
>
> [ACV24] Fast parallel sampling under isoperimetry.
>
> [BCESZ22] Towards a Theory of Non-Log-Concave Sampling: First-Order Stationarity Guarantees for Langevin Monte Carlo
>
> [AM22] Sampling from the Sherrington-Kirkpatrick Gibbs measure via algorithmic stochastic localization
>
> **"The Lipschitz constant of the score... affects the stepsize $h$...Could the authors expand on the possible dimension dependence on the uniform Lipschitz assumption and its effects on their bound?"**
>
> We acknowledge that the uniform Lipschitz assumption in Problem (b) is indeed strong; however, it is a common simplification adopted in prior works [CRYR24], [CCLL22]. As discussed in [SBDD22], the Lipschitz constant must be large to capture multimodal distributions, reflecting a trade-off between model expressivity and training stability. In particular, in settings with well-separated modes, the score's Lipschitz constant can grow significantly, even becoming unbounded near the zero point [YFZS24].
>
> To ensure convergence of the Picard iterations under these conditions, the product $L_s^2 e^{h_n} h_n$ must be sufficiently small. This requirement implies that the length of each time slice, $h_n$, should scale as $O(1/L_s^2)$. Consequently, the number of time slices becomes $N = O(L_s^2 \log d)$, leading to an **overall iteration complexity of $O(L_s^2 \log d)$**.
>
> Furthermore, our regrouping of time grids provides a flexible framework that could help mitigate Lipschitz singularities in other diffusion models, such as E-TSDM, as studied in [YFZS24]. We will expand on this point in the revised version.
>
> [CRYR24] Accelerating Diffusion Models with Parallel Sampling: Inference at Sub-Linear Time Complexity
>
> [CCLL22] Sampling is as easy as learning the score: theory for diffusion models with minimal data assumptions
>
> [YFZS24] Lipschitz Singularities in Diffusion Models
>
> [SBDD22] Can Push-forward Generative Models Fit Multimodal Distributions?

---

### Decision · Program_Chairs · 2025-05-01

**Decision:**

Accept (spotlight poster)

**Comment:**

This paper presents a novel parallel sampling algorithm that reduces the iteration complexity from O(\log^2 d) to O(\log d) for sampling from isoperimetric and diffusion-based models. The key innovation is the diagonal reordering of Picard iterations, which removes the need for O(\log d) sequential steps.

All reviewers agree that the paper makes a strong theoretical contribution to the field of parallel sampling. The reduction in complexity is both significant and impactful. While an empirical evaluation would strengthen the paper, the novelty and rigor of the work are clear.